# Escaping Saddle Points with Bias-Variance Reduced Local Perturbed SGD for Communication Efficient Nonconvex Distributed Learning

**Tomoya Murata**[*]
NTT DATA Mathematical Systems Inc., Tokyo, Japan
Graduate School of Information Science and Technology, The University of Tokyo

**Taiji Suzuki**[†]
Graduate School of Information Science and Technology, The University of Tokyo
Center for Advanced Intelligence Project, RIKEN, Tokyo, Japan

## Abstract

In recent centralized nonconvex distributed learning and federated learning, local methods are one of the promising approaches to reduce communication time. However, existing work has mainly focused on studying first-order optimality guarantees. On the other side, second-order optimality guaranteed algorithms, i.e., algorithms escaping saddle points, have been extensively studied in the non-distributed optimization literature. In this paper, we study a new local algorithm called Bias-Variance Reduced Local Perturbed SGD (BVR-L-PSGD), that combines the existing bias-variance reduced gradient estimator with parameter perturbation to find second-order optimal points in centralized nonconvex distributed optimization. BVR-L-PSGD enjoys second-order optimality with nearly the same communication complexity as the best known one of BVR-L-SGD to find first-order optimality. Particularly, the communication complexity is better than non-local methods when the local datasets heterogeneity is smaller than the smoothness of the local loss. In an extreme case, the communication complexity approaches to $\widetilde{\Theta}(1)$ when the local datasets heterogeneity goes to zero. Numerical results validate our theoretical findings.

## 1 Introduction

Distributed learning is an attractive approach to reduce the total execution time by utilizing the parallel computations. However, the communication time in distributed learning can be a main bottleneck in the entire process due to huge parameter size typical in deep learning or low bandwidth communication environments.

To reduce communication time, one of the promising approaches is the usage of local methods such as local SGD (also called as Parallel Restart SGD or FedAvg). In local SGD, each worker independently executes multiple updates of the local model based on his own local dataset, and the server periodically communicates and aggregates the local models. Many paper have studied local SGD [25, 30, 7, 6, 15, 13, 28, 27]. Particularly, for convex objectives, it has been shown in [27], for the first time, the communication complexity (that is the necessary number of communication rounds to achieve given desired optimization error) of local SGD can be smaller than the one of minibatch

---

[*]murata@msi.co.jp
[†]taiji@mist.i.u-tokyo.ac.jp

36th Conference on Neural Information Processing Systems (NeurIPS 2022).

SGD when the *heterogeneity* of the local datasets is extremely small. In traditional distributed learning, the local datasets are typically random subsets of the global dataset and in this case the heterogeneity of the local datasets may become quite small. However, in the recent federated learning regimes [16, 24, 18], it is often the case that the heterogeneity of the local datasets is not too small. Also, the analysis in [27] has only focused on convex cases. Hence, the superiority of local SGD to minibatch SGD is still quite limited.

Recently, more communication efficient local methods than local SGD have been proposed for possibly nonconvex objectives to guarantee first-order optimality [12, 22, 19]. SCAFFOLD [12] is a new local algorithm based on the idea of reducing their called client-drift by using a similar formulation to the variance reduction technique [10]. They have shown that the communication complexity of SCAFFOLD can be smaller than the one of minibatch SGD for not too heterogenous local datasets under the *quadraticity* of the (possibly nonconvex) local objectives, which is quite limited. For general nonconvex objectives, the communication complexity of SCAFFOLD is same as minibatch SGD. More recently, Murate and Suzuki [19] have proposed Bias-Variance Reduced Local SGD (BVR-L-SGD). BVR-L-SGD utilizes their proposed *bias-variance reduced estimator* that simultaneously reduces the bias caused by local gradient steps and the variance caused by stochastization of the gradients in local optimization based on the formulation of SARAH like variance reduction [20]. They have shown that the communication complexity of BVR-L-SGD is smaller than minibatch SGD for not too heterogeneous local datasets *for general nonconvex objectives*. Specifically, BVR-L-SGD is superior to minibatch SGD when the Hessian heterogeneity of the local datasets is small relative to the smoothness of the local loss in order sense.

On the other side, there are vast work that has studied second-order optimality guarantees, which is much more challenging to ensure but desirable than first-order one, in non-distributed nonconvex optimization. Several approaches are known and one of the simplest approaches is parameter perturbation [4, 8, 9]. However, almost all existing analysis of local methods have only focused on achieving first-order optimality. As an exception, Vlaski et al. [26] have analysed second-order guarantees of local SGD with parameter perturbation, but the obtained communication complexity is much worse than the one of minibatch SGD and no benefit of localization has been shown.

**Open question.** For local methods, it is not well-studied how to find second-order optimal points with low communication cost and thus we have the following research questions:
*Is there a first-order distributed optimization algorithm with second-order optimality guarantees which satisfies that (i) the communication complexity is smaller than non-local methods for not too heterogeneous local datasets; and (ii) the communication complexity approaches to $\Theta(1)$ when the heterogeneity of local datasets goes to zero?*

Note that the both properties are desirable in distributed optimization. We expect that local methods are superior to non-local methods for not highly heterogeneous local datasets. Furthermore, when the local datasets are nearly identical, it is expected that a few communications are sufficient to optimize the global objective. *Only in the case of first-order optimality*, Murata and Suzuki [19] have shown that their proposed BVR-L-SGD satisfies (i) and (ii), and the question has been positively answered. However, the question is still open in the case of second-order optimality[3].

### Main Contributions

We propose a new local algorithm called Bias-Variance Reduced Local *Perturbed* SGD (BVR-L-PSGD) for nonconvex distributed learning to efficiently find second-order optimal points, which positively answered the above research questions.

The algorithm is based on a simple combination of the existing bias and variance reduced gradient estimator and parameter perturbation. In our algorithm, parameter perturbation is carried out *at every*

---

[3]Since there are communication efficient distributed optimization algorithms that find first-order stationary points like BVR-L-SGD, we can apply generic algorithms which guarantee second-order optimality to them [29, 1]. However, this naive approach does not possess the aforementioned property (ii) because the generic framework requires at least $\Omega(1/\varepsilon^{3/2})$ communication rounds to guarantee second-order optimality due to the multiple negative curvature exploitation steps in their framework for any communication efficient algorithms with first-order optimality guarantees. Also, this approach requires explicit negative curvature exploitation, that is complicated and makes the whole algorithm less practical.

| Algorithm | Communication Rounds | Assumptions | Guarantee |
|---|---|---|---|
| Minibatch SGD | $\frac{1}{\varepsilon^2} + \frac{1}{\mathcal{B}P\varepsilon^4}$ | 2-3, BSGV | 1st-order |
| Noisy Minibatch SGD [9] | $\frac{1}{\varepsilon^2} + \frac{1}{\mathcal{B}P\varepsilon^4}$ | 2-4 | **2nd-order** |
| Minibatch SARAH [21] | $\frac{1}{\varepsilon^2} + \frac{\sqrt{n}}{\mathcal{B}P\varepsilon^2} + \frac{n}{\mathcal{B}P}$ | 2-3 | 1st-order |
| SSRGD [17] | $\frac{1}{\varepsilon^2} + \frac{\sqrt{n}}{\varepsilon^{\frac{3}{2}}}$ | 2-5, $\mathcal{B} \geq \frac{\sqrt{n}}{P}$ | **2nd-order** |
| Local SGD [30] | $\frac{1}{\mathcal{B}\varepsilon^2} + \frac{1}{\mathcal{B}P\varepsilon^4} + \frac{1}{\varepsilon^3}$ | 2-3, 5 | 1st-order |
| SCAFFOLD [12] | $\frac{1}{\varepsilon^2} + \frac{1}{\mathcal{B}P\varepsilon^4}$ | 2-3, BSGV | 1st-order |
| SCAFFOLD [12] | $\frac{1}{\mathcal{B}\varepsilon^2} + \frac{1}{\mathcal{B}P\varepsilon^4} + \frac{\zeta}{\varepsilon^2}$ | 1-3, BSGV, quadraticity | 1st-order |
| BVR-L-SGD [19] | $\frac{1}{\sqrt{\mathcal{B}}\varepsilon^2} + \frac{\sqrt{n}}{\mathcal{B}P\varepsilon^2} + \frac{\zeta}{\varepsilon^2}$ | 1-4 | 1st-order |
| **BVR-L-PSGD (this paper)** | $\frac{1}{\sqrt{\mathcal{B}}\varepsilon^2} + \frac{\sqrt{n}}{\mathcal{B}P\varepsilon^2} + \frac{\zeta}{\varepsilon^2}$ | 1-5 | **2nd-order** |

Table 1: Comparison of the order of the necessary number of communication rounds to achieve desired optimization error $\varepsilon$ in terms of given optimization criteria (described in the column of "Guarantee") in nonconvex optimization. "Assumptions" indicates the necessary assumptions to derive the results (the numbers correspond to Assumptions 2, 3, 4, 5 in Section 2 respectively). BSGV means the bounded stochastic gradient variance assumption, that is $\mathbb{E}_{z \sim D_p} \|\nabla \ell(x, z) - \nabla f_p(x)\|^2 \leq \sigma^2$. $\mathcal{B}$ is the local computation budget , which is defined in Section 2. $P$ is the number of workers. $n$ is the total number of samples. The gradient Lipschitzness $L$, Hessian Lipschitzness $\rho$, the gradient boundedness $G$ are regarded as $\Theta(1)$ for ease of presentation. In this notation, *Hessian heterogeneity $\zeta$ always satisfies $\zeta \leq \Theta(L) = \Theta(1)$.*

*local update* and it is not necessary to determine whether or not to add noise by checking the norm of the global gradients, which is often required in several previous non-distributed algorithms [5, 17].

We analyse BVR-L-PSGD for general nonconvex smooth objectives. The most challenging part of our analysis is to ensure that our algorithm efficiently *escapes global saddle points* even *in local optimization*. To realize this, it is necessary to analyse the behavior of the bias-variance reduced estimator around the saddle points by carefully evaluating the degree of some kind of asymptotic consistency of the estimator around the saddle points. This point has never been pursued in previous work and has a unique difficulty of our analysis.

The comparison of the communication complexities of our method with the most relevant existing results is given in Table 1. Our proposed method enjoys second-order optimality with nearly the same communication complexity as the one of BVR-L-SGD, which achieves the best known communication complexity to achieve first-order optimality. This means that our method finds second-order optimal points without hurting the communication efficiency of the state-of-the-art first-order optimality guaranteed method. Particularly, the communication complexity is better than minibatch SGD when Hessian heterogeneity $\zeta$ is small relative to smoothness $L$. Also, the communication complexity approaches to $\widetilde{O}(1)$ when heterogeneity $\zeta$ goes to zero and the local computation budget $\mathcal{B}$ (see Section 2) goes to infinity. Hence, our method enjoys the aforementioned two desired properties.

**Related Work**

Here, we briefly review the related studies to our paper.

**Local methods.** Several recent papers have studied local algorithms combined with variance reduction technique [23, 2, 14, 11]. Sharma et al. [23] have proposed a local variant of SPIDER [3] and shown that the proposed algorithm achieves the optimal total computational complexity. However, the communication complexity essentially matches the ones of non-local SARAH and no advantage of localization has been shown. Khanduri et al. [14] have proposed STEM and its variants based on their called two-sided momentum, but again the communication complexity does not improve non-local methods. Also, Das et al. [2] have considered a SPIDER like local algorithm called FedGLOMO but the derived communication complexity is even worse than minibatch SARAH. Karimireddy et al. [11] have proposed Mime, which is a general framework to mitigate client-drift. Particularly, under $\delta$-Bounded Hessian Dissimilarity (BHD)[4], their MimeMVR achieves communication complexity of $1/(P\varepsilon^2) + \delta/(\sqrt{P}\varepsilon^3) + \delta/\varepsilon^2$ when $\mathcal{B} \to \infty$, that is better than the one of minibatch SGD $1/\varepsilon^2$ when

$\delta \leq \sqrt{P}\varepsilon$. However, the asymptotic rate is still worse than the one of BVR-L-SGD $\zeta/\varepsilon^2$ because $\zeta \leq \delta$ always holds.

**Second-order guarantee.** Neon [29] and Neon2 [1] are generic first-order methods with second-order guarantees, that repeatedly run a first-order guaranteed algorithm and negative curvature descent. Another approach is a parameter perturbation for SGD. For the first time, Ge et al. [4] have shown that SGD with a simple parameter perturbation escapes saddle points efficiently. Later, the analysis has been refined by [8, 9]. Recently, applying variance reduction technique to second-order guaranteed methods has been also studied [5, 17] and particularly Li et al. [17] have proposed SSRGD that combines SARAH [20] with parameter perturbation and shown that SSRGD nearly achieves the optimal computational complexity with second-order optimality guarantees.

## 2  Problem Definition and Assumptions

In this section, we first introduce several notations and definitions used in this paper. Then, the problem settings are described and theoretical assumptions used in our analysis are given.

**Notation.** $\|\cdot\|$ denotes the Euclidean $L_2$ norm $\|\cdot\|_2$: $\|x\| = \sqrt{\sum_i x_i^2}$ for vector $x$. For a matrix $X$, $\|X\|$ denotes the induced norm by the Euclidean $L_2$ norm. For a natural number $m$, $[m]$ means the set $\{1, 2, \ldots, m\}$. For a set $A$, $\#A$ means the number of elements, which is possibly $\infty$. For any number $a, b$, $a \vee b$ and $a \wedge b$ denote $\max\{a, b\}$ and $\min\{a, b\}$ respectively. We denote the uniform distribution over $A$ by $\mathrm{Unif}(A)$. Given $K, T, S \in \mathbb{N}$, let $I(k, t, s)$ be integer $k + Kt + KTs$ for $k \in [K] \cup \{0\}$, $t \in [T-1] \cup \{0\}$ and $s \in [S-1] \cup \{0\}$. Note that $I(K, t, s) = I(0, t+1, s)$ and $I(k, T, s) = I(k, 0, s+1)$ for $k \in [K] \cup \{0\}$, $t \in [T-1] \cup \{0\}$ and $s \in [S-1] \cup \{0\}$. $\boldsymbol{B}_r^d$ denotes the set $\{x \in \mathbb{R}^d | \|x\| \leq r\}$, which is the Euclidean ball in $\mathbb{R}^d$ with radius $r$.

**Definition 2.1** (Gradient Lipschitzness). A differentiable function $f : \mathbb{R}^d \to \mathbb{R}$ is $L$-gradient Lipschitz if $\|\nabla f(x) - \nabla f(y)\| \leq L\|x - y\|, \forall x, y \in \mathbb{R}^d$.

**Definition 2.2** (Hessian Lipschitzness). A twice differentiable function $f : \mathbb{R}^d \to \mathbb{R}$ is $\rho$-Hessian Lipschitz if $\left\|\nabla^2 f(x) - \nabla^2 f(y)\right\| \leq \rho\|x - y\|, \forall x, y \in \mathbb{R}^d$.

**Definition 2.3** (Second-order optimality). For a $\rho$-Hessian Lipschitz function $f$, $x \in \mathbb{R}^d$ is an $\varepsilon$-second-order optimal point of $f$ if $\|\nabla f(x)\| \leq \varepsilon$ and $\nabla^2 f(x) \succeq -\sqrt{\rho\varepsilon}I$.

### 2.1  Problem Settings

**Objective function.** We want to minimize nonconvex smooth objective $f(x) := \frac{1}{P}\sum_{p=1}^{P} f_p(x)$, where $f_p(x) := \mathbb{E}_{z \sim D_p}[\ell(x, z)]$ for $x \in \mathbb{R}^d$, where $D_p$ is the data distribution associated with worker $p$. In this paper, we focus on offline settings (i.e., $\#\mathrm{supp}(D_p) < \infty$ for every $p \in [P]$) for simple presentation. It is easy to extend our results to online settings. Also, just for simplicity, it is assumed that each local dataset has an equal number of samples, i.e., $\#\mathrm{supp}(D_p) = n/P$ for every $p, p' \in [P]$, where $n$ is the total number of samples.

**Optimization criteria.** Since objective function $f$ is nonconvex, it is generally difficult to find a global minima of $f$. Previous work in distributed learning has mainly focused on finding first-order stationary points of $f$. In this study, we aim to find $\varepsilon$-second-order stationary points of $f$ in distributed learning settings.

**Data access constraints and communication settings.** It is assumed that each worker $p$ can only access the own data distribution $D_p$ without communication. Aggregation (e.g., summation) of all the worker's $d$-dimensional parameters or broadcast of a $d$-dimensional parameter from one worker to the other workers can be realized by single communication. [5]

**Evaluation criteria: communication complexity.** In this paper, we compare *communication complexities* of optimization algorithms to satisfy the aforementioned optimization criteria. In typical situations, single communication is more time-consuming than single stochastic gradient computation. Let $\mathcal{C}$ be the single communication cost and $\mathcal{G}$ be the single stochastic gradient computation cost. Using these notations, $\mathcal{C} \geq \mathcal{G}$ is assumed. We expect that increasing the number of available stochastic

---

[4]$\delta$-BHD condition in [11] requires $\|\nabla^2\ell(x, z) - \nabla^2 f(x)\| \leq \delta$ for every $x \in \mathbb{R}^d$, $z \sim D_p$ and $p \in [P]$. Note that $\delta$-BHD condition requires both intra Hessian dissimilarity boundedness $\|\nabla^2 f_i(x) - \nabla^2 f(x)\|$, which is bounded by $\zeta$ under Assumption 1, and additionally inner Hessian dissimilarity $\|\nabla^2\ell(x, z) - \nabla^2 f_i(x)\|$. Hence, $\delta$-BHD is much stronger than Assumption 1 and it is possible that $\delta \gg \zeta$.

gradients in a single communication round leads to faster convergence. Hence, it is natural to increase the number of stochastic gradient computations in a single communication round unless the total stochastic gradient computation time exceeds $\mathcal{C}$ to reduce the total running time. This motivates the concept of **local computation budget** $\mathcal{B}$ ($\leq \mathcal{C}/\mathcal{G}$): given a communication and computational environment, it is assumed that *each worker can only computes at most $\mathcal{B}$ single stochastic gradients per communication round on average*. Then, we compare the communication complexity, that is *the total number of communication rounds of a distributed optimization algorithm to achieve the desired optimization accuracy*. From the definition, we can see that the communication complexity on a fixed local computation budget $\mathcal{B} := \mathcal{C}/\mathcal{G}$ captures the best achievable total running time of an algorithm.

## 2.2 Theoretical Assumptions

In this paper, we assume the following five assumptions. The first one has already been adopted in several previous work [12, 19]. The other ones are standard in the nonconvex optimization literature to guarantees second-order optimality.

**Assumption 1** (Hessian heterogeneity [12, 19]). $\{f_p\}_{p=1}^P$ is second-order $\zeta$-heterogeneous, i.e., for any $p, p' \in [P]$, $\left\| \nabla^2 f_p(x) - \nabla^2 f_{p'}(x) \right\| \leq \zeta, \forall x \in \mathbb{R}^d$.

Assumption 1 characterizes the heterogeneity of local objectives $\{f_p\}_{p=1}^P$ in terms of Hessians and has a important role in our analysis. Intuitively, we expect that relatively small heterogeneity parameter $\zeta$ to the smoothness parameter $L$ (defined in Assumption 2) reduces the necessary number of communication rounds to optimize the global objective. Especially when the local objectives are identical, i.e., $D_p = D_{p'}$ for every $p, p' \in [P]$, $\zeta$ becomes zero. When each $D_p$ is the empirical distribution of $n/P$ IID samples from common data distribution $D$, we have $\|\nabla^2 f_p(x) - \nabla^2 f_{p'}(x)\| \leq \widetilde{\Theta}(\sqrt{P/n}L)$ with high probability by matrix Hoeffding's inequality under Assumption 2 for fixed $x$. Hence, in traditional distributed learning regimes, Assumption 1 naturally holds. An important remark is that *Assumption 2 implies $\zeta \leq 2L$, i.e., the heterogeneity is bounded by the smoothness.* Even in federated learning regimes, we expect $\zeta \ll 2L$ for some problems practically.

**Assumption 2** (Gradient Lipschitzness). $\forall p \in [P], z \in \text{supp}(D_p)$, $\ell(\cdot, z)$ is $L$-gradient Lipschitz.

**Assumption 3** (Existence of global optimum). $f$ has a global minimizer $x_* \in \mathbb{R}^d$.

**Assumption 4** (Hessian Lipschitzness). $\forall p \in [P], z \in \text{supp}(D_p)$, $\ell(\cdot, z)$ is $\rho$-Hessian Lipschitz.

**Assumption 5** (Bounded stochastic gradient). $\forall p \in [P], z \in \text{supp}(D_p)$, $\nabla \ell(\cdot, z)$ is $G$-bounded, i.e., $\|\nabla \ell(x, z)\| \leq G, \forall x \in \mathbb{R}^d$.

In our analysis, $G$ has no significant impact because $G$ only depends on our theoretical communication complexity in logarithmic order.

# 3 Main Ideas and Proposed Algorithm

Our proposed algorithm is based on a natural combination of (i) *Bias-Variance Reduced (BVR) estimator*; and (ii) *parameter perturbation at each local update*. The first idea has been proposed by [19] to find first-order stationary points with small communication complexity. The second one is a well-known approach to find second-order stationary points in non-distributed nonconvex optimization [4, 8, 9]. In this section, we illustrate these two ideas and provide its concrete procedures.

## 3.1 Review of BVR Estimator [19]

The bias-variance reduced estimator aims to efficiently find first-order stationary points by simultaneously reducing the bias caused by local gradient descent steps and the variance caused by stochastization of the used gradients.

First we consider why the standard local SGD is not sufficient to achieve fast convergence and sometimes slower than minibatch SGD. Recall that in local SGD each worker takes the update rules

---

[5]In this work, it is assumed that all the workers can participate in a single communication. It is not so hard to extend our algorithm and analysis to worker sampling settings, which is more realistic in cross-device federated learning.

---

**Algorithm 1** BVR-L-PSGD($\widetilde{x}_0, \eta, b, K, T, S, r$)

1: Add noise $x_0 = \widetilde{x}_0 + \eta\xi_{-1}$, where $\xi_{-1} \sim \mathrm{Unif}(\boldsymbol{B}_r^d)$.
2: **for** $s = 0$ to $S - 1$ **do**
3:    **for** $p = 1$ to $P$ in parallel **do**
4:       $v_{I(0,0,s)}^{(p)} = \nabla f_p(x_{I(0,0,s)})$.
5:    **end for**
6:    Communicate $\{v_{I(0,0,s)}^{(p)}\}_{p=1}^P$. Set $v_{I(0,0,s)} = \frac{1}{P}\sum_{p=1}^P v_{I(0,0,s)}^{(p)}$.
7:    **for** $t = 0$ to $T - 1$ **do**
8:       **for** $p = 1$ to $P$ in parallel **do**
9:          $g_{I(0,t,s)}^{(p)} = \frac{1}{Kb}\sum_{l=1}^{Kb}\nabla\ell(x_{I(0,t,s)}, z_{l,I(0,t,s)})$,
10:         $g_{I(0,t,s)}^{(p),\mathrm{ref}} = \frac{1}{Kb}\sum_{l=1}^{Kb}\nabla\ell(x_{I(0,t-1,s)}, z_{l,I(0,t,s)})$ ($z_{l,I(0,t,s)} \overset{i.i.d.}{\sim} D_p$).
11:         $v_{I(0,t,s)}^{(p)} = \mathbb{1}_{t\geq 1}(g_{I(0,t,s)}^{(p)} - g_{I(0,t,s)}^{(p),\mathrm{ref}} + v_{I(0,t-1,s)}^{(p)}) + \mathbb{1}_{t=0}v_{I(0,0,s)}^{(p)}$.
12:       **end for**
13:       Communicate $\{v_{I(0,t,s)}^{(p)}\}_{p=1}^P$. Set $v_{I(0,t,s)} = \frac{1}{P}\sum_{p=1}^P v_{I(0,t,s)}^{(p)}$.
14:       Randomly select $p_{t,s} \sim \mathrm{Unif}[P]$. # Only worker $p_{t,s}$ runs local optimization.
15:       **for** $k = 0$ to $K - 1$ **do**
16:          $b_k = \mathbb{1}_{k\equiv 0 \ (\mathrm{mod}\lceil\sqrt{K}\rceil)}\lceil\sqrt{K}\rceil b + \mathbb{1}_{k\not\equiv 0 \ (\mathrm{mod}\lceil\sqrt{K}\rceil)}b$.
17:          $g_{I(k,t,s)} = \frac{1}{b_k}\sum_{l=1}^{b_k}\nabla\ell(x_{I(k,t,s)}, z_{l,I(k,t,s)})$,
18:          $g_{I(k,t,s)}^{\mathrm{ref}} = \frac{1}{b_k}\sum_{l=1}^{b_k}\nabla\ell(x_{I(k-1,t,s)}), z_{l,I(k,t,s)})$ ($z_{l,I(k,t,s)} \overset{i.i.d.}{\sim} D_{p_{t,s}}$).
19:          $v_{I(k,t,s)} = \mathbb{1}_{k\geq 1}(g_{I(k,t,s)} - g_{I(k,t,s)}^{\mathrm{ref}} + v_{I(k-1,t,s)}) + \mathbb{1}_{k=0}v_{I(0,t,s)}$.
20:          Update $\widetilde{x}_{I(k+1,t,s)} = x_{I(k,t,s)} - \eta v_{I(k,t,s)}$.
21:          Add noise $x_{I(k+1,t,s)} = \widetilde{x}_{I(k+1,t,s)} + \eta\xi_{I(k,t,s)}$, where $\xi_{I(k,t,s)} \sim \mathrm{Unif}(\boldsymbol{B}_r^d)$.
22:       **end for**
23:       Communicate $x_{I(0,t+1,s)}$.
24:    **end for**
25: **end for**

---

of $x_{k+1}^{(p)} = x_k^{(p)} - \eta g_k^{(p)}$ for $k \in [\mathcal{B}/b]$ in each communication round, where $g_k^{(p)}$ is a stochastic gradient with minibatch size $b$ at $x_k^{(p)}$ on local dataset $D_p$ and $\mathcal{B}$ is given local computation budget. In typical convergence analysis, we need to bound the expected deviation of $g_k^{(p)}$ from ideal global gradient $\nabla f(x_k)$, that is $\mathbb{E}\|g_k^{(p)} - \nabla f(x_k^{(p)})\|^2 = \|\nabla f_p(x_k^{(p)}) - \nabla f(x_k^{(p)})\|^2 + \mathbb{E}\|g_k^{(p)} - \nabla f_p(x_k^{(p)})\|^2$. The former term is called *bias* and the latter one is called *variance*. A typical assumption to bound the first term is *bounded gradient heterogeneity assumption*, that requires $\|\nabla f_p(x) - \nabla f(x)\| \leq \zeta_1$ for every $x \in \mathbb{R}^d$ and $p \in [P]$. Under this assumption, the first term is only bounded by $\zeta_1$, that is a constant. The second term is typically bounded by $\sigma^2/b$ for $g_k^{(p)}$ with minibatch size $b$, when the variance of a single stochastic gradient is bounded by $\sigma^2$. These facts show that the bias is still a constant and does not vanish even if minibatch size $b$ is enhanced and the variance vanishes. This is why local SGD can be worse than minibatch SGD when $\zeta_1$ is not too small. Also, we can see that the variance is still a constant for fixed minibatch size $b$ and this is a common reason why minibatch SGD and local SGD only show slow convergences. These observations give critical motivations of the simultaneous reduction of the bias and variance.

The bias-variance reduced estimator $v_k^{(p)}$ is defined as $v_k^{(p)} := (1/b)\sum_{l=1}^b(\nabla\ell(x_k^{(p)}, z_l) - \nabla\ell(x_0, z_l)) + \nabla f(x_0)$ (SVRG version). It is known that the bias caused by localization can be bounded by $\zeta\|x_k^{(p)} - x_0\|$ and the variance caused by stochastization can be bounded by $(L^2/b)\|x_k^{(p)} - x_0\|^2$, where $\zeta$ is the Hessian heterogeneity of $\{f_p\}$ and $L$ is the smoothness of $\ell$. This implies that both the bias and variance of $v_k^{(p)}$ converges to zero as $x_k^{(p)}$ and $x_0$ go to $x_*$. In other words, *bias-variance reduced estimator $v_k^{(p)}$ is asymptotically consistent to the global gradient* $\nabla f(x_k^{(p)})$ by using periodically computed global full gradients $\nabla f(x_0)$. We actually adopt SARAH version of BVR estimator as in [19] rather than SVRG one due to its theoretical advantages.

## 3.2 Parameter Perturbation at Local Updates

Although the bias-variance reduced estimator is useful to guarantee first-order optimality with small communication complexity in noncovex optimization, the algorithm often gets stuck at saddle points. To tackle this problem, we borrow the ideas of escaping saddle points in non-distributed nonconvex optimization. Particularly, to efficiently find second-order optimal points, we utilize *parameter perturbation*. Parameter perturbation is a familiar approach in non-distributed nonconvex optimization. Specifically, Jin et al. [8, 9] have considered the update rule of $x_{k+1} = x_k - \eta \nabla f(x_k) + \eta \xi_k$, where $\xi_k \sim \text{Unif}(\mathcal{B}_r^d)$ for some small radius $r$. This algorithm is called Perturbed GD (PGD) or Noisy GD. Similar to this formulation, we add noise at each local update, i.e., $x_{k+1}^{(p)} = \widetilde{x}_{k+1}^{(p)} + \eta \xi_k^{(p)}$, where $\widetilde{x}_{k+1}^{(p)} = x_k^{(p)} - \eta v_k^{(p)}$. The intuition behind the noise addition is that random noise has some components along the negative curvature directions of the global objective around the saddle point, and we expect that noise addition helps the parameter proceed to the decreasing directions of $f$ and escape the saddle points.

**Necessity of local perturbation.** Perturbing the global model at the server side is an intuitive way, but not sufficient for communication efficiency when we want to utilize small heterogeneity of the local datasets (i.e., $\zeta \ll L$). The bias-variance reduced estimator with local perturbation enables to escape *multiple* global saddle points *in local optimization* and achieves second-order optimality with communication complexity $\widetilde{\Theta}(\zeta/\varepsilon^2)$ for sufficiently large $\mathcal{B}$. In contrast, perturbing the global parameter at the server side only ensures to escape *single* global saddle point at each round and only achieves communication complexity of $\widetilde{\Theta}(L/\varepsilon^2)$. This is the reason why local perturbation rather than global one is adopted.

### 3.3 Concrete Procedures

The full description of our proposed Bias-Variance Reduced Local Perturbed SGD (BVR-L-PSGD) is given in Algorithm 1. When we set the noise size $r = 0$, Algorithm 1 essentially matches BVR-L-SGD. Additionally setting $K = 1$, Algorithm 1 matches SARAH. The algorithm requires $\Theta(ST)$ communication rounds. At each communication round, each worker computes large batch stochastic gradients and the server constructs $v_{I(0,t,s)}$ by aggregating them. $v_{I(0,t,s)}$ is used as an estimator of $\nabla f(x_{I(0,t,s)})$ to reduce computational cost. In line 14-21, we randomly select worker $p_{t,s}$ and only worker $p_{t,s}$ runs local optimization as described above. In the local optimization, we use SARAH like bias variance reduced estimator (line 16-18) rather than SVRG one and add noise (line 20) at each local update.

## 4 Convergence Analysis

In this section, we provide convergence theory of BVR-L-PSGD (Algorithm 1). All the omitted proofs are found in the supplementary material. For simple presentations, we use $\widetilde{\Theta}$ symbol to hide an extra poly-logarithmic factors that depend on $L, \rho, G, K, b, T, S, 1/\varepsilon, 1/q$, where $q$ represents the confidence parameter in high probability bounds.

### 4.1 Finding First-Order Stationary Points

First, we derive Descent Lemma for BVR-L-PSGD and first-order optimality guarantees by using it.

**Proposition 4.1** (Descent Lemma). *Let $S \in \mathbb{N}$ and $I(k, t, s) \geq I(k_0, t_0, s_0) \in [KTS] \cup \{0\}$. Suppose that Assumptions 1, 2, 3 and 5 hold. Given $q \in (0, 1)$, $r > 0$, if we appropriately choose $\eta = \widetilde{\Theta}(1/L \wedge 1/(K\zeta) \wedge \sqrt{b/K}/L \wedge \sqrt{Pb}/(\sqrt{KT}L))$, it holds that*

$$f(x_{I(k,t,s)}) \leq f(x_{I(k_0,t_0,s_0)}) - \frac{\eta}{2} \sum_{i=I(k_0,t_0,s_0)}^{I(k-1,t,s)} \|\nabla f(x_i)\|^2 + \eta \Delta_I r^2 + R_1$$

*with probability at least $1 - 3q$. Here, $\Delta_I := I(k, t, s) - I(k_0, t_0, s_0)$, $R_1 := -\frac{1}{4\eta} \sum_{i=I(k_0,t_0,s_0)}^{I(k,t,s)-1} \|x_{i+1} - x_i\|^2 + \frac{c_\eta}{\eta} \left( \frac{\Delta_I \wedge K}{K} \sum_{i=I(0,t_0,s_0)}^{I(k_0,t_0,s_0)-1} \|x_{i+1} - x_i\|^2 + \frac{\Delta_I \wedge KT}{KT} \sum_{i=I(0,0,s_0)}^{I(0,t_0,s_0)-1} \|x_{i+1} - x_i\|^2 \right)$ for some universal constant $c_\eta > 0$.*

From Proposition 4.1 with $I(k_0, t_0, s_0) \leftarrow 0$ and $I(k, t, s) \leftarrow KTS$ gives the following corollary.

**Corollary 4.2.** *Suppose that Assumptions 1, 2, 3 and 5 hold. Under the same setting as in Proposition 4.3 and $S \geq \Theta((f(x_0) - f(x_*))/(\eta KT\varepsilon^2))$, with probability at least $1 - 3q$, there exists $i \in [KTS - 1] \cup \{0\}$ such that $\|\nabla f(\widetilde{x}_i)\| \leq \varepsilon$.*

*Remark* (Communication complexity). The total number of communication rounds $\Theta(TS)$ becomes $\widetilde{O}\left(T + \left(L/K + \zeta + L/\sqrt{Kb} + \sqrt{T}L/\sqrt{KPb}\right)(f(\widetilde{x}_0) - f(x_*))/\varepsilon^2\right)$. Given local computation budget $\mathcal{B}$, we set $T := \Theta(1 + n/(\mathcal{B}P))$ and $Kb := \Theta(\mathcal{B})$ with $b \leq \Theta(\sqrt{\mathcal{B}})$. Then, we have the averaged number of local computations per communication round $Kb + n/(PT) = \Theta(\mathcal{B})$ and the communication complexity $TS$ with budget $\mathcal{B}$ becomes

$$\widetilde{O}\left(1 + \frac{n}{\mathcal{B}P} + \frac{L}{\sqrt{\mathcal{B}}\varepsilon^2} + \frac{\sqrt{n}L}{\mathcal{B}P\varepsilon^2} + \frac{\zeta}{\varepsilon^2}\right),$$

which matches the best known communication complexity [19].

## 4.2 Escaping Saddle Points

Next, we show that BVR-L-PSGD implicitly exploits the negative curvature of $f$ around saddle points and efficiently escapes the saddle points by utilizing the asymptotic consistency of BVR estimator and the parameter perturbation at each local update.

We rely on the technique of coupling sequence [9]. Given saddle point $\widetilde{x}_{I(k_0,t_0,s_0)}$ and $\hat{I} \geq I(k_0, t_0, s_0)$, we define a new sequence $\{x_i'\}_{i=I(k_0,t_0,s_0)}^{\infty}$ as follows:

(1) $\langle \xi_{\widetilde{I}}', \boldsymbol{e}_{\min} \rangle = -\langle \xi_{\widetilde{I}}, \boldsymbol{e}_{\min} \rangle$; (2) $\langle \xi_{\widetilde{I}}', \boldsymbol{e}_j \rangle = \langle \xi_{\widetilde{I}}, \boldsymbol{e}_j \rangle$ for $j \in \{2, \ldots, d\}$; and (3) All the other randomness is completely same as the one of $\{x_i\}_{i=0}^{KTS-1}$. Let $r_0 := |\langle \xi_{\widetilde{I}}, \boldsymbol{e}_{\min}\rangle|$. Note that $|\langle \xi_{\widetilde{I}} - \xi_{\widetilde{I}}', \boldsymbol{e}_{\min}\rangle| = 2r_0$ and thus $\|\xi_{\widetilde{I}} - \xi_{\widetilde{I}}'\| = 2r_0$. Also, observe that $x_{\widetilde{I}+1} - x_{\widetilde{I}+1}' = \eta\langle \xi_{\widetilde{I}} - \xi_{\widetilde{I}}', \boldsymbol{e}_{\min}\rangle \boldsymbol{e}_{\min}$. We define $\widetilde{I}$ used in the definition of coupling sequence as follows:

$$\widetilde{I} := \begin{cases} I(k_0, t_0, s_0), & (1/(\eta\lambda) \leq \sqrt{K}) \\ I(k_0', t_0, s_0) - 1, & (\sqrt{K} < 1/(\eta\lambda) \leq K) \\ I(0, t_0 + 1, s_0) - 1, & (K < 1/(\eta\lambda) \leq KT) \\ I(0, 0, s_0 + 1) - 1. & (KT < 1/(\eta\lambda)) \end{cases}$$

Here, $k_0'$ is the minimum index $k$ that satisfies $k > k_0$ and $k \equiv 0 \ (\mathrm{mod}\lceil\sqrt{K}\rceil)$. We can easily check that $\widetilde{I} - I(k_0, t_0, s_0) \leq 1/(\eta\lambda)$.

Then, we show that either of the two sequences $\{x_i\}$ or $\{x_i'\}$ efficiently escapes the saddle points by bounding the norm of the cumulative difference of $x_i$ and $x_i'$ from below. The novel and most difficult part of the analysis is to evaluate the norm of the cumulative difference of the deviations $\|\sum_{i=\widetilde{I}}^{\mathcal{J}}(1 - \eta\mathcal{H})^{\mathcal{J}-i}(v_i - \nabla f(x_i) - v_i' + \nabla f(x_i'))\|$ generated by the two sequences, where $v_i'$ denotes the BVR estimator at iteration $i$ generated by sequence $\{x_i'\}$.

**Proposition 4.3** (Implicit Negative Curvature Exploitation). *Let $I(k_0, t_0, s_0) \in [KTS] \cup \{0\}$. Suppose that Assumptions 1, 2, 3, 4 and 5 hold, $\|\nabla f(\widetilde{x}_{I(k_0,t_0,s_0)})\| \leq \varepsilon$ and the minimum eigenvalue $\lambda_{\min}$ of $\mathcal{H} := \nabla f(\widetilde{x}_{I(k_0,t_0,s_0)})$ satisfies $\lambda := -\lambda_{\min} > \sqrt{\rho\varepsilon}$. Under $b = \Omega(K \vee 1/(\sqrt{K}\rho\varepsilon) \vee T/(PK))$, if we appropriately choose $\mathcal{J}_{I(k_0,t_0,s_0)} = \widetilde{\Theta}(1/(\eta\lambda))$, $\eta = \widetilde{\Theta}(1/L \wedge 1/(K\zeta) \wedge \sqrt{b/K}/L \wedge \sqrt{Pb}/(\sqrt{KT}/L))$, with $\mathcal{F}_{I(k_0,t_0,s_0)} := c_{\mathcal{F}}\eta\mathcal{J}_{I(k_0,t_0,s_0)}r^2$ and $r := c_r\varepsilon$ ($c_{\mathcal{F}} = \Theta(1)$ and $c_r = \widetilde{\Theta}(1)$) we have*

$$f(x_{I(k_0,t_0,s_0)+\mathcal{J}_{I(k_0,t_0,s_p)}}) - f(x_{I(k_0,t_0 s_0)}) \leq -\mathcal{F}_{I(k_0,t_0,s_0)} + R_2$$

*with probability at least $1/2 - 9q/2$. Here, $R_2 := \frac{2c_\eta}{\eta}\frac{\mathcal{J}_{I(k_0,t_0,s_0)} \wedge K}{K}\sum_{i=I(0,t_0,s_0)}^{I(k_0,t_0,s_0)-1}\|x_{i+1} - x_i\|^2 + \frac{2c_\eta}{\eta}\frac{\mathcal{J}_{I(k_0,t_0,s_0)} \wedge KT}{KT}\sum_{i=I(0,0,s_0)}^{I(0,t_0,s_0)-1}\|x_{i+1} - x_i\|^2$ for some universal constant $c_\eta > 0$.*

Proposition 4.3 says the function value decreases by roughly $\mathcal{F}_{I(k_0,t_0,s_0)}$ and the global model escapes saddle points with probability at least $1/2$ after $\mathcal{J}_{I(k_0,t_0,s_0)}$ local steps.

### 4.3 Finding Second-Order Stationary Points

In this subsection, we derive final theorem that guarantees the second-order optimality of the global model by combining Propositions 4.1 and 4.3.

**Theorem 4.4** (Final Theorem). *Suppose that Assumptions 1, 2, 3, 4 and 5 hold. Under* $b = \Omega(K \vee 1/(\sqrt{K}\rho\varepsilon) \vee T/(PK))$, *if we appropriately choose* $\eta = \widetilde{\Theta}(1/L \wedge 1/(K\zeta) \wedge \sqrt{b/K}/L \wedge \sqrt{Pb}/(\sqrt{KTL}))$, $r = \widetilde{\Theta}(\varepsilon)$ *and* $S = \Theta(1 + (f(\widetilde{x}_0) - f(x_*))/(\eta KT\varepsilon^2))$, *with probability at least* $1/2$, *there exists* $i \in [KTS] \cup \{0\}$ *such that* $\widetilde{x}_i$ *is* $\varepsilon$-*second-order optimal point of* $f^6$.

*Remark* (High probability bound). Theorem 4.4 guarantees that Algorithm 1 finds an approximate second-order optimal point in $KTS$ iterations with probability at least $1/2$. Repeating Algorithm 1 $\log_2(1/q)$ times guarantees that the same statement holds with probability at least $1 - q$.

*Remark* (Communication complexity). The total number of communication rounds $TS$ is given by $\widetilde{O}\left(T + \left(L/K + \zeta + L/\sqrt{Kb} + \sqrt{T}L/\sqrt{KPb}\right)(f(\widetilde{x}_0) - f(x_*))/\varepsilon^2\right)$. Given local computation budget $\mathcal{B}$, we set $T := \Theta(1 + n/(\mathcal{B}P))$ and $Kb := \Theta(\mathcal{B})$ with $b \leq \Theta(\sqrt{\mathcal{B}})$. Then, we have the averaged number of local computations per communication round $Kb + n/(PT) = \Theta(\mathcal{B})$ and the communication complexity $\Theta(TS)$ with budget $\mathcal{B}$ becomes

$$\widetilde{O}\left(1 + \frac{n}{\mathcal{B}P} + \frac{L}{\sqrt{\mathcal{B}}\varepsilon^2} + \frac{\sqrt{n}L}{\mathcal{B}P\varepsilon^2} + \frac{\zeta}{\varepsilon^2}\right).$$

This implies that for $\zeta = o(L)$ the communication complexity is strictly smaller than the one of minibatch SGD $\widetilde{O}(1 + L/\varepsilon^2 + G^2/(\mathcal{B}P\varepsilon^4))$. Note that the rate matches to the one of BVR-L-SGD [19]. Hence, our method finds second-order optimal points without hurting communication efficiency of the state-of-the-art first-order optimality guaranteed method. Furthermore, when $\mathcal{B} \to \infty$, we have $\widetilde{\Theta}(1 + \zeta/\varepsilon^2)$, that goes to $\widetilde{\Theta}(1)$ as $\zeta \to 0$.

In summary, BVR-L-PSGD enjoys the desirable properties (i) and (ii) described in Section 1.

## 5 Numerical Resutls

In this section, we give some experimental results to verify our theoretical findings.

**Data Preparation.** We artificially generated heterogeneous local datasets from CIFAR10[7] dataset. The data preparation procedure is completely in accordance with [19] and the details are found in [19]. We set homogeneity parameter $q$ to $0.35$, which captures how similar the local datasets are ($q = 0.1$ corresponds to I.I.D. case and higher $q$ does to higher heterogeneity).

**Model.** We conducted our experiments using a two-hidden layers fully connected neural network with 100 hidden units and softplus activation. For loss function, we used the standard cross-entropy loss. We initialized parameters by uniformly sampling the parameters from $[-1/100, 1/100]$.

**Implemented Algorithms.** Minibatch SGD, Noisy Minibatch SGD, BVR-L-SGD [19] and our proposed BVR-L-PSGD were implemented. We set $K = 64$ and $b = 16$, and thus $\mathcal{B} = 1024$. For BVR-L-PSGD, the noise radius was tuned from $r \in \{0.5, 2.5, 12.5\}$. For each algorithm, we tuned learning rate $\eta$ from $\{0.005, 0.01, 0.05, 0.1, 0.5, 1.0\}$. The details of the tuning procedure are found in the supplementary material.

**Evaluation.** We compared the implemented algorithms using six criteria of train gradient norm $\|\nabla f(x)\|$; train loss; train accuracy; test gradient norm; test loss and test accuracy against the number of communication rounds. The total number of communication rounds was fixed to $1,000$ for each algorithm. We independently repeated the experiments 5 times and report the mean and standard deviation of the above criteria. Due to the space limitation, we will only report train gradient norm, train loss and test accuracy in the main paper. The full results are found in the supplementary material.

**Results.** Figure 1 shows the performances of BVR-L-SGD and our proposed algorithm. We can

---

[6]One limitation of Theorem 4.4 is that it only guarantees the existence of $\varepsilon$-second-order optimal point $\widetilde{x}_i$ in the history of $\{\widetilde{x}_i\}_{i=0}^{KTS-1}$. However, this is also the case in the existing studies [5, 17]. We empirically found that the outputs of each communication rounds showed stable performances (see Section 5).

[7]`https://www.cs.toronto.edu/~kriz/cifar.html`.

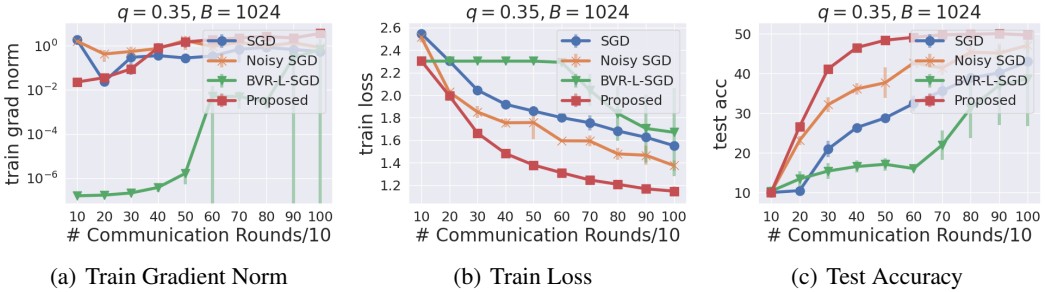

Figure 1: Comparison of (a) train gradient norm; (b) train loss; and (iii) test accuracy against the number of communication rounds for a three layered DNN on heterogeneous CIFAR10.

see that the both algorithms got stuck at a small gradient norm region in initial rounds. After that BVR-L-SGD showed unstable convergence and took a lot of time to escape the stucked region. In contrast, our proposed method efficiently escaped the stucked region and consistently achieves better train loss and test accuracy than BVR-L-SGD. Also, our method consistently outperformed Minibatch SGD and Noisy Minibatch SGD.

# 6   Conclusion

In this paper, we have studied a new local algorithm called Bias-Variance Reduced Local Perturbed SGD (BVR-L-PSGD) based on a combination of the bias-variance reduced gradient estimator with parameter perturbation to efficiently find second-order optimal points in centralized nonconvex distributed optimization. We have shown that BVR-L-PSGD enjoys second-order optimality without hurting the best known communication complexity for first-order optimality guarantees. Particularly, the communication complexity is better than non-local methods when Hessian heterogeneity $\zeta$ of local datasets is smaller than the smoothness of the local loss $L$ in order sense. Also, for sufficiently large $\mathcal{B}$, the communication complexity of our method approaches to $\widetilde{\Theta}(1)$ when the local datasets heterogeneity $\zeta$ goes to zero. The numerical results have validated our theoretical findings.

## Acknowledgement

TS was partially supported by JSPS KAKENHI (20H00576) and JST CREST. The authors would like to thank Kazusato Oko for his helpful advice.

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
