# Supplementary Material:

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

(1 + \tfrac{n}{\mathcal{B}P} + \tfrac{L}{\sqrt{\mathcal{B}}\varepsilon^2} + \tfrac{\sqrt{n}L}{\mathcal{B}P\varepsilon^2} + \tfrac{\zeta}{\varepsilon^2}\right),$$

which matches the best known communication complexity [20].

## 4.2 Escaping Saddle Points

Next, we show that BVR-L-PSGD implicitly exploits the negative curvature of $f$ around saddle points and efficiently escapes the saddle points by utilizing the asymptotic consistency of BVR estimator and the parameter perturbation at each local update.

We rely on the technique of coupling sequence [10]. Given saddle point $\widetilde{x}_{I(k_0, t_0, s_0)}$ and $\hat{I} \geq I(k_0, t_0, s_0)$, we define a new sequence $\{x'_i\}_{i=I(k_0, t_0, s_0)}^{\infty}$ as follows:

(1) $\langle \xi'_{\widetilde{I}}, e_{\min}\rangle = -\langle \xi_{\widetilde{I}}, e_{\min}\rangle$; (2) $\langle \xi'_{\widetilde{I}}, e_j\rangle = \langle \xi_{\widetilde{I}},

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

# A  Supplementary Material for Numerical Results

In this section, we give additional information and numerical results that complement the contents in Section 5.

### Parameter Tuning

For the implemented algorithms, learning rate $\eta$ was tuned. Also, for Noisy Minibatch SGD and BVR-L-PSGD, noise radius $r$ was also tuned. We ran each algorithm for all the patterns of the tuning parameters and chose the ones that maximized the minimum train accuracy.

### Additional Numerical Results

Here, we provide the full results of our numerical experiments. Figures 2 and 3 show the comparisons of the six criterion, i.e., train gradient norm, train loss, train accuracy, test gradient norm, test loss and test accuracy with fixed local computation budget $\mathcal{B} = 1,024$ under $q = 0.1$ (I.I.D. case) and $q = 0.35$ (heterogeneous case) respectively.

### Computing Infrastructures

- OS: Ubuntu 16.04.6
- CPU: Intel(R) Xeon(R) CPU E5-2680 v4 @ 2.40GHz
- CPU Memory: 128 GB.
- GPU: NVIDIA Tesla P100.
- GPU Memory: 16 GB
- Programming language: Python 3.7.3.
- Deep learning framework: Pytorch 1.3.1.

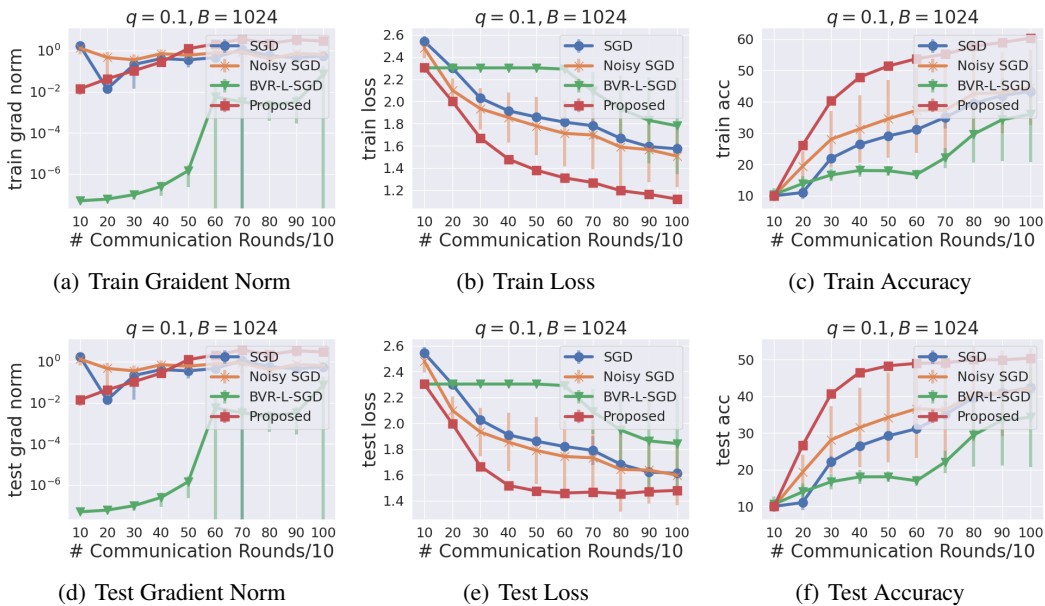

Figure 2: Comparison of the six criterion against the number of communication rounds for a three layered DNN on I.I.D. CIFAR10 with $q = 0.1$.

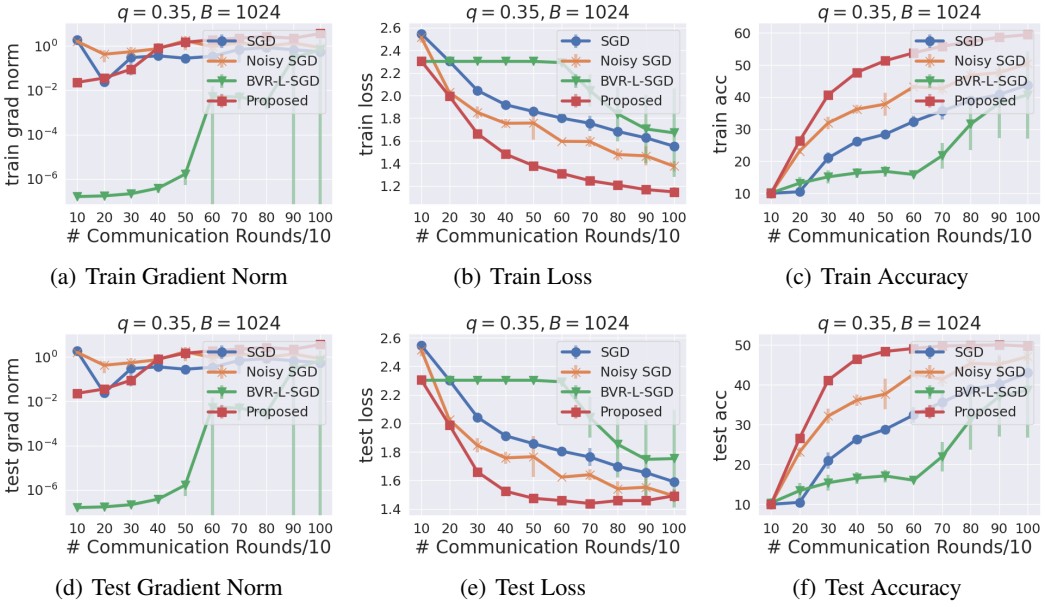

Figure 3: Comparison of the six criterion against the number of communication rounds for a three layered DNN on heterogeneous CIFAR10 with $q = 0.35$.

# B  Convergence Analysis

In this section, complete analysis of BVR-L-PSGD is provided. Particularly, detailed proofs of Proposition 4.1, Corollary 4.2 (Subsection B.3), Proposition 4.3 (Subsection B.4 and Theorem 4.4 (Subsection B.5) are given.

## B.1  Miscellaneous Results

**Lemma B.1.** *Let $A \in \mathbb{R}^{d \times d}$ with the smallest and largest eigenvalues $\lambda_{\min} \in (-\infty, 0)$ and $\lambda_{\max} \in [0, 1)$ respectively. Then, for $J \in \mathbb{N} \cup \{0\}$, it holds that*

$$\|A(1 - A)^J\| \le (-\lambda_{\min})(1 - \lambda_{\min})^J + \frac{e}{J + 1}.$$

*Proof.* First, when $J = 0$, trivially $\|A(1 - A)^J\| \le \max\{-\lambda_{\min}, \lambda_{\max}\} < (-\lambda_{\min}) + e$. Thus, we assume $J > 0$. Note that $\|A(1 - A)^J\| = \sup_{\sigma \in [\lambda_{\min}, \lambda_{\max}]} |\sigma(1 - \sigma)^J|$. We consider the two cases $\sigma \in [\lambda_{\min}, 0)$ and $\sigma \in [0, \lambda_{\max}]$.

In the former case, $h(\sigma) := |\sigma(1 - \sigma)^J| = -\sigma(1 - \sigma)^J$ is monotonically decreasing function on $(-\infty, 0)$ because the derivative function $h'(\sigma) = -(1 - \sigma)^J + J\sigma(1 - \sigma)^{J-1} = (1 - \sigma)^{J-1}((J + 1)\sigma - 1) < 0$, and hence $\sup_{\sigma \in [\lambda_{\min}, 0)} h(\sigma) \le (-\lambda_{\min})(1 - \lambda_{\min})^J$.

In the latter case, $h(\sigma) = \sigma(1 - \sigma)^J$ has the derivative function $h'(\sigma) = (1 - \sigma)^J - J\sigma(1 - \sigma)^{J-1} = (1 - \sigma)^{J-1}(1 - (J + 1)\sigma)$. Thus, it holds that $h'(1/(J + 1)) = 0$, $h'(\sigma) > 0$ for $\sigma \in [0, 1/(J + 1))$ and $h'(\sigma) < 0$ for $\sigma \in (1/(J + 1), 1)$. Hence, for $\sigma \in [0, \lambda_{\max}]$ with $\lambda_{\max} \in [0, 1)$, $h(\sigma) \le h(1/(J + 1)) \le e/(J + 1)$.

In summary, we have shown that $\sup_{\sigma \in [\lambda_{\min}, \lambda_{\max}]} h(\sigma) \le (-\lambda_{\min})(1 - \lambda_{\min})^J + e/(J + 1)$. This is the desired result. □

## B.2  Concentration Inequalities

**Lemma B.2** (Corollary 8 in [9]). *Let $X_1, \ldots, X_n$ be random vectors in $\mathbb{R}^d$. Suppose that $\{X_i\}_{i=1}^n$ and corresponding filtrations $\{\mathfrak{F}_i\}_{i=1}^n$ satisfies the following conditions:*

$$\mathbb{E}[X_i \mid \mathfrak{F}_{i-1}] = 0 \text{ and } \mathbb{P}(\|X_i\| \ge s \mid \mathfrak{F}_{i-1}) \le 2e^{-\frac{s^2}{2\sigma_i^2}}, \forall s \in \mathbb{R}, \forall i \in [n]$$

*for random variables $\{\sigma_i\}_{i=1}^n$ with $\sigma_i \in \mathfrak{F}_{i-1}$ ($i \in [n]$). Then, for any $q \in (0, 1)$ and $A > a > 0$, with probability at least $1 - q$ it holds that*

$$\sum_{i=1}^n \sigma_i^2 \ge A \text{ or } \left\| \sum_{i=1}^n X_i \right\| \le c \sqrt{ \max \left\{ \sum_{i=1}^n \sigma_i^2, a \right\} \left( \log \frac{2d}{q} + \log\log \frac{A}{a} \right) }$$

*for some constant $c > 0$.*

Note that if $X$ is bounded and centered random vector, i.e., $\|X\| \le \sigma$ a.s. and $\mathbb{E}[X] = 0$, it holds that $\mathbb{P}(\|X\| \ge s) \le 2e^{-s^2/2\sigma^2}$ for every $s \in \mathbb{R}$. Hence, $\|X_i\| \le \sigma_i^2$ a.s. and $\mathbb{E}[X_i] = 0$ conditioned on $\mathfrak{F}_{i-1}$ is a sufficient condition for applying Lemma B.2.

## B.3  Finding First-Order Stationary Points

**Proof of Proposition 4.1**

We fix $k \in [K] \cup \{0\}$, $t \in [T - 1] \cup \{0\}$ and $s \in [S - 1] \cup \{0\}$. From $L$-smoothness of $f$, we have

$$f(x_{I(k+1,t,s)}) \le f(x_{I(k,t,s)}) + \langle \nabla f(x_{I(k,t,s)}), x_{I(k+1,t,s)} - x_{I(k,t,s)} \rangle + \frac{L}{2} \|x_{I(k+1,t,s)} - x_{I(k,t,s)}\|^2.$$

From this inequality, we have

$$\begin{aligned}
f(x_{I(k+1,t,s)}) &\leq f(x_{I(k,t,s)}) + \langle \nabla f(x_{I(k,t,s)}) - v_{I(k,t,s)} + \xi_{I(k,t,s)}, x_{I(k+1,t,s)} - x_{I(k,t,s)} \rangle \\
&\quad + \langle v_{I(k,t,s)} - \xi_{I(k,t,s)}, x_{I(k+1,t,s)} - x_{I(k,t,s)} \rangle + \frac{L}{2} \|x_{I(k+1,t,s)} - x_{I(k,t,s)}\|^2 \\
&= f(x_{I(k,t,s)}) + \langle \nabla f(x_{I(k,t,s)}) - v_{I(k,t,s)} + \xi_{I(k,t,s)}, x_{I(k+1,t,s)} - x_{I(k,t,s)} \rangle \\
&\quad - \left(\frac{1}{\eta} - \frac{L}{2}\right) \|x_{I(k+1,t,s)} - x_{I(k,t,s)}\|^2 \\
&= f(x_{I(k,t,s)}) + \frac{\eta}{2} \|v_{I(k,t,s)} - \xi_{I(k,t,s)} - \nabla f(x_{I(k,t,s)})\|^2 - \frac{\eta}{2} \|\nabla f(x_{I(k,t,s)})\|^2 \\
&\quad + \frac{1}{2\eta} \|x_{I(k+1,t,s)} - x_{I(k,t,s)}\|^2 - \left(\frac{1}{\eta} - \frac{L}{2}\right) \|x_{I(k+1,t,s)} - x_{I(k,t,s)}\|^2 \\
&= f(x_{I(k,t,s)}) + \frac{\eta}{2} \|v_{I(k,t,s)} - \xi_{I(k,t,s)} - \nabla f(x_{I(k,t,s)})\|^2 - \frac{\eta}{2} \|\nabla f(x_{I(k,t,s)})\|^2 \\
&\quad - \left(\frac{1}{2\eta} - \frac{L}{2}\right) \|x_{I(k+1,t,s)} - x_{I(k,t,s)}\|^2 \\
&\leq f(x_{I(k,t,s)}) + \eta \|v_{I(k,t,s)} - \nabla f(x_{I(k,t,s)})\|^2 - \frac{\eta}{2} \|\nabla f(x_{I(k,t,s)})\|^2 \\
&\quad - \left(\frac{1}{2\eta} - \frac{L}{2}\right) \|x_{I(k+1,t,s)} - x_{I(k,t,s)}\|^2 + \eta \|\xi_I(k,t,s)\|^2 \\
&\leq f(x_{I(k,t,s)}) + \eta \|v_{I(k,t,s)} - \nabla f(x_{I(k,t,s)})\|^2 - \frac{\eta}{2} \|\nabla f(x_{I(k,t,s)})\|^2 \\
&\quad - \left(\frac{1}{2\eta} - \frac{L}{2}\right) \|x_{I(k+1,t,s)} - x_{I(k,t,s)}\|^2 + \eta r^2. \tag{1}
\end{aligned}$$

Here, for the first equality we used the fact $v_{I(k,t,s)} - \xi_{I(k,t,s)} = -(1/\eta)(x_{I(k+1,t,s)} - x_{I(k,t,s)})$. The second equality follows from the facts $v_{I(k,t,s)} - \xi_{I(k,t,s)} = (1/\eta)(x_{I(k+1,t,s)} - x_{I(k,t,s)})$ and $\langle a - b, -b \rangle = (1/2)(\|a - b\|^2 - \|a\|^2 + \|b\|^2)$ for any $a, b \in \mathbb{R}^d$. For the second inequality, we used the relation $\|a + b\| \leq 2(\|a\|^2 + \|b\|^2)$ for any $a, b \in \mathbb{R}^d$. The last inequality holds from the definition of $\xi_{I(k,t,s)}$.

Thus, for every $k, k_0 \in [K-1]$, $t, t_0 \in [T-1]$ and $s, s_0 \in [S-1]$ $(I(k,t,s) \geq I(k_0,t_0,s_0))$, we have

$$\begin{aligned}
f(x_{I(k,t,s)}) &\leq f(x_{I(k_0,t_0,s_0)}) + \eta \sum_{i=I(k_0,t_0,s_0)}^{I(k,t,s)-1} \|v_i - \nabla f(x_i)\|^2 \\
&\quad - \frac{\eta}{2} \sum_{i=I(k_0,t_0,s_0)}^{I(k,t,s)-1} \|\nabla f(x_i)\|^2 - \left(\frac{1}{2\eta} - \frac{L}{2}\right) \sum_{i=I(k_0,t_0,s_0)}^{I(k,t,s)-1} \|x_{i+1} - x_i\|^2 \\
&\quad + \eta(I(k,t,s) - I(k_0,t_0,s_0))r^2. \tag{2}
\end{aligned}$$

Now we bound the deviation $\|v_{I(k,t,s)} - \nabla f(x_{I(k,t,s)})\|^2$. Observe that

$$
\begin{aligned}
v_{I(k,t,s)} - \nabla f(x_{I(k,t,s)}) &= g_{I(k,t,s)} - g_{I(k,t,s)}^{\mathrm{ref}} + v_{I(k-1,t,s)} - \nabla f(x_{I(k,t,s)}) \\
&= g_{I(k,t,s)} - g_{I(k,t,s)}^{\mathrm{ref}} + \nabla f_{p_{t,s}}(x_{I(k-1,t,s)}) - \nabla f_{p_{t,s}}(x_{I(k,t,s)}) \\
&\quad + \nabla f_{p_{t,s}}(x_{I(k,t,s)}) - \nabla f_{p_{t,s}}(x_{I(k-1,t,s)}) + \nabla f(x_{I(k-1,t,s)}) - \nabla f(x_{I(k,t,s)}) \\
&\quad + v_{I(k-1,t,s)} - \nabla f(x_{I(k-1,t,s)}) \\
&= \sum_{\kappa=0}^{k-1}(g_{I(\kappa+1,t,s)} - g_{I(\kappa+1,t,s)}^{\mathrm{ref}} + \nabla f_{p_{t,s}}(x_{I(\kappa,t,s)}) - \nabla f_{p_{t,s}}(x_{I(\kappa+1,t,s)})) \\
&\quad + \sum_{\kappa=0}^{k-1}(\nabla f_{p_{t,s}}(x_{I(\kappa+1,t,s)}) - \nabla f_{p_{t,s}}(x_{I(\kappa,t,s)}) + \nabla f(x_{I(\kappa,t,s)}) - \nabla f(x_{I(\kappa+1,t,s)})) \\
&\quad + v_{I(0,t,s)} - \nabla f(x_{I(0,t,s)}).
\end{aligned}
$$

Further, we have

$$
\begin{aligned}
v_{I(0,t,s)} - \nabla f(x_{I(0,t,s)}) &= \frac{1}{P}\sum_{p=1}^{P}(g_{I(0,t,s)}^{(p)} - g_{I(0,t,s)}^{(p),\mathrm{ref}} + v_{I(0,t-1,s)} - \nabla f(x_{I(0,t,s)}) \\
&= \frac{1}{P}\sum_{p=1}^{P}(g_{I(0,t,s)}^{(p)} - g_{I(0,t,s)}^{(p),\mathrm{ref}} + \nabla f(x_{I(0,t-1,s)}) - \nabla f(x_{I(0,t,s)}) \\
&\quad + v_{I(0,t-1,s)} - \nabla f(x_{I(0,t-1,s)}) \\
&= \sum_{\tau=0}^{t-1}\frac{1}{P}\sum_{p=1}^{P}(g_{I(0,\tau+1,s)}^{(p)} - g_{I(0,\tau+1,s)}^{(p),\mathrm{ref}} + \nabla f(x_{I(0,\tau,s)}) - \nabla f(x_{I(0,\tau+1,s)}) \\
&\quad + v_{I(0,0,s)} - \nabla f(x_{I(0,0,s)}).
\end{aligned}
$$

Note that the last term is exactly zero from the definition of $v_{I(0,0,s)}$.

We define

$$
\begin{cases}
\alpha_{I(\kappa,t,s)} := & g_{I(\kappa,t,s)} - g_{I(\kappa,t,s)}^{\mathrm{ref}} + \nabla f_{p_{t,s}}(x_{I(\kappa-1,t,s)}) - \nabla f_{p_{t,s}}(x_{I(\kappa,t,s)}), \\
\beta_{I(\kappa,t,s)} := & \nabla f_{p_{t,s}}(x_{I(\kappa,t,s)}) - \nabla f_{p_{t,s}}(x_{I(\kappa-1,t,s)}) + \nabla f(x_{I(\kappa-1,t,s)}) - \nabla f(x_{I(\kappa,t,s)}), \\
\gamma_{I(0,\tau,s)} := & \frac{1}{P}\sum_{p=1}^{P}(g_{I(0,\tau,s)}^{(p)} - g_{I(0,\tau,s)}^{(p),\mathrm{ref}} + \nabla f(x_{I(0,\tau-1,s)}) - \nabla f(x_{I(0,\tau,s)}),
\end{cases}
$$

and

$$
\begin{cases}
A_{I(k,t,s)} := & \sum_{\kappa=0}^{k-1}\alpha_{I(\kappa+1,t,s)}, \\
B_{I(k,t,s)} := & \sum_{\kappa=0}^{k-1}\beta_{I(\kappa+1,t,s)}, \\
C_{I(0,t,s)} := & \sum_{\tau=0}^{t-1}\gamma_{I(0,\tau+1,s)}.
\end{cases}
$$

Note that $\mathbb{E}[A_{I(k,t,s)}] = \mathbb{E}[C_{I(k,t,s)}] = 0$. Using these definitions, we have

$$
\|v_{I(k,t,s)} - \nabla f(x_{I(k,t,s)})\|^2 \le 3(\|A_{I(k,t,s)}\|^2 + \|B_{I(k,t,s)}\|^2 + \|C_{I(k,t,s)}\|^2).
$$

We denote all the randomness up to iteration $I(\kappa-1,t,s)$ as $\mathfrak{F}_{I(\kappa-1,t,s)}$.

**Bounding $\|A_{I(k,t,s)}\|$**

Let $\alpha_{l,I(\kappa,t,s)} := \nabla\ell(x_{I(\kappa,t,s)}, z_{l,I(\kappa,t,s)}) - \nabla\ell(x_{I(\kappa-1,t,s)}, z_{l,I(\kappa,t,s)}) + \nabla f_{p_{t,s}}(x_{I(\kappa-1,t,s)}) + \nabla f_{p_{t,s}}(x_{I(\kappa,t,s)})$. Then, $\alpha_{I(\kappa,t,s)} = (1/b)\sum_{l=1}^{b}\alpha_{l,I(\kappa,t,s)}$. Observe that $\alpha_{l,I(\kappa,t,s)}$ satisfies

$$
\mathbb{E}[\alpha_{l,I(\kappa,t,s)} \mid \mathfrak{F}_{I(\kappa-1,t,s)}] = 0
$$

and

$$
\mathbb{P}(\|\alpha_{l,I(\kappa,t,s)}\| \ge s \mid \mathfrak{F}_{I(\kappa-1,t,s)}) \le 2e^{-\frac{s^2}{2\left(\sigma_{I(\kappa,t,s)}^{(\alpha)}\right)^2}}
$$

for every $s \in \mathbb{R}$ and $\kappa \in [k]$, where $\sigma_{I(\kappa,t,s)}^{(\alpha)} := 2L\|x_{I(\kappa,t,s)} - x_{I(\kappa-1,t,s)}\|$. Here, we used the fact that $\|\nabla \ell(x_{I(\kappa,t,s)}, z_{l,I(\kappa,t,s)}) - \nabla \ell(x_{I(\kappa-1,t,s)}, z_{l,I(\kappa,t,s)}) + \nabla f_{p_{t,s}}(x_{I(\kappa-1,t,s)}) + \nabla f_{p_{t,s}}(x_{I(\kappa,t,s)})\| \le 2L\|x_{I(\kappa,t,s)} - x_{I(\kappa-1,t,s)}\|$ from $L$-smoothness of $\ell$. Note that $\{\alpha_{l,I(\kappa,t,s)}\}_{l=1}^{b_k}$ is I.I.D. sequence with at least $b$ samples and $\|\alpha_{\ell,I(\kappa,t,s)}\| \le 4G$ almost surely from Assumption 5. From these results, we can use Lemma B.2 with $A = 4KG$ and $a = \widetilde{\varepsilon}$ ($\widetilde{\varepsilon}$ is some positive number and will be defined later) and get

$$\|A_I(k,t,s)\|^2 \le \frac{c^2}{b} \left( \left( \sum_{\kappa=0}^{k-1} \left( \sigma_{I(\kappa+1,t,s)}^{(\alpha)} \right)^2 \right) + \widetilde{\varepsilon} \right) \left( \log \frac{2d}{q} + \log\log \frac{4KG}{\widetilde{\varepsilon}} \right)$$

with probability at least $1 - q$ for some constant $c > 0$. Also, note that $\|A_I(k,t,s)\| \le 4kG$ almost surely.

**Bounding $\|B_{I(k,t,s)}\|$**

Observe that

$$\begin{aligned}
\beta_{I(\kappa,t,s)} &= \nabla f_{p_{t,s}}(x_{I(\kappa,t,s)}) - \nabla f_{p_{t,s}}(x_{I(\kappa-1,t,s)}) + \nabla f(x_{I(\kappa-1,t,s)}) - \nabla f(x_{I(\kappa,t,s)}) \\
&= (\nabla f_{p_{t,s}} - \nabla f)(x_{I(\kappa,t,s)}) - (\nabla f_{p_{t,s}} - \nabla f)(x_{I(\kappa-1,t,s)}) \\
&= \left( \int_0^1 (\nabla^2 f_{p_{t,s}} - \nabla^2 f)(\theta x_{I(\kappa,t,s)} + (1-\theta)x_{I(\kappa-1,t,s)})d\theta \right) (x_{I(\kappa,t,s)} - x_{I(\kappa-1,t,s)}).
\end{aligned}$$

Hence, from Assumption 1, we get

$$\|\beta_{I(\kappa,t,s)}\| \le \zeta \|x_{I(\kappa,t,s)} - x_{I(\kappa-1,t,s)}\| =: \sigma_{I(\kappa,t,s)}^{(\beta)}.$$

This gives

$$\left\| B_{I(k,t,s)} \right\|^2 \le k \sum_{\kappa=0}^{k-1} \left( \sigma_{I(\kappa+1,t,s)}^{(\beta)} \right)^2.$$

Here we used the relation $\left( \sum_{i=1}^m |a_i| \right)^2 \le m \sum_{i=1}^m a_i^2$ for every $\{a_i\}_{i=1}^m \subset \mathbb{R}$. Also, note that $\|B_I(k,t,s)\| \le 4kG$ almost surely.

**Bounding $\|C_{I(0,t,s)}\|$**

The argument is similar to the one of the case of the first term. From Lemma B.2, the third term $\|C_{I(0,t,s)}\|$ can be bounded as

$$\|C_I(0,t,s)\|^2 \le \frac{c^2}{PKb} \left( \left( \sum_{\tau=0}^{t-1} \left( \sigma_{I(0,\tau+1,s)}^{(\gamma)} \right)^2 \right) + \widetilde{\varepsilon} \right) \left( \log \frac{2d}{q} + \log\log \frac{4KTG}{\widetilde{\varepsilon}} \right),$$

with probability at least $1 - q$, where $\sigma_{I(0,\tau,s)}^{(\gamma)} := 2L \sum_{\kappa=0}^{K-1} \|x_{I(\kappa+1,\tau-1,s)} - x_{I(\kappa,\tau-1,s)}\|$ ($\ge 2L\|x_{I(0,\tau,s)} - x_{I(0,\tau-1,s)}\|$). Here, we used the fact that $\{g_{I(0,\tau,s)}^{(p)} - g_{I(0,\tau,s)}^{(p),\mathrm{ref}}\}_{p=1}^P$ is independent and each of them is constructed from $Kb$ i.i.d. data samples. Also, note that $\|C_I(k,t,s)\| \le 4TG$ almost surely.

Put the three results all together, we obtain

$$\begin{aligned}
\|v_{I(k,t,s)} - \nabla f(x_{I(k,t,s)})\|^2 \le{}& \frac{3c^2}{b} \left( \left( \sum_{\kappa=0}^{k-1} \left( \sigma_{I(\kappa+1,t,s)}^{(\alpha)} \right)^2 \right) + \widetilde{\varepsilon} \right) \left( \log \frac{2KTSd}{q} + \log\log \frac{4KG}{\widetilde{\varepsilon}} \right) \\
&+ 3k \sum_{\kappa=0}^{k-1} \left( \sigma_{I(\kappa+1,t,s)}^{(\beta)} \right)^2 \\
&+ \frac{3c^2}{PKb} \left( \left( \sum_{\tau=0}^{t-1} \left( \sigma_{I(0,\tau+1,s)}^{(\gamma)} \right)^2 \right) + \widetilde{\varepsilon} \right) \left( \log \frac{2KTSd}{q} + \log\log \frac{4TG}{\widetilde{\varepsilon}} \right)
\end{aligned}$$

for every $k \in [K-1]$, $t \in [T-1]$ and $s \in [S-1]$ with probability at least $1 - 3q$ for some constant $c > 0$. We set $q \leftarrow q/(KTS)$. Now, we set

$$6c^2\widetilde{\varepsilon}\left(\log\frac{2KTSd}{q} + \log\log\frac{4KTG}{\widetilde{\varepsilon}}\right) \leq r^2.$$

Then, we have

$$
\begin{aligned}
\|v_{I(k,t,s)} - \nabla f(x_{I(k,t,s)})\|^2 \leq{}& \frac{3c^2}{b}\left(\sum_{\kappa=0}^{k-1}\left(\sigma^{(\alpha)}_{I(\kappa+1,t,s)}\right)^2\right)\left(\log\frac{2KTSd}{q} + \log\log\frac{G}{\widetilde{\varepsilon}}\right) \\
&+ 3k\sum_{\kappa=0}^{k-1}\left(\sigma^{(\beta)}_{I(\kappa+1,t,s)}\right)^2 \\
&+ \frac{3c^2}{PKb}\left(\sum_{\tau=0}^{t-1}\left(\sigma^{(\gamma)}_{I(0,\tau+1,s)}\right)^2\right)\left(\log\frac{2KTSd}{q} + \log\log\frac{G}{\widetilde{\varepsilon}}\right) \\
&+ r^2
\end{aligned}
\tag{3}
$$

for every $I(k,t,s) \in [KTS] \cup \{0\}$.

Let

$$
\begin{aligned}
V(k,t,s) :={}& 12c^2\left(\frac{L^2}{b} + K\zeta^2 + \frac{L^2T}{Pb}\right)\left(\sum_{\kappa=0}^{k-1}\|x_{I(\kappa+1,t,s)} - x_{I(\kappa,t,s)}\|^2 + \frac{1}{T}\sum_{\tau=0}^{t-1}\sum_{\kappa=0}^{K-1}\|x_{I(\kappa+1,\tau,s)} - x_{I(\kappa,\tau,s)}\|^2\right) \\
&\times\left(\log\frac{2KTSd}{q} + \log\log\frac{4KTG}{\widetilde{\varepsilon}}\right).
\end{aligned}
$$

Observe that $\|v_{I(k,t,s)} - \nabla f(x_{I(k,t,s)})\|^2 \leq V(k,t,s) + r^2$ and $V(k,t,s) \leq V(k',t',s)$ for $k' \geq k$ and $t' \geq t$.

Now, we bound $\sum_{i=I(k_0,t_0,s_0)}^{I(k,t,s)}\|v_i - \nabla f(x_i)\|^2$ by dividing three cases.

**Case I.** $s = s_0$ and $t = t_0$.

We bound $\sum_{i=I(k_-,t_-,s_-)}^{I(k,t,s)}\|v_i - \nabla f(x_i)\|^2$ for general $k_-, t_-$ and $s_-$ with $k_- \leq k$, $t_- = t$ and $s_- = s$.

$$
\begin{aligned}
&\sum_{i=I(k_-,t_-,s_-)}^{I(k,t_-,s_-)}\|v_i - \nabla f(x_i)\|^2 \\
&\leq \sum_{k'=k_-}^{k} V(k',t_-,s_-) + (k - k_- + 1)r^2 \\
&\leq (k - k_- + 1)V(k,t_-,s_-) + (k - k_- + 1)r^2 \\
&\leq 12c^2\left(\frac{KL^2}{b} + K^2\zeta^2 + \frac{KL^2T}{Pb}\right) \\
&\quad\times\left(\frac{k - k_- + 1}{K}\sum_{\kappa=0}^{k-1}\|x_{I(\kappa+1,t_-,s_-)} - x_{I(\kappa,t_-,s_-)}\|^2 + \frac{k - k_- + 1}{KT}\sum_{\tau=0}^{t_--1}\sum_{\kappa=0}^{K-1}\|x_{I(\kappa+1,\tau,s_-)} - x_{I(\kappa,\tau,s_-)}\|^2\right) \\
&\quad\times\left(\log\frac{2KTSd}{q} + \log\log\frac{4KTG}{\widetilde{\varepsilon}}\right) \\
&\quad+ (k - k_- + 1)r^2.
\end{aligned}
$$

Since

$$\frac{k-k_-+1}{K}\sum_{\kappa=0}^{k-1}\|x_{I(\kappa+1,t_-,s_-)}-x_{I(\kappa,t_-,s_-)}\|^2+\frac{k-k_-+1}{KT}\sum_{\tau=0}^{t_--1}\sum_{\kappa=0}^{K-1}\|x_{I(\kappa+1,\tau,s_-)}-x_{I(\kappa,\tau,s_-)}\|^2$$

$$\leq\sum_{\kappa=k_-}^{k-1}\|x_{I(\kappa+1,t_-,s_-)}-x_{I(\kappa,t_-,s_-)}\|^2+\frac{k-k_-+1}{K}\sum_{\kappa=0}^{k_--1}\|x_{I(\kappa+1,t_-,s_-)}-x_{I(\kappa,t_-,s_-)}\|^2$$

$$+\frac{k-k_-+1}{KT}\sum_{\tau=0}^{t_--1}\sum_{\kappa=0}^{K-1}\|x_{I(\kappa+1,\tau,s_-)}-x_{I(\kappa,\tau,s_-)}\|^2$$

$$=\sum_{i=I(k_-,t_-,s_-)}^{I(k,t_-,s_-)-1}\|x_{i+1}-x_i\|^2+\frac{(I(k,t,s)-I(k_-,t_-,s_-)+1)\wedge K}{K}\sum_{i=I(0,t_-,s_-)}^{I(k_-,t_-,s_-)-1}\|x_{i+1}-x_i\|^2$$

$$+\frac{(I(k,t_-,s_-)-I(k_-,t_-,s_-)+1)\wedge KT}{KT}\sum_{i=I(0,0,s_-)}^{I(0,t_-,s_-)-1}\|x_{i+1}-x_i\|^2,$$

we get

$$\sum_{i=I(k_-,t_-,s_-)}^{I(k,t_-,s_-)}\|v_i-\nabla f(x_i)\|^2$$

$$\leq 12c^2\left(\frac{KL^2}{b}+K^2\zeta^2+\frac{KL^2T}{Pb}\right)$$

$$\times\left(\sum_{i=I(k_-,t_-,s_-)}^{I(k,t_-,s_-)-1}\|x_{i+1}-x_i\|^2+\frac{(I(k,t_-,s_-)-I(k_-,t_-,s_-)+1)\wedge K}{K}\sum_{i=I(0,t_-,s_-)}^{I(k_-,t_-,s_-)-1}\|x_{i+1}-x_i\|^2\right.$$

$$\left.+\frac{(I(k,t_-,s_-)-I(k_-,t_-,s_-)+1)\wedge KT}{KT}\sum_{i=I(0,0,s_-)}^{I(0,t_-,s_-)-1}\|x_{i+1}-x_i\|^2\right)$$

$$\times\left(\log\frac{2KTSd}{q}+\log\log\frac{4KTG}{\widetilde{\varepsilon}}\right)$$

$$+(I(k,t_-,s_-)-I(k_-,t_-,s_-)+1)r^2.$$

Setting $k_-\leftarrow k_0$, $t_-\leftarrow t_0$ and $s_-\leftarrow s_0$ gives the desired bound.

**Case II.** $s=s_0$ and $t>t_0$.

Note that $I(k,t,s_0)-I(k_0,t_0,s_0)\geq K$. Again, we consider $\sum_{i=I(k_-,t_-,s_-)}^{I(k,t,s_-)}\|v_i-\nabla f(x_i)\|^2$ for general $k_-,t_-$ and $s_-$ with $k\geq k_-$, $t>t_-$ and $s=s_-$.

$$\sum_{i=I(k_-,t_-,s_-)}^{I(k,t,s_-)}\|v_i-\nabla f(x_i)\|^2$$

$$\leq\sum_{i=I(k_-,t_-,s_-)}^{I(K-1,t_-,s_-)}\|v_i-\nabla f(x_i)\|^2+\sum_{t'=t_-+1}^{t-1}\sum_{i=I(0,t',s_-)}^{I(K-1,t',s_-)}\|v_i-\nabla f(x_i)\|^2+\sum_{i=I(0,t,s_-)}^{I(k,t,s_-)}\|v_i-\nabla f(x_i)\|^2.$$

Using the result of Case I, the first term can be bounded as follows:

$$\sum_{i=I(k_-,t_-,s_-)}^{I(K-1,t_-,s_-)} \|v_i - \nabla f(x_i)\|^2$$

$$\leq 12c^2 \left( \frac{KL^2}{b} + K^2\zeta^2 + \frac{KL^2T}{Pb} \right)$$

$$\times \left( \sum_{i=I(k_-,t_-,s_-)}^{I(K-1,t_-,s_-)-1} \|x_{i+1} - x_i\|^2 + \frac{(I(K-1,t_-,s_-) - I(k_-,t_-,s_-)+1) \wedge K}{K} \sum_{i=I(0,t_-,s_-)}^{I(k_-,t_-,s_-)-1} \|x_{i+1} - x_i\|^2 \right.$$

$$\left. + \frac{(I(K-1,t_-,s_-) - I(k_-,t_-,s_-)+1) \wedge KT}{KT} \sum_{i=I(0,0,s_-)}^{I(0,t_-,s_-)-1} \|x_{i+1} - x_i\|^2 \right)$$

$$\times \left( \log \frac{2KTSd}{q} + \log\log \frac{4KTG}{\widetilde{\varepsilon}} \right)$$

$$+ (I(K-1,t_-,s_-) - I(k_-,t_-,s_-)+1)r^2.$$

Similarly, the second term can be bounded as:

$$\sum_{t'=t_-+1}^{t-1} \sum_{i=I(0,t',s_-)}^{I(K-1,t',s_-)} \|v_i - \nabla f(x_i)\|^2$$

$$\leq 12c^2 \left( \frac{KL^2}{b} + K^2\zeta^2 + \frac{KL^2T}{Pb} \right)$$

$$\times \left( \sum_{t'=t_-+1}^{t-1} \sum_{i=I(0,t',s_-)}^{I(K-1,t',s_-)-1} \|x_{i+1} - x_i\|^2 \right.$$

$$\left. + \sum_{t'=t_-+1}^{t-1} \frac{(I(K-1,t',s_-) - I(0,t',s_-)+1) \wedge KT}{KT} \sum_{i=I(0,0,s_-)}^{I(0,t',s_-)-1} \|x_{i+1} - x_i\|^2 \right)$$

$$\times \left( \log \frac{2KTSd}{q} + \log\log \frac{4KTG}{\widetilde{\varepsilon}} \right)$$

$$\leq 12c^2 \left( \frac{KL^2}{b} + K^2\zeta^2 + \frac{KL^2T}{Pb} \right)$$

$$\times \left( \sum_{i=I(0,t_-+1,s_-)}^{I(K-1,t-1,s_-)-1} \|x_{i+1} - x_i\|^2 + \sum_{t'=t_-+1}^{t-1} \frac{(I(K-1,t',s_-) - I(0,t',s_-)+1) \wedge KT}{KT} \sum_{i=I(0,t_-,s_-)}^{I(0,t',s_-)-1} \|x_{i+1} - x_i\|^2 \right.$$

$$\left. + \frac{(I(K-1,t-1,s_-) - I(0,t_-+1,s_-)+1) \wedge KT}{KT} \sum_{i=I(0,0,s_-)}^{I(0,t_-,s_-)-1} \|x_{i+1} - x_i\|^2 \right)$$

$$\times \left( \log \frac{2KTSd}{q} + \log\log \frac{4KTG}{\widetilde{\varepsilon}} \right)$$

$$+ (I(K-1,t-1,s_-) - I(0,t_-+1,s_-)+1)r^2.$$

Now, we bound the second term as follows:

$$\sum_{t'=t_-+1}^{t-1} \frac{(I(K-1,t',s_-)-I(0,t',s_-)+1)\wedge KT}{KT} \sum_{i=I(0,t_-,s_-)}^{I(0,t',s_-)-1} \|x_{i+1}-x_i\|^2$$

$$\leq \frac{(I(K-1,t_-+1,s_-)-I(0,t_-+1,s_-)+1)\wedge KT}{KT} \sum_{i=I(0,t_-,s_-}^{I(k_-,t_-,s_-)-1} \|x_{i+1}-x_i\|^2 + \sum_{i=I(k_-,t_-,s_-)}^{I(0,t_-+1,s_-)-1} \|x_{i+1}-x_i\|^2$$

$$+ \sum_{t'=t_-+2}^{t-1} \frac{(I(K-1,t',s_-)-I(0,t',s_-)+1)\wedge KT}{KT} \sum_{i=I(0,t_-,s_-)}^{I(0,t',s_-)-1} \|x_{i+1}-x_i\|^2$$

$$\leq \frac{(I(K-1,t_-+1,s_-)-I(0,t_-+1,s_-)+1)\wedge KT}{KT} \sum_{i=I(0,t_-,s_-}^{I(k_-,t_-,s_-)-1} \|x_{i+1}-x_i\|^2 + \sum_{i=I(k_-,t_-,s_-)}^{I(K-1,t-1,s_)-1} \|x_{i+1}-x_i\|^2.$$

Using this, we have

$$\sum_{t'=t_-+1}^{t-1} \sum_{i=I(0,t',s_-)}^{I(K-1,t',s_-)} \|v_i-\nabla f(x_i)\|^2$$

$$\leq 12c^2\left(\frac{KL^2}{b}+K^2\zeta^2+\frac{KL^2T}{Pb}\right)$$

$$\times \left(2\sum_{i=I(k_-,t_-,s_-)}^{I(K-1,t-1,s_-)-1} \|x_{i+1}-x_i\|^2 + \frac{(I(K-1,t_-+1,s_-)-I(0,t_-+1,s_-)+1)\wedge K}{K}\sum_{i=I(0,t_-,s_-)}^{I(k_-,t_-,s_-)-1} \|x_{i+1}-x_i\|^2\right.$$

$$\left.+ \frac{(I(K-1,t-1,s_-)-I(0,t_-+1,s_-)+1)\wedge KT}{KT}\sum_{i=I(0,0,s_-)}^{I(0,t_-,s_-)-1} \|x_{i+1}-x_i\|^2\right)$$

$$\times \left(\log\frac{2KTSd}{q}+\log\log\frac{4KTG}{\widetilde{\varepsilon}}\right)$$

$$+ (I(K-1,t-1,s_-)-I(0,t_-+1,s_-)+1)r^2.$$

Finally, we bound the last term:

$$\sum_{i=I(0,t,s_-)}^{I(k,t,s_-)-1} \|v_i-\nabla f(x_i)\|^2$$

$$\leq 12c^2\left(\frac{KL^2}{b}+K^2\zeta^2+\frac{KL^2T}{Pb}\right)$$

$$\times \left(\sum_{i=I(0,t,s_-)}^{I(k,t,s_-)-1} \|x_{i+1}-x_i\|^2 + \frac{(I(k,t,s_-)-I(0,t,s_-)+1)\wedge KT}{KT}\sum_{i=I(0,0,s_-)}^{I(0,t,s_-)-1} \|x_{i+1}-x_i\|^2\right)$$

$$\times \left(\log\frac{2KTSd}{q}+\log\log\frac{4KTG}{\widetilde{\varepsilon}}\right)$$

$$\leq 12c^2\left(\frac{KL^2}{b}+K\zeta^2+\frac{KL^2T}{Pb}\right)$$

$$\times \left(\sum_{i=I(k_-,t_-,s_-)}^{I(k,t,s_-)-1} \|x_{i+1}-x_i\|^2 + \frac{(I(k,t,s_-)-I(0,t,s_-)+1)\wedge KT}{KT}\sum_{i=I(0,0,s_-)}^{I(k_-,t_-,s_-)-1} \|x_{i+1}-x_i\|^2\right)$$

$$\times \left(\log\frac{2KTSd}{q}+\log\log\frac{4KTG}{\widetilde{\varepsilon}}\right)$$

$$+ (I(k,t,s_-)-I(0,t,s_-)+1)r^2.$$

Summing the upper bounds of the three terms, we get

$$\sum_{i=I(k_-,t_-,s_-)}^{I(k,t,s_-)} \|v_i - \nabla f(x_i)\|^2$$

$$\leq 48c^2 \left( \frac{KL^2}{b} + K^2\zeta^2 + \frac{KL^2T}{Pb} \right)$$

$$\times \left( \sum_{i=I(k_-,t_-,s_-)}^{I(k,t,s_-)-1} \|x_{i+1} - x_i\|^2 + \frac{(I(k,t,s_-) - I(k_-,t_-,s_-) + 1) \wedge K}{K} \sum_{i=I(0,t_-,s_-)}^{I(k_-,t_-,s_-)-1} \|x_{i+1} - x_i\|^2 \right.$$

$$\left. + \frac{(I(k,t,s_-) - I(k_-,t_-,s_-) + 1) \wedge KT}{KT} \sum_{i=I(0,0,s_-)}^{I(0,t_-,s_-)-1} \|x_{i+1} - x_i\|^2 \right)$$

$$\times \left( \log \frac{2KTSd}{q} + \log\log \frac{4KTG}{\widetilde{\varepsilon}} \right)$$

$$+ (I(k,t,s_-) - I(k_-,t_-,s_-) + 1)r^2.$$

Setting $k_- \leftarrow k_0$, $t_- \leftarrow t_0$ and $s_- \leftarrow s_0$ gives the desired bound.

**Case III.** $s > s_0$

In this case, note that $I(k,t,s) - I(k_0,t_0,s_0) \geq KT$ holds. Observe that

$$\sum_{i=I(k_0,t_0,s_0)}^{I(k,t,s)} \|v_i - \nabla f(x_i)\|^2$$

$$\leq \sum_{i=I(k_0,t_0,s_0)}^{I(K-1,T-1,s_0)} \|v_i - \nabla f(x_i)\|^2 + \sum_{s'=s_0+1}^{s-1} \sum_{i=I(0,0,s')}^{I(K-1,T-1,s')} \|v_i - \nabla f(x_i)\|^2 + \sum_{i=I(0,0,s)}^{I(k,t,s)} \|v_i - \nabla f(x_i)\|^2.$$

Using the result of Case II, we bound the three terms.

The first term can be bounded as follows:

$$\sum_{i=I(k_0,t_0,s_0)}^{I(K-1,T-1,s_0)} \|v_i - \nabla f(x_i)\|^2$$

$$\leq 48c^2 \left( \frac{KL^2}{b} + K^2\zeta^2 + \frac{KL^2T}{Pb} \right)$$

$$\times \left( \sum_{i=I(k_0,t_0,s_0)}^{I(K-1,T-1,s_0)-1} \|x_{i+1} - x_i\|^2 + \frac{(I(K-1,T-1,s_0) - I(k_0,t_0,s_0) + 1) \wedge K}{K} \sum_{i=I(0,t_0,s_0)}^{I(k_0,t_0,s_0)-1} \|x_{i+1} - x_i\|^2 \right.$$

$$\left. + \frac{(I(K-1,T-1,s_0) - I(k_0,t_0,s_0) + 1) \wedge KT}{KT} \sum_{i=I(0,0,s_0)}^{I(0,t_0,s_0)-1} \|x_{i+1} - x_i\|^2 \right)$$

$$\times \left( \log \frac{2KTSd}{q} + \log\log \frac{G}{\widetilde{\varepsilon}} \right)$$

$$+ (I(K-1,T-1,s_0) - I(k_0,t_0,s_0) + 1)r^2.$$

Similarly, the second term can be bounded as

$$\sum_{s'=s_0+1}^{s-1} \sum_{i=I(0,0,s')}^{I(K-1,T-1,s')} \|v_i - \nabla f(x_i)\|^2$$

$$\leq 48c^2 \left(\frac{KL^2}{b} + K^2\zeta^2 + \frac{KL^2T}{Pb}\right) \left(\sum_{s'=s_0+1}^{s-1} \sum_{i=I(0,0,s')}^{I(K-1,T-1,s')-1} \|x_{i+1} - x_i\|^2\right) \left(\log\frac{2KTSd}{q} + \log\log\frac{4KTG}{\widetilde{\varepsilon}}\right)$$

$$+ (I(K-1,T-1,s-1) - I(k_0,t_0,s_0+1) + 1)r^2.$$

We bound the last term as

$$\sum_{i=I(0,0,s)}^{I(k,t,s)} \|v_i - \nabla f(x_i)\|^2$$

$$\leq 48c^2 \left(\frac{KL^2}{b} + K^2\zeta^2 + \frac{KL^2T}{Pb}\right)$$

$$\times \left(\sum_{i=I(0,0,s)}^{I(k,t,s)-1} \|x_{i+1} - x_i\|^2\right) \left(\log\frac{KTSd}{q} + \log\log\frac{4KTG}{\widetilde{\varepsilon}}\right)$$

$$+ (I(k,t,s) - I(0,0,s) + 1)r^2.$$

Summing up the three terms, we get

$$\sum_{i=I(k_0,t_0,s_0)}^{I(k,t,s)} \|v_i - \nabla f(x_i)\|^2$$

$$\leq 48c^2 \left(\frac{KL^2}{b} + K^2\zeta^2 + \frac{KL^2T}{Pb}\right)$$

$$\times \left(\sum_{i=I(k_0,t_0,s_0)}^{I(k,t,s)-1} \|x_{i+1} - x_i\|^2 + \frac{(I(k,t,s) - I(k_0,t_0,s_0) + 1) \wedge K}{K} \sum_{i=I(0,t_0,s_0)}^{I(k_0,t_0,s_0)-1} \|x_{i+1} - x_i\|^2\right.$$

$$\left. + \frac{(I(k,t,s) - I(k_0,t_0,s_0) + 1) \wedge KT}{KT} \sum_{i=I(0,0,s_0)}^{I(0,t_0,s_0)-1} \|x_{i+1} - x_i\|^2\right)$$

$$\times \left(\log\frac{2KTSd}{q} + \log\log\frac{4KTG}{\widetilde{\varepsilon}}\right)$$

$$+ (I(k,t,s) - I(k_0,t_0,s_0) + 1)r^2.$$

Combining the three cases, we obtain

$$\sum_{i=I(k_0,t_0,s_0)}^{I(k,t,s)} \|v_i - \nabla f(x_i)\|^2$$

$$\leq 48c^2 \left(\frac{KL^2}{b} + K^2\zeta^2 + \frac{KL^2T}{Pb}\right)$$

$$\times \left(\sum_{i=I(k_0,t_0,s_0)}^{I(k,t,s)-1} \|x_{i+1} - x_i\|^2 + \frac{(I(k,t,s) - I(k_0,t_0,s_0) + 1) \wedge K}{K} \sum_{i=I(0,t_0,s_0)}^{I(k_0,t_0,s_0)-1} \|x_{i+1} - x_i\|^2\right.$$

$$\left. + \frac{(I(k,t,s) - I(k_0,t_0,s_0) + 1) \wedge KT}{KT} \sum_{i=I(0,0,s_0)}^{I(0,t_0,s_0)-1} \|x_{i+1} - x_i\|^2\right)$$

$$\times \left(\log\frac{2KTSd}{q} + \log\log\frac{4KTG}{\widetilde{\varepsilon}}\right)$$

$$+ (I(k,t,s) - I(k_0,t_0,s_0) + 1)r^2.$$

Combining this bound with (2), we obtain

$$f(x_{I(k,t,s)})$$

$$\leq f(x_{I(k_0,t_0,s_0)}) - \frac{\eta}{2}\sum_{i=I(k_0,t_0,s_0)}^{I(k,t,s)-1} \|\nabla f(x_i)\|^2$$

$$- \left(\frac{1}{2\eta} - \frac{L}{2} - 48c^2\eta\left(\frac{KL^2}{b} + K^2\zeta^2 + \frac{KTL^2}{Pb}\right)\left(\log\frac{2KTSd}{q} + \log\log\frac{4KTG}{\widetilde{\varepsilon}}\right)\right)\sum_{i=I(k_0,t_0,s_0)}^{I(k,t,s)-1} \|x_{i+1} - x_i\|^2$$

$$+ \left\{48c^2\eta\left(\frac{KL^2}{b} + K^2\zeta^2 + \frac{KTL^2}{Pb}\right)\left(\log\frac{2KTSd}{q} + \log\log\frac{4KTG}{\widetilde{\varepsilon}}\right)\right.$$

$$\times \left(\frac{(I(k,t,s) - I(k_0,t_0,s_0)) \wedge K}{K} \sum_{i=I(0,t_0,s_0)}^{I(k_0,t_0,s_0)-1} \|x_{i+1} - x_i\|^2\right.$$

$$\left.\left. + \frac{(I(k,t,s) - I(k_0,t_0,s_0)) \wedge KT}{KT} \sum_{i=I(0,0,s_0)}^{I(0,t_0,s_0)-1} \|x_{i+1} - x_i\|^2\right)\right\}$$

$$+ \eta(I(k,t,s) - I(k_0,t_0,s_0))r^2$$

with probability at least $1 - 3q$.

We can choose $\eta = \widetilde{\Theta}(1/L \wedge \sqrt{b/K}/L \wedge 1/(K\zeta) \wedge \sqrt{Pb}/(\sqrt{KT}L))$ such that $\eta \leq 1/(8L)$ and

$$48c^2\eta\left(\frac{KL^2}{b} + K^2\zeta^2 + \frac{KTL^2}{Pb}\right)\left(\log\frac{2KTSd}{q} + \log\log\frac{4KTG}{\widetilde{\varepsilon}}\right) \leq \frac{c_\eta}{\eta}.$$

for some constant $c_\eta \in (0, 1/4)$. Then, the above result can be simplified as

$$f(x_{I(k,t,s)}) \le f(x_{I(k_0,t_0,s_0)}) - \frac{\eta}{2} \sum_{i=I(k_0,t_0,s_0)}^{I(k,t,s)-1} \|\nabla f(x_i)\|^2$$

$$- \frac{1}{8\eta} \sum_{i=I(k_0,t_0,s_0)}^{I(k,t,s)-1} \|x_{i+1} - x_i\|^2$$

$$+ \frac{c_\eta}{\eta} \left( \frac{(I(k,t,s) - I(k_0,t_0,s_0)) \wedge K}{K} \sum_{i=I(0,t_0,s_0)}^{I(k_0,t_0,s_0)-1} \|x_{i+1} - x_i\|^2 \right.$$

$$+ \frac{(I(k,t,s) - I(k_0,t_0,s_0)) \wedge KT}{KT} \sum_{i=I(0,0,s_0)}^{I(0,t_0,s_0)-1} \|x_{i+1} - x_i\|^2 \bigg)$$

$$+ \eta(I(k,t,s) - I(k_0,t_0,s_0))r^2 \tag{4}$$

with probability at least $1 - 3q$. $\qquad\square$

Also, we bound $\|x_{I(k,t,s)} - x_{I(k_0,t_0,s_0)}\|^2$. Note that

$$\|x_{I(k,t,s)} - x_{I(k_0,t_0,s_0)}\|^2 \le (I(k,t,s) - I(k_0,t_0,s_0)) \sum_{i=I(k_0,t_0,s_0)}^{I(k,t,s)-1} \|x_{i+1} - x_i\|^2$$

$$\le 8\eta(I(k,t,s) - I(k_0,t_0,s_0)) \bigg\{ f(x_{I(k_0,t_0,s_0)}) - f(x_{I(k,t,s)})$$

$$- \frac{1}{8\eta} \sum_{i=I(k_0,t_0,s_0)}^{I(k-1,t,s)} \|x_{i+1} - x_i\|^2$$

$$+ \frac{c_\eta}{\eta} \left( \frac{(I(k,t,s) - I(k_0,t_0,s_0)) \wedge K}{K} \sum_{i=I(0,t_0,s_0)}^{I(k_0,t_0,s_0)-1} \|x_{i+1} - x_i\|^2 \right.$$

$$+ \frac{(I(k,t,s) - I(k_0,t_0,s_0)) \wedge KT}{KT} \sum_{i=I(0,0,s_0)}^{I(0,t_0,s_0)-1} \|x_{i+1} - x_i\|^2 \bigg)$$

$$+ \eta(I(k,t,s) - I(k_0,t_0,s_0))r^2 \bigg\}.$$

for every $k, k_0 \in [K-1]$, $t, t_0 \in [T-1]$ and $s, s_0 \in [S-1]$ ($I(k,t,s) \ge I(k_0,t_0,s_0)$) with probability at least $1 - 3q$.

By the way, we also derive (loose) almost sure bound as follows: From (1) and the fact that $\|v_{I(k,t,s)} - \nabla f(x_{I(k,t,s)})\|^2 \le 3(4KG^2 + 4KG^2 + 4TG^2) \le 36KTG^2$ almost surely, it holds that

$$f(x_{I(k,t,s)}) - f(x_{I(k_0,t_0,s_0)}) \le 36\eta KT(I(k,t,s) - I(k_0,t_0,s_0))G^2 + \eta(I(k,t,s) - I(k_0,t_0,s_0))r^2$$

$$\le 36\eta K^2 T^2 S G^2 + \eta KTS r^2$$

$$\le 36\eta K^2 T^2 S(G^2 + r^2) \tag{5}$$

almost surely.

**Proof of Corollary 4.2**

Using Proposition 4.1 with $I(k,t,s) = KTS$ and $I(k_0, t_0, s_0) = 0$, we have

$$f(x_{KTS}) \le f(x_0) - \frac{\eta}{2} \sum_{i=0}^{KTS-1} \|\nabla f(x_i)\|^2$$

$$- \frac{1}{8\eta} \sum_{i=0}^{KTS-1} \|x_{i+1} - x_i\|^2 + \eta KTSr^2.$$

Then, $-\|\nabla f(x_i)\|^2 \le -(1/2)\|\nabla f(\widetilde{x}_i)\|^2 + \|\nabla f(x_i) - f(\widetilde{x}_i)\|^2 \le -(1/2)\|\nabla f(\widetilde{x}_{I(k,t,s)})\|^2 + \eta^2 L^2 r^2 \le -(1/2)\|\nabla f(\widetilde{x}_i)\|^2 + r^2$ gives

$$f(x_{KTS}) \le f(x_0) - \frac{\eta}{4} \sum_{i=0}^{KTS-1} \|\nabla f(\widetilde{x}_i)\|^2$$

$$- \frac{1}{8\eta} \sum_{i=0}^{KTS-1} \|x_{i+1} - x_i\|^2 + 2\eta KTSr^2.$$

Choosing $c_r \le 1/4$ immediately leads the desired result. $\qquad\square$

## B.4   Escaping Saddle Points

Given $\{x_i\}_{i=0}^{KTS-1}$, we introduce the concept of coupling sequence [10]. Given $x_{I(k_0, t_0, s_0)}$, let $\{e_i\}_{i=1}^d$ be the normalized eigenvectors of $\nabla f(\widetilde{x}_{I(k_0, t_0, s_0)})$ associated with the eigenvalues $\lambda_1 < \cdots < \lambda_d$. We set $e_{\min} := e_1$ and $\lambda_{\min} := \lambda_1$. We assume that $\lambda := -\lambda_{\min} > \sqrt{\rho\varepsilon}$.

Then, for given $\hat{I} \ge I(k_0, t_0, s_0)$, we define coupling sequence $\{x_i'\}_{i=0}^{KTS-1}$ as follows: (1) $\langle \xi_{\widetilde{I}}', e_{\min} \rangle = -\langle \xi_{\widetilde{I}}, e_{\min} \rangle$; (2) $\langle \xi_{\widetilde{I}}', e_j \rangle = \langle \xi_{\widetilde{I}}, e_j \rangle$ for $j \in \{2, \ldots, d\}$; and (3) All the other randomness is completely same as the one of $\{x_i\}_{i=0}^{KTS-1}$. Let $r_0 := |\langle \xi_{\widetilde{I}}, e_{\min} \rangle|$. Note that $|\langle \xi_{\widetilde{I}} - \xi_{\widetilde{I}}', e_{\min} \rangle| = 2r_0$ and thus $\|\xi_{\widetilde{I}} - \xi_{\widetilde{I}}'\| = 2r_0$. Also, observe that $x_{\widetilde{I}+1} - x_{\widetilde{I}+1}' = \eta\langle \xi_{\widetilde{I}} - \xi_{\widetilde{I}}', e_{\min} \rangle e_{\min}$. We define $\widetilde{I}$ used in the definition of the coupling sequence as follows:

$$\widetilde{I} := \begin{cases} I(k_0, t_0, s_0), & (1/(\eta\lambda) \le \sqrt{K}) \\ I(k_0', t_0, s_0) - 1, & (\sqrt{K} < 1/(\eta\lambda) \le K) \\ I(0, t_0+1, s_0) - 1, & (K < 1/(\eta\lambda) \le KT) \\ I(0, 0, s_0+1) - 1. & (KT < 1/(\eta\lambda)) \end{cases}$$

Here, $k_0'$ is the minimum index $k$ that satisfies $k > k_0$ and $k \equiv 0 \pmod{\lceil\sqrt{K}\rceil}$. We can easily check that $\widetilde{I} - I(k_0, t_0, s_0) \le 1/(\eta\lambda)$.

Note that

$$\mathbb{P}\left(r_0 \ge \frac{qr}{2\sqrt{d}}\right) \ge 1 - q \tag{6}$$

from the arguments in Section A.2 of [8].

To prove Proposition 4.3, first note that the following result:

**Proposition B.3.** *Let $k_0 \in [K] \cup \{0\}$, $t_0 \in [T-1] \cup \{0\}$ and $s_0 \in [S-1] \cup \{0\}$. Fix any $\mathcal{J} \in \{1, \ldots, I(0, 0, S) - I(k_0, t_0, s_0)\}$ and $\mathcal{F} > 0$. Under the same conditions as Proposition 4.1, it holds that*

$$\min\left\{ f(x_{I(k_0, t_0, s_0)+\mathcal{J}}) - f(x_{I(k_0, t_0 s_0)}), f(x_{I(k_0, t_0, s_0)+\mathcal{J}}') - f(x_{I(k_0, t_0 s_0)}') \right\}$$

$$\le -\mathcal{F} + \frac{2c_\eta}{\eta}\left( \frac{\mathcal{J} \wedge K}{K} \sum_{i=I(0,t_0,s_0)}^{I(k_0,t_0,s_0)-1} \|x_{i+1} - x_i\|^2 + \frac{\mathcal{J} \wedge KT}{KT} \sum_{i=I(0,0,s_0)}^{I(0,t_0,s_0)-1} \|x_{i+1} - x_i\|^2 \right)$$

*or $\forall J \in [\mathcal{J}] : \max\left\{ \|x_{I(k_0, t_0, s_0)+J} - x_{I(k_0, t_0, s_0)}\|^2, \|x_{I(k_0, t_0, s_0)+J}' - x_{I(k_0, t_0, s_0)}'\|^2 \right\}$*

$$\le 8\eta\mathcal{J}\left(\mathcal{F} + 2\eta\mathcal{J}r^2\right)$$

*with probability at least* $1 - 6q$.

*Proof.* First note that $x_i = x'_i$ for $i \leq I(k_0, t_0, s_0)$. From the bounds of $\|x_{I(k_0,t_0,s_0)+J} - x_{I(k_0,t_0,s_0)}\|^2$ and $\|x'_{I(k_0,t_0,s_0)+J} - x'_{I(k_0,t_0,s_0)}\|^2$, we can see that

$$\max\left\{ \|x_{I(k_0,t_0,s_0)+J} - x_{I(k_0,t_0,s_0)}\|^2, \|x'_{I(k_0,t_0,s_0)+J} - x'_{I(k_0,t_0,s_0)}\|^2 \right\}$$

$$\leq 8\eta J \left( \max\left\{ f(x_{I(k_0,t_0,s_0)}) - f(x_{I(k_0,t_0 s_0)+J}) - \frac{1}{8\eta} \sum_{i=I(k_0,t_0,s_0)}^{I(k_0,t_0,s_0)+J-1} \|x_{i+1} - x_i\|^2, \right.\right.$$

$$\left. f(x'_{I(k_0,t_0,s_0)}) - f(x'_{I(k_0,t_0 s_0)+J}) - \frac{1}{8\eta} \sum_{i=I(k_0,t_0,s_0)}^{I(k_0,t_0,s_0)+J-1} \|x'_{i+1} - x'_i\|^2 \right\}$$

$$\left. + \frac{c_\eta}{\eta} \left( \frac{J \wedge K}{K} \sum_{i=I(0,t_0,s_0)}^{I(k_0,t_0,s_0)-1} \|x_{i+1} - x_i\|^2 + \frac{J \wedge KT}{KT} \sum_{i=I(0,0,s_0)}^{I(0,t_0,s_0)-1} \|x_{i+1} - x_i\|^2 \right) + \eta J r^2 \right)$$

for every $J \in [\mathcal{J}]$ with probability at least $1 - 6q$.

We define $I(k_J, t_J, s_J) := I(k_0, t_0, s_0) + J$. Note that $s_J \geq s_0$. From (4),

$$f(x_{I(k_0,t_0,s_0)}) - f(x_{I(k_0,t_0,s_0)+J}) - \frac{1}{8\eta} \sum_{i=I(k_0,t_0,s_0)}^{I(k_0,t_0,s_0)+J-1} \|x_{i+1} - x_i\|^2$$

$$= f(x_{I(k_0,t_0,s_0)}) - f(x_{I(k_0,t_0,s_0)+\mathcal{J}})$$

$$+ f(x_{I(k_0,t_0,s_0)+\mathcal{J}}) - f(x_{I(k_0,t_0,s_0)+J}) - \frac{1}{8\eta} \sum_{i=I(k_0,t_0,s_0)}^{I(k_0,t_0,s_0)+J-1} \|x_{i+1} - x_i\|^2$$

$$\leq f(x_{I(k_0,t_0,s_0)}) - f(x_{I(k_0,t_0,s_0)+\mathcal{J}})$$

$$- \frac{1}{8\eta} \sum_{i=I(k_0,t_0,s_0)}^{I(k_0,t_0,s_0)+J-1} \|x_{i+1} - x_i\|^2$$

$$+ \frac{c_\eta}{\eta} \left( \frac{(\mathcal{J}-J) \wedge K}{K} \sum_{i=I(0,t_J,s_J)}^{I(k_J,t_J,s_J)-1} \|x_{i+1} - x_i\|^2 + \frac{(\mathcal{J}-J) \wedge KT}{KT} \sum_{i=I(0,0,s_J)}^{I(0,t_J,s_J)-1} \|x_{i+1} - x_i\|^2 \right)$$

$$+ \eta(\mathcal{J}-J)r^2$$

$$\leq f(x_{I(k_0,t_0,s_0)}) - f(x_{I(k_0,t_0,s_0)+\mathcal{J}})$$

$$+ \frac{c_\eta}{\eta} \left( \frac{\mathcal{J} \wedge K}{K} \sum_{i=I(0,t_0,s_0)}^{I(k_0,t_0,s_0)-1} \|x_{i+1} - x_i\|^2 + \frac{\mathcal{J} \wedge KT}{KT} \sum_{i=I(0,0,s_0)}^{I(0,t_0,s_0)-1} \|x_{i+1} - x_i\|^2 \right)$$

$$+ \eta \mathcal{J} r^2.$$

Here, for the last inequality, we used the fact that $I(0, t_J, s_J) \geq I(0, t_0, s_0)$. Also, we assumed $c_\eta \leq 1/8$.

Similarly, we can show that

$$f(x'_{I(k_0,t_0,s_0)}) - f(x'_{I(k_0,t_0,s_0)+J}) - \frac{1}{8\eta} \sum_{i=I(k_0,t_0,s_0)}^{I(k_0,t_0,s_0)+J-1} \|x'_{i+1} - x'_i\|^2$$

$$\leq f(x'_{I(k_0,t_0,s_0)}) - f(x'_{I(k_0,t_0,s_0)+\mathcal{J}})$$

$$+ \frac{c_\eta}{\eta} \left( \frac{\mathcal{J} \wedge K}{K} \sum_{i=I(0,t_0,s_0)}^{I(k_0,t_0,s_0)-1} \|x_{i+1} - x_i\|^2 + \frac{\mathcal{J} \wedge KT}{KT} \sum_{i=I(0,0,s_0)}^{I(0,t_0,s_0)-1} \|x_{i+1} - x_i\|^2 \right)$$

$$+ \eta \mathcal{J} r^2.$$

Therefore, we get

$$\max\left\{\|x_{I(k_0,t_0,s_0)+J} - x_{I(k_0,t_0,s_0)}\|^2, \|x'_{I(k_0,t_0,s_0)+J} - x'_{I(k_0,t_0,s_0)}\|^2\right\}$$

$$\leq 8\eta J\Bigg\{-\min\left\{f(x_{I(k_0,t_0,s_0)+\mathcal{J}}) - f(x_{I(k_0,t_0 s_0)}), f(x'_{I(k_0,t_0,s_0)+\mathcal{J}}) - f(x'_{I(k_0,t_0 s_0)})\right\}$$

$$+ \frac{2c_\eta}{\eta}\left(\frac{\mathcal{J} \wedge K}{K}\sum_{i=I(0,t_0,s_0)}^{I(k_0,t_0,s_0)-1}\|x_{i+1} - x_i\|^2 + \frac{\mathcal{J} \wedge KT}{KT}\sum_{i=I(0,0,s_0)}^{I(0,t_0,s_0)-1}\|x_{i+1} - x_i\|^2\right) + 2\eta\mathcal{J}r^2\Bigg\}$$

for every $J \in [\mathcal{J}]$. Now, suppose that

$$\min\left\{f(x_{I(k_0,t_0,s_0)+\mathcal{J}}) - f(x_{I(k_0,t_0 s_0)}), f(x'_{I(k_0,t_0,s_0)+\mathcal{J}}) - f(x'_{I(k_0,t_0 s_0)})\right\}$$

$$> -\mathcal{F} + \frac{2c_\eta}{\eta}\left(\frac{\mathcal{J} \wedge K}{K}\sum_{i=I(0,t_0,s_0)}^{I(k_0,t_0,s_0)-1}\|x_{i+1} - x_i\|^2 + \frac{\mathcal{J} \wedge KT}{KT}\sum_{i=I(0,0,s_0)}^{I(0,t_0,s_0)-1}\|x_{i+1} - x_i\|^2\right). \quad (7)$$

Then, using (7), we obtain

$$\max\left\{\|x_{I(k_0,t_0,s_0)+J} - x_{I(k_0,t_0,s_0)}\|^2, \|x'_{I(k_0,t_0,s_0)+J} - x'_{I(k_0,t_0,s_0)}\|^2\right\}$$

$$\leq 8\eta\mathcal{J}(\mathcal{F} + 2\eta\mathcal{J}r^2).$$

This finishes the proof. $\qquad\square$

We fix $k_0 \in [K-1]$, $t_0 \in [T-1]$, $s_0 \in [S-1]$ and $\mathcal{J}_{I(k_0,t_0,s_0)} \in \mathbb{N}$. Let $\mathcal{F}_{I(k_0,t_0,s_0)} := c_\mathcal{F}\eta\mathcal{J}_{I(k_0,t_0,s_0)}r^2$. From this definition, (4) immediately implies that

$$f(x_{I(k_0,t_0,s_0)+J}) - f(x_{I(k_0,t_0,s_0)})$$

$$\leq \frac{c_\eta}{\eta}\left(\frac{J \wedge K}{K}\sum_{i=I(0,t_0,s_0)}^{I(k_0,t_0,s_0)-1}\|x_{i+1} - x_i\|^2 + \frac{J \wedge KT}{KT}\sum_{i=I(0,0,s_0)}^{I(0,t_0,s_0)-1}\|x_{i+1} - x_i\|^2\right) + \eta\mathcal{J}r^2.$$

$$= \frac{2}{c_\mathcal{F}}\mathcal{F}_{I(k_0,t_0,s_0)} + \frac{c_\eta}{\eta}\left(\frac{\mathcal{J}_{I(k_0,t_0,s_0)} \wedge K}{K}\sum_{i=I(0,t_0,s_0)}^{I(k_0,t_0,s_0)-1}\|x_{i+1} - x_i\|^2 + \frac{\mathcal{J}_{I(k_0,t_0,s_0)} \wedge KT}{KT}\sum_{i=I(0,0,s_0)}^{I(0,t_0,s_0)-1}\|x_{i+1} - x_i\|^2\right)$$
$$\quad (8)$$

for every $J \in [\mathcal{J}_{I(k_0,t_0,s_0)}]$ with probability at least $1 - 3q$. Here, for simplifying the notations, we set $\mathcal{F} := \mathcal{F}_{I(k_0,t_0,s_0)}$ and $\mathcal{J} := \mathcal{J}_{I(k_0,t_0,s_0)}$.

We want to show the following proposition:

**Proposition B.4.** *Under the same conditions as Proposition 4.3, it holds that*

$$\max\left\{\|x_{I(k_0,t_0,s_0)+J} - x_{I(k_0,t_0,s_0)}\|^2, \|x'_{I(k_0,t_0,s_0)+J} - x'_{I(k_0,t_0,s_0)}\|^2\right\} > 8(c_\mathcal{F} + 2)\eta^2\mathcal{J}^2r^2$$

*for some $J \in [\mathcal{J}]$ with probability at least $1 - 3q$.*

*Proof.* We consider the event $H$ that is an intersection of (6), (14) and (15) (derived later), which holds probability at least $1 - 3q$. From now, the arguments are conditioned on $H$. Observe that $8\eta\mathcal{J}(\mathcal{F} + 2\eta\mathcal{J}r^2) = 8(c_\mathcal{F} + 2)\eta^2\mathcal{J}^2r^2$.

Suppose that

$$\max\left\{\|x_{I(k_0,t_0,s_0)+J} - x_{I(k_0,t_0,s_0)}\|^2, \|x'_{I(k_0,t_0,s_0)+J} - x'_{I(k_0,t_0,s_0)}\|^2\right\} \leq 8(c_\mathcal{F} + 2)\eta^2\mathcal{J}^2r^2,$$

which implies

$$\max\left\{\|x_{I(k_0,t_0,s_0)+J} - x_{I(k_0,t_0,s_0)}\|, \|x'_{I(k_0,t_0,s_0)+J} - x'_{I(k_0,t_0,s_0)}\|\right\} \leq 2\sqrt{2(c_\mathcal{F} + 2)}\eta\mathcal{J}r.$$

for every $J \in [\mathcal{J}]$. Then, we have

$$\max \left\{ \|x_{I(k_0,t_0,s_0)+J} - \widetilde{x}_{I(k_0,t_0,s_0)}\|, \|x'_{I(k_0,t_0,s_0)+J} - \widetilde{x}_{I(k_0,t_0,s_0)}\| \right\}$$
$$\leq \max \left\{ \|x_{I(k_0,t_0,s_0)+J} - x_{I(k_0,t_0,s_0)}\|, \|x'_{I(k_0,t_0,s_0)+J} - x'_{I(k_0,t_0,s_0)}\| \right\} + \eta r$$
$$\leq 4\sqrt{c_{\mathcal{F}} + 2}\eta \mathcal{J}r =: U_\Delta.$$

We will derive a contradiction. Now, we consider the quantity $\|x_i - x'_i\|^2$ for $i > \widetilde{I}$. $w_i$ denotes $x_i - x'_i$. Since $\xi_i = \xi'_i$ for $i \neq \hat{I}$, for $I \geq \widetilde{I}$, we have that

$$\begin{aligned}
w_{I+1} &= x_{I+1} - x'_{I+1} \\
&= w_I - \eta(v_I - v'_I) - \eta(\xi_I - \xi'_I) \\
&= w_I - \eta(\nabla f(x_I) - \nabla f(x'_I) + v_I - \nabla f(x_I) - v'_I + \nabla f(x'_I)) \\
&= w_I - \eta((\mathcal{H} + \Delta_I)w_I + v_I - \nabla f(x_I) - v'_I + \nabla f(x'_I)) \\
&= (1 - \eta\mathcal{H})w_I - \eta(\Delta_I w_I + y_I) \\
&= \eta(1 - \eta\mathcal{H})^{I-\widetilde{I}}\hat{\xi}_{\widetilde{I}} - \eta \sum_{i=\widetilde{I}}^{I}(1 - \eta\mathcal{H})^{I-i}(\Delta_i w_i + y_i),
\end{aligned}$$

where $\mathcal{H} := \nabla^2 f(\widetilde{x}_{I(k_0,t_0,s_0)})$, $\Delta_i := \int_0^1 (\nabla^2 f(\theta x_i + (1-\theta)x'_i) - \mathcal{H})d\theta$, $y_i := v_i - \nabla f(x_i) - v'_i + \nabla f(x'_i)$ and $\hat{\xi}_i = \xi_i - \xi'_i$. Let $\lambda := -\lambda_{\min}(\nabla^2 f(\widetilde{x}_{I(k_0,t_0,s_0)})) > \sqrt{\rho\varepsilon}$. For the last inequality, we used $\widetilde{x}_{\widetilde{I}} = \widetilde{x}'_{\widetilde{I}}$.

First we give an upper bound of the term $\|\eta(1-\eta\mathcal{H})^{I-\widetilde{I}}\hat{\xi}_{\widetilde{I}}\|$. Since $\hat{\xi}_{\widetilde{I}} = \xi_{\widetilde{I}} - \xi'_{\widetilde{I}} = 2\langle \xi_{\widetilde{I}}, e_{\min}\rangle e_{\min}$, we have

$$\eta(1-\eta\mathcal{H})^{I-\widetilde{I}}\hat{\xi}_{\widetilde{I}} = 2\eta(1+\eta\lambda)^{I-\widetilde{I}}\langle \xi_{\widetilde{I}}, e_{\min}\rangle e_{\min}.$$

Since $r_0 = 2|\langle \xi_{\widetilde{I}}, e_{\min}\rangle|$, we have

$$\left\| \eta(1-\eta\mathcal{H})^{I-\widetilde{I}}\hat{\xi}_{\widetilde{I}} \right\| = \eta(1+\eta\lambda)^{I-\widetilde{I}}r_0 =: U_{\hat{\xi}}(I). \tag{9}$$

From now, we will show that the following claims hold for $I \in \{0, \ldots, I(k_0, t_0, s_0) + \mathcal{J}\}$ with probability at least $1 - q$ using mathematical induction:

$$\|w_I\| \leq c_{\text{upper}}^{(w)} \cdot \eta(1+\eta\lambda)^{I-\widetilde{I}}r_0 =: U_w(I)$$

for $c_{\text{upper}}^{(w)} = \widetilde{\Theta}(1) > 0$, and

$$\|y_I\| \leq c_{\text{upper}}^{(y)} \cdot \eta^2\lambda \left( L + \frac{\sqrt{K}L}{\sqrt{b}} + K\zeta + \frac{\sqrt{KT}L}{\sqrt{Pb}} \right)(1+\eta\lambda)^{I-\widetilde{I}}r_0 =: U_y(I)$$

for some $c_{\text{upper}}^{(y)} = \widetilde{\Theta}(1) > 0$. Observe that $U_\xi$, $U_w$ and $U_y$ are monotonically increasing with respect to $I$ for $I \geq \widetilde{I}$). First we check the case $I \in \{0, \ldots, \widetilde{I}\}$. In this case, the both claims trivially holds from the definition of $\{x'_i\}_{i=0}^{KTS-1}$ because $w_i = y_i = 0$ for $i \leq \widetilde{I}$. Suppose that the two claims hold for the cases $\{0, \ldots, I\}$ with $I \geq \widetilde{I}$. We want to show that the two claims also hold for the case $I + 1 > \widetilde{I}$.

$$\|w_{I+1}\| \leq \eta \sum_{i=\widetilde{I}}^{I}(1+\eta\lambda)^{I-i}\|\Delta_i w_i\| + \eta \sum_{i=\widetilde{I}}^{I}(1+\eta\lambda)^{I-i}\|y_i\| + U_{\hat{\xi}}(I+1).$$

Here we used inequality (9). Observe that

$$\begin{aligned}
\|\Delta_i w_i\| &\leq \|\Delta_i\|\|w_i\| \\
&\leq \|\Delta_i\|U_w(i) \\
&\leq \|\Delta_i\|U_w(I+1)
\end{aligned}$$

and

$$\|\Delta_i\| \leq \rho \int_0^1 \|\theta x_i + (1-\theta)x_i' - \widetilde{x}_{I(k_0,t_0,s_0)}\| d\theta$$
$$\leq \rho \max\{\|x_i - \widetilde{x}_{I(k_0,t_0,s_0)}\|, \|x_i' - \widetilde{x}'_{I(k_0,t_0,s_0)}\|\}$$
$$\leq \rho U_\Delta.$$

Hence, we get

$$\eta \sum_{i=\widetilde{I}}^{I}(1+\eta\lambda)^{I-i}\|\Delta_i w_i\|$$
$$\leq \eta(I-\widetilde{I})\rho U_\Delta U_w(I+1)$$
$$\leq \eta\rho\mathcal{J} U_\Delta U_w(I+1) \tag{10}$$

Similarly, from the inductive assumption on $\|y_i\|$,

$$\eta \sum_{i=\widetilde{I}}^{I}(1+\eta\lambda)^{I-i}\|y_i\|$$
$$\leq c_{\text{upper}}^{(y)}\eta \cdot \eta\lambda\mathcal{J} \cdot \eta\left(L + \frac{\sqrt{K}L}{\sqrt{b}} + K\zeta + \frac{\sqrt{KT}L}{\sqrt{Pb}}\right)(1+\eta\lambda)^{I-\widetilde{I}}r_0$$
$$\leq \left(\frac{c_{\text{upper}}^{(y)}\eta\lambda\mathcal{J}\left(L + \frac{\sqrt{K}L}{\sqrt{b}} + K\zeta + \frac{\sqrt{KT}L}{\sqrt{Pb}}\right)}{c_{\text{upper}}^{(w)}}\right)U_w(I). \tag{11}$$

These results imply

$$\|w_{I+1}\| \leq \eta\rho\mathcal{J} U_\Delta U_w(I+1) + \left(\frac{c_{\text{upper}}^{(y)}\eta\lambda\mathcal{J} \cdot \eta\left(L + \frac{\sqrt{K}L}{\sqrt{b}} + K\zeta + \frac{\sqrt{KT}L}{\sqrt{Pb}}\right)}{c_{\text{upper}}^{(w)}}\right)U_w(I) + U_{\hat{\xi}}(I+1)$$
$$\leq \left(\eta\rho\mathcal{J} U_\Delta + \frac{c_{\text{upper}}^{(y)}\eta\lambda\mathcal{J} \cdot \eta\left(L + \frac{\sqrt{K}L}{\sqrt{b}} + K\zeta + \frac{\sqrt{KT}L}{\sqrt{Pb}}\right)}{c_{\text{upper}}^{(w)}} + \frac{1}{c_{\text{upper}}^{(w)}}\right)U_w(I+1).$$

Here, we again used the monotonicity of $U_w(i)$ with respect to $i$. Now, we define $\mathcal{J} := \mathcal{J}_{I(k_0,t_0,s_0)} := c_{\mathcal{J}}/(\eta\lambda)\ (\leq c_{\mathcal{J}}/(\eta\sqrt{\rho\varepsilon}))$ for some $c_{\mathcal{J}} = \widetilde{\Theta}(1) \geq 2$, which does not depend on index $I(k_0,t_0,s_0)$ and will be determined later. Also, we set $c_{\text{upper}}^{(w)} \geq 3$ and $c_{\text{upper}}^{(y)} := c_{\text{upper}}^{(w)}$. These definitions with appropriate $\eta \leq 1/(c_{\mathcal{J}}(L + \sqrt{K}L/\sqrt{b} + K\zeta + \sqrt{KT}L/\sqrt{Pb}))\} \times 1/(6c_{\text{upper}}^{(w)}) = \widetilde{\Theta}(1/L \wedge \sqrt{b/K}/L) \wedge 1/(K\zeta) \wedge \sqrt{Pb}/(\sqrt{KT}L)$ and $c_r \leq 1/(24\sqrt{c_{\mathcal{F}} + 2c_{\mathcal{J}}^2}c_{\text{upper}}^{(w)})$ give

$$\eta\rho\mathcal{J} U_\Delta \leq 4c_r\sqrt{c_{\mathcal{F}} + 4} \times \eta^2\mathcal{J}^2\rho\varepsilon \leq \frac{1}{6c_{\text{upper}}^{(w)}} \leq \frac{1}{18} \tag{12}$$

and

$$\frac{c_{\text{upper}}^{(y)}\eta\lambda\mathcal{J} \cdot \eta\left(L + \frac{\sqrt{K}L}{\sqrt{b}} + K\zeta + \frac{\sqrt{KT}L}{\sqrt{Pb}}\right)}{c_{\text{upper}}^{(w)}} \leq \frac{1}{6c_{\text{upper}}^{(w)}} \leq \frac{1}{18}. \tag{13}$$

Hence, we obtain

$$\|w_{I+1}\| \leq \frac{4}{9}U_w(I+1) \leq U_w(I+1).$$

Next, we consider the quantity $\|y_{I+1}\|$. Let $k,t,s$ be $I+1 = I(k,t,s)$. We define

$$\begin{cases}
\alpha_{I(\kappa,t,s)} := g_{I(\kappa,t,s)} - g_{I(\kappa,t,s)}^{\text{ref}} + \nabla f_{p_{t,s}}(x_{I(\kappa-1,t,s)}) - \nabla f_{p_{t,s}}(x_{I(\kappa,t,s)}), \\
\beta_{I(\kappa,t,s)} := \nabla f_{p_{t,s}}(x_{I(\kappa,t,s)}) - \nabla f_{p_{t,s}}(x_{I(\kappa-1,t,s)}) + \nabla f(x_{I(\kappa-1,t,s)}) - \nabla f(x_{I(\kappa,t,s)}), \\
\gamma_{I(0,\tau,s)} := \frac{1}{P}\sum_{p=1}^{P}(g_{I(0,\tau,s)}^{(p)} - g_{I(0,\tau,s)}^{(p),\text{ref}} + \nabla f(x_{I(0,\tau-1,s)}) - \nabla f(x_{I(0,\tau,s)}).
\end{cases}$$

Similarly, we define

$$
\begin{cases}
\alpha'_{I(\kappa,t,s)} := & g'_{I(\kappa,t,s)} - (g^{\mathrm{ref}}_{I(\kappa,t,s)})' + \nabla f_{p_{t,s}}(x'_{I(\kappa-1,t,s)}) - \nabla f_{p_{t,s}}(x'_{I(\kappa,t,s)}), \\
\beta'_{I(\kappa,t,s)} := & \nabla f_{p_{t,s}}(x'_{I(\kappa,t,s)}) - \nabla f_{p_{t,s}}(x'_{I(\kappa-1,t,s)}) + \nabla f(x'_{I(\kappa-1,t,s)}) - \nabla f(x'_{I(\kappa,t,s)}), \\
\gamma'_{I(0,\tau,s)} := & \frac{1}{P}\sum_{p=1}^{P}((g^{(p)}_{I(0,\tau,s)})' - (g^{(p),\mathrm{ref}}_{I(0,\tau,s)})' + \nabla f(x'_{I(0,\tau-1,s)}) - \nabla f(x'_{I(0,\tau,s)})
\end{cases}
$$

that are associated with sequence $\{x'_i\}_{i=I(k_0,t_0,s_0)}^{\infty}$. Let $\hat{\alpha}_{I(\kappa,t,s)} = \alpha_{I(\kappa,t,s)} - \alpha'_{I(\kappa,t,s)}$, $\hat{\beta}_{I(\kappa,t,s)} = \beta_{I(\kappa,t,s)} - \beta'_{I(\kappa,t,s)}$ and $\hat{\gamma}_{I(\kappa,t,s)} = \gamma_{I(\kappa,t,s)} - \gamma'_{I(\kappa,t,s)}$. Then we further define

$$
\begin{cases}
\hat{A}_{I(k,t,s)} := & \sum_{\kappa=0}^{k-1}\hat{\alpha}_{I(\kappa+1,t,s)}, \\
\hat{B}_{I(k,t,s)} := & \sum_{\kappa=0}^{k-1}\hat{\beta}_{I(\kappa+1,t,s)}, \\
\hat{C}_{I(0,t,s)} := & \sum_{\tau=0}^{t-1}\hat{\gamma}_{I(0,\tau+1,s)}
\end{cases}
$$

These definitions give

$$
\begin{aligned}
y_{I+1} &= v_{I+1} - \nabla f(x_{I+1}) - v'_{I+1} + \nabla f(x'_{I+1}) \\
&= \hat{A}_{I(k,t,s)} + \hat{B}_{I(k,t,s)} + \hat{C}_{I(0,t,s)} \\
&\quad + v_{I(0,0,s)} - \nabla f(x_{I(0,0,s)}) - v'_{I(0,0,s)} + \nabla f(x'_{I(0,0,s)})
\end{aligned}
$$

This implies

$$
\begin{aligned}
\|y_{I+1}\| &= \|v_{I+1} - \nabla f(x_{I+1}) - v'_{I+1} + \nabla f(x'_{I+1})\| \\
&\leq \left\|\hat{A}_{I(k,t,s)}\right\| + \left\|\hat{B}_{I(k,t,s)}\right\| + \left\|\hat{C}_{I(0,t,s)}\right\|.
\end{aligned}
$$

Here, we used the fact that $v_{I(0,0,s)} - \nabla f(x_{I(0,0,s)}) - v'_{I(0,0,s)} + \nabla f(x'_{I(0,0,s)}) = 0$.

**Bounding $\|\hat{A}_{I(k,t,s)}\|$**

Observe that $\hat{\alpha}_I(\kappa,t,s)$ satisfies

$$
\mathbb{E}[\hat{\alpha}_I(\kappa,t,s) \mid \mathcal{F}_{I(\kappa-1,t,s)}] = 0.
$$

Let

$$
\begin{aligned}
&\hat{u}^{(\alpha)}_{l,I(\kappa,t,s)} \\
&:= \nabla\ell(x_{I(\kappa,t,s)}, z_{l,I(\kappa,t,s)}) - \nabla\ell(x'_{I(\kappa,t,s)}, z_{l,I(\kappa,t,s)}) - (\nabla\ell(x_{I(\kappa-1,t,s)}, z_{l,I(\kappa,t,s)}) - \nabla\ell(x'_{I(\kappa-1,t,s)}, z_{l,I(\kappa,t,s)})) \\
&\quad + (\nabla f_{p_{t,s}}(x_{I(\kappa-1,t,s)}) - \nabla f_{p_{t,s}}(x'_{I(\kappa-1,t,s)})) - (\nabla f_{p_{t,s}}(x_{I(\kappa,t,s)}) - \nabla f_{p_{t,s}}(x'_{I(\kappa,t,s)})).
\end{aligned}
$$

Note that $\mathbb{E}[\hat{u}_{l,I(\kappa,t,s)}^{(\alpha)}|\mathcal{F}_{I(\kappa,t,s)-1}] = 0$ and $\hat{\alpha}_{I(\kappa,t,s)} = (1/b)\sum_{l=1}^{b'} \hat{u}_{l,I(\kappa,t,s)}^{(\alpha)}$. Observe that

$$
\begin{aligned}
&\|\hat{u}_{l,I(\kappa,t,s)}^{(\alpha)}\| \\
&= \|\nabla\ell(x_{I(\kappa,t,s)}, z_{l,I(\kappa,t,s)}) - \nabla\ell(x'_{I(\kappa,t,s)}, z_{l,I(\kappa,t,s)}) - (\nabla\ell(x_{I(\kappa-1,t,s)}, z_{l,I(\kappa,t,s)}) - \nabla\ell(x'_{I(\kappa-1,t,s)}, z_{l,I(\kappa,t,s)})) \\
&\quad + (\nabla f_{p_{t,s}}(x_{I(\kappa-1,t,s)}) - \nabla f_{p_{t,s}}(x'_{I(\kappa-1,t,s)})) - (\nabla f_{p_{t,s}}(x_{I(\kappa,t,s)}) - \nabla f_{p_{t,s}}(x'_{I(\kappa,t,s)}))\| \\
&= \Big\| \int_0^1 \nabla^2\ell(\theta x_{I(\kappa,t,s)} + (1-\theta)x'_{I(\kappa,t,s)}, z_{l,I(\kappa,t,s)})d\theta(x_{I(\kappa,t,s)} - x'_{I(\kappa,t,s)}) \\
&\quad - \int_0^1 \nabla^2\ell(\theta x_{I(\kappa-1,t,s)} + (1-\theta)x'_{I(\kappa-1,t,s)}, z_{l,I(\kappa,t,s)})d\theta(x_{I(\kappa-1,t,s)} - x'_{I(\kappa-1,t,s)}) \\
&\quad + \int_0^1 \nabla^2 f_{p_{t,s}}(\theta x_{I(\kappa,t,s)} + (1-\theta)x'_{I(\kappa-1,t,s)})d\theta(x_{I(\kappa,t,s)} - x'_{I(\kappa,t,s)}) \\
&\quad - \int_0^1 \nabla^2 f_{p_{t,s}}(\theta x_{I(\kappa-1,t,s)} + (1-\theta)x'_{I(\kappa-1,t,s)})d\theta(x_{I(\kappa-1,t,s)} - x'_{I(\kappa-1,t,s)}) \Big\| \\
&= \|\mathcal{H}_{z_{l,I(\kappa,t,s)}} w_{I(\kappa,t,s)} + \Delta_{z_{l,I(\kappa,t,s)},I(\kappa,t,s)} w_{I(\kappa,t,s)} - (\mathcal{H}_{z_{l,I(\kappa,t,s)}} w_{I(\kappa-1,t,s)} + \Delta_{z_{l,I(\kappa,t,s)},I(\kappa-1,t,s)} w_{I(\kappa-1,t,s)}) \\
&\quad + \mathcal{H}_{p_{t,s}} w_{I(\kappa,t,s)} + \Delta_{p_{t,s},I(\kappa,t,s)} w_{I(\kappa,t,s)} - (\mathcal{H}_{p_{t,s}} w_{I(\kappa-1,t,s)} + \Delta_{p_{t,s},I(\kappa-1,t,s)} w_{I(\kappa-1,t,s)})\| \\
&\le \|(\mathcal{H}_{z_{l,I(\kappa,t,s)}} - \mathcal{H}_{p_{t,s}})(w_{I(\kappa,t,s)} - w_{I(\kappa-1,t,s)})\| \\
&\quad + \|(\Delta_{I(\kappa,t,s),z_{l,I(\kappa,t,s)}} - \Delta_{I(\kappa,t,s)})w_{I(\kappa,t,s)}\| + \|(\Delta_{I(\kappa-1,t,s),z_{l,I(\kappa,t,s)}} - \Delta_{I(\kappa-1,t,s)})w_{I(\kappa-1,t,s)}\| \\
&\le 2L\|w_{I(\kappa,t,s)} - w_{I(\kappa-1,t,s)}\| \\
&\quad + 2\rho\max\{\|x_{I(\kappa,t,s)} - \widetilde{x}_{I(k_0,t_0,s_0)}\|, \|x'_{I(\kappa,t,s)} - \widetilde{x}_{I(k_0,t_0,s_0)}\|, \\
&\qquad\quad \|x_{I(\kappa-1,t,s)} - \widetilde{x}_{I(k_0,t_0,s_0)}\|, \|x'_{I(\kappa-1,t,s)} - \widetilde{x}_{I(k_0,t_0,s_0)}\|\}(\|w_{I(\kappa,t,s)}\| + \|w_{I(\kappa-1,t,s)}\|) \\
&\le 2L\|w_{I(\kappa,t,s)} - w_{I(\kappa-1,t,s)}\| + 4\rho U_\Delta U_w(I+1).
\end{aligned}
$$

Here, $\mathcal{H}_z := \nabla^2\ell(\widetilde{x}_{I(k_0,t_0,s_0)}, z)$, $\mathcal{H}_{p_{t,s}} := \nabla^2 f_{p_{t,s}}(\widetilde{x}_{I(k_0,t_0,s_0)})$, $\Delta_{z,I(\kappa,t,s)} := \int_0^1 (\nabla^2\ell(\theta x_{I(\kappa,t,s)} + (1-\theta)x'_{I(\kappa,t,s)}, z) - \mathcal{H}_z)d\theta$ and $\Delta_{p_{t,s},I(\kappa,t,s)} := \int_0^1 (\nabla^2 f_{p_{t,s}}(\theta x_{I(\kappa,t,s)} + (1-\theta)x'_{I(\kappa,t,s)}) - \mathcal{H}_{p_{t,s}})d\theta$.
We define

$$
\hat{\sigma}_{I(\kappa,t,s)}^{(\alpha)} := 2L\|w_{I(\kappa,t,s)} - w_{I(\kappa-1,t,s)}\| + 4\rho U_\Delta U_w(I+1).
$$

Here, for the last inequality, we used the inductive assumption on $\|w_{I(\kappa,t,s)}\|$ for $I(\kappa,t,s) \le I(k-1,t,s)$ and the proven bound for $\|w_{I(k,t,s)}\|$. Also, we used the simple fact that $(1+\eta\lambda)^{I(\kappa,t,s)-I(k_0,t_0,s_0)} \le (1+\eta\lambda)^{I+1-I(k_0,t_0,s_0)}$ Hence, we have

$$
\mathbb{P}(\|\hat{u}_{l,I(\kappa,t,s)}^{(\alpha)}\| \ge s \mid \mathfrak{F}_{I(\kappa-1,t,s)}) \le 2e^{-\frac{s^2}{2\left(\hat{\sigma}_{I(\kappa,t,s)}^{(\alpha)}\right)^2}}
$$

for every $s \in \mathbb{R}$ and $\kappa \in [k]$. Also note that $\{\hat{u}_{l,I(\kappa,t,s)}^{(\alpha)}\}_{l=1}^{b_\kappa}$ is i.i.d. sequence conditioned on $\mathfrak{F}_{I(\kappa-1,t,s)}$. Also note that $\|\hat{\alpha}_{I(\kappa,t,s)}\| \le 8G$ almost surely from Assumption 5. From these results, we can use Lemma B.2 with $A = 8kG$ and $a = \widetilde{\varepsilon}'$ ($\widetilde{\varepsilon}'$ is some positive number and will be defined later) and get

$$
\left\|\hat{A}_{I(k,t,s)}\right\| \le c\sqrt{\left(\left(\sum_{\kappa=0}^{k-1} \frac{1}{b_{\kappa+1}}\left(\hat{\sigma}_{I(\kappa+1,t,s)}^{(\alpha)}\right)^2\right) + \widetilde{\varepsilon}'\right)\left(\log\frac{2KTSd}{q} + \log\log\frac{8KG}{\widetilde{\varepsilon}'}\right)} \quad (14)
$$

for every $k \in [K] \cup \{0\}$, $t \in [T-1] \cup \{0\}$ and $s \in [S] \cup \{0\}$ with probability at least $1-q$ for some constant $c > 0$. Note that this event always holds under $H$.

**Bounding** $\|\hat{B}_{I(k,t,s)}\|$

Observe that

$$
\begin{aligned}
\hat{B}_{I(k,t,s)} &= \nabla f_{p_{t,s}}(x_{I(k,t,s)}) - \nabla f_{p_{t,s}}(x_{I(0,t,s)}) + \nabla f(x_{I(0,t,s)}) - \nabla f(x_{I(k,t,s)}) \\
&\quad + \nabla f_{p_{t,s}}(x'_{I(k,t,s)}) - \nabla f_{p_{t,s}}(x'_{I(0,t,s)}) + \nabla f(x'_{I(0,t,s)}) - \nabla f(x'_{I(k,t,s)}) \\
&= \int_0^1 \nabla^2 f_{p_{t,s}}(\theta x_{I(k,t,s)} + (1-\theta)x'_{I(k,t,s)})d\theta(x_{I(k,t,s)} - x'_{I(k,t,s)}) \\
&\quad - \int_0^1 \nabla^2 f_{p_{t,s}}(\theta x_{I(0,t,s)} + (1-\theta)x'_{I(0,t,s)})d\theta(x_{I(0,t,s)} - x'_{I(0,t,s)}) \\
&\quad + \int_0^1 \nabla^2 f(\theta x_{I(k,t,s)} + (1-\theta)x'_{I(k,t,s)})d\theta(x_{I(k,t,s)} - x'_{I(k,t,s)}) \\
&\quad - \int_0^1 \nabla^2 f(\theta x_{I(0,t,s)} + (1-\theta)x'_{I(0,t,s)})d\theta(x_{I(0,t,s)} - x'_{I(0,t,s)}) \\
&= (\mathcal{H}_{p_{t,s}} + \Delta_{p_{t,s},I(\kappa,t,s)})w_{I(k,t,s)} - (\mathcal{H}_{p_{t,s}} + \Delta_{p_{t,s},I(0,t,s)})w_{I(0,t,s)} \\
&\quad + (\mathcal{H} + \Delta_{I(0,t,s)})w_{I(0,t,s)} - (\mathcal{H} + \Delta_{I(k,t,s)})w_{I(k,t,s)} \\
&= (\mathcal{H}_{p_{t,s}} - \mathcal{H})(w_{I(k,t,s)} - w_{I(0,t,s)}) \\
&\quad + (\Delta_{I(k,t,s),p_{t,s}} - \Delta_{I(k,t,s)})w_{I(k,t,s)} - (\Delta_{I(0,t,s),p_{t,s}} - \Delta_{I(0,t,s)})w_{I(0,t,s)}.
\end{aligned}
$$

This implies that

$$
\begin{aligned}
\left\|\hat{B}_{I(k,t,s)}\right\| &\leq \zeta\|w_{I(k,t,s)} - w_{I(0,t,s)}\| + 4\rho U_\Delta U_w(I+1) \\
&\leq \zeta \sum_{\kappa=0}^{k-1} \|w_{I(\kappa+1,t,s)} - w_{I(\kappa,t,s)}\| + 4\rho U_\Delta U_w(I+1).
\end{aligned}
$$

**Bounding** $\|\hat{C}_{I(0,t,s)}\|$

The argument is similar to the case of $\|\hat{A}_{I(k,t,s)}\|$. From Lemma B.2, the third term $\|\hat{C}_{I(0,t,s)}\|$ can be bounded as

$$
\left\|\hat{C}_{I(0,t,s)}\right\| \leq \frac{c}{\sqrt{PKb}} \sqrt{\left(\left(\sum_{\tau=0}^{t-1}\left(\hat{\sigma}_{I(0,\tau+1,s)}^{(\gamma)}\right)^2\right) + \widetilde{\varepsilon}'\right)\left(\log\frac{2KTSd}{q} + \log\log\frac{8TG}{\widetilde{\varepsilon}'}\right)} \quad (15)
$$

for every $t \in [T-1] \cup \{0\}$ and $s \in [S-1] \cup \{0\}$ with probability at least $1 - q$, where

$$
\sigma_{I(0,\tau,s)}^{(\gamma)} := 2L\|w_{I(0,\tau,s)} - w_{I(0,\tau-1,s)}\| + 4\rho U_\Delta U_w(I+1).
$$

Here, we used the facts that $\{g_{I(0,\tau,s)}^{(p)} - g_{I(0,\tau,s)}^{(p)\mathrm{ref}} + \nabla f_p(x_{I(0,\tau-1,s)}) - \nabla f_p(x_{I(0,\tau,s)})\}_{p=1}^P$ has mean zero and each of them is constructed from $Kb$ i.i.d. data samples, and $\{(g_{I(0,\tau,s)}^{(p)})' - (g_{I(0,\tau,s)}^{(p)\mathrm{ref}})' + \nabla f_p(x'_{I(0,\tau-1,s)}) - \nabla f_p(x'_{I(0,\tau,s)})\}_{p=1}^P$ possesses the same property.

Hence, we have

$$\|y_{I+1}\|$$
$$= \|v_{I+1} - \nabla f(x_{I+1}) - v'_{I+1} + \nabla f(x'_{I+1})\|$$
$$\leq \left\|\hat{A}_{I(k,t,s)}\right\| + \left\|\hat{B}_{I(k,t,s)}\right\| + \left\|\hat{C}_{I(0,t,s)}\right\|$$
$$\leq \left\{ c\sqrt{8L^2 \sum_{\kappa=0}^{k-1} \frac{1}{b_{\kappa+1}} \|w_{I(\kappa+1,t,s)} - w_{I(\kappa,t,s)}\|^2 + 32K\rho^2 U_\Delta^2 U_w (I+1)^2 + \widetilde{\varepsilon}'} \right.$$
$$+ \zeta \sum_{\kappa=0}^{k-1} \|w_{I(\kappa+1,t,s)} - w_{I(\kappa,t,s)}\| + 4\rho U_\Delta U_w (I+1)$$
$$\left. + \frac{c}{\sqrt{PKb}} \sqrt{8L^2 \sum_{\tau=0}^{t-1} \|w_{I(0,\tau+1,s)} - w_{I(0,\tau,s)}\|^2 + 32T\rho^2 U_\Delta^2 U_w (I+1)^2 + \widetilde{\varepsilon}'} \right\}$$
$$\times \sqrt{\log\frac{2KTSd}{q} + \log\log\frac{8KTG}{\widetilde{\varepsilon}'}}$$

Now, we further bound the term $\|w_{I(\kappa+1,\tau,s)} - w_{I(\kappa,\tau,s)}\|$.

To do this, it is important to carefully distinguish the three cases: $I(\kappa+1,\tau,s) = \widetilde{I}+1$, $I(\kappa+1,\tau,s) < \widetilde{I}+1$ and $I(\kappa+1,\tau,s) > \widetilde{I}+1$.

For the former case, note that $\|w_{\widetilde{I}+1} - w_{\widetilde{I}}\| = \|w_{\widetilde{I}+1}\| = \eta r_0$. Also note that $\|w_{I(\kappa+1,\tau,s)} - w_{I(\kappa,\tau,s)}\| = 0$ for $I(\kappa+1,\tau,s) < \widetilde{I}+1$.

**Case I.** $1/(\eta\lambda) \leq \sqrt{K}$.

In this case, $\widetilde{I} = I(k_0, t_0, s_0)$. Suppose that $s = s_0$ and $t = t_0$. Then, since $1/(\eta\lambda) \leq \sqrt{K}$, it holds that

$$\sum_{\kappa=0}^{k-1} \frac{1}{b_{\kappa+1}} \|w_{I(\kappa+1,t,s)} - w_{I(\kappa,t,s)}\|^2 \leq \frac{1}{b} \sum_{i \in \{I(0,t,s),\ldots,I(k-1,t,s)\}\setminus\{\widetilde{I}\}} \|w_{i+1} - w_i\|^2 + \frac{\eta^2 r_0^2}{b}$$
$$\leq \frac{1}{b} \sum_{i \in \{I(0,t,s),\ldots,I(k-1,t,s)\}\setminus\{\widetilde{I}\}} \|w_{i+1} - w_i\|^2 + \frac{\eta^4 \lambda^2 K r_0^2}{b}$$

and

$$\sum_{\kappa=0}^{k-1} \|w_{I(\kappa+1,t,s)} - w_{I(\kappa,t,s)}\| \leq \sum_{i \in \{I(0,t,s),\ldots,I(k-1,t,s)\}\setminus\{\widetilde{I}\}} \|w_{i+1} - w_i\| + \eta r_0$$
$$\leq \sum_{i \in \{I(0,t,s),\ldots,I(k-1,t,s)\}\setminus\{\widetilde{I}\}} \|w_{i+1} - w_i\| + \eta^2 \lambda K r_0.$$

Also, $\sum_{\tau=0}^{t-1} \|w_{I(0,\tau+1,s)} - w_{I(0,\tau,s)}\|^2 = 0$.

Next, suppose that $s = s_0$ and $t > t_0$. Since $I(0,t,s) > I(k_0,t_0,s_0)$, $\|w_{I(k_0+1,t_0,s_0)} - w_{I(k_0,t_0,s_0)}\|$ does not appear in the two terms

$$\sum_{\kappa=0}^{k-1} \frac{1}{b_{\kappa+1}} \|w_{I(\kappa+1,t,s)} - w_{I(\kappa,t,s)}\|^2 \leq \frac{1}{b} \sum_{\kappa=0}^{k-1} \|w_{I(\kappa+1,t,s)} - w_{I(\kappa,t,s)}\|^2$$

and

$$\sum_{\kappa=0}^{k-1} \|w_{I(\kappa+1,t,s)} - w_{I(\kappa,t,s)}\|.$$

Also, since $I(k_0, t_0, s_0) > I(0, 0, s)$,

$$\sum_{\tau=0}^{t-1} \|w_{I(0,\tau+1,s)} - w_{I(0,\tau,s)}\|^2 = \sum_{\tau=\{0,\dots,t-1\}\setminus\{t_0\}} \|w_{I(0,\tau+1,s)} - w_{I(0,\tau,s)}\|^2 + \|w_{I(0,t_0+1,s)} - w_{I(0,t_0,s)}\|^2$$

$$\leq K \sum_{i\in\{I(0,0,s),\dots,I(0,t,s)-1\}\setminus\{I(0,t_0,s_0),\dots,I(0,t_0+1,s_0)-1\}} \|w_{i+1} - w_i\|^2$$

$$+ 2K \sum_{i\in\{I(0,t_0,s),\dots,I(0,t_0+1,s)-1\}\setminus\{I(k_0,t_0,s_0)\}} \|w_{i+1} - w_i\|^2 + 2\eta^2 r_0^2$$

$$\leq 2K \sum_{i\in\{I(0,0,s),\dots,I(0,t,s)-1\}\setminus\{I(k_0,t_0,s_0)\}} \|w_{i+1} - w_i\|^2 + 2\eta^2 r_0^2$$

$$= 2K \sum_{i\in\{I(0,0,s),\dots,I(0,t,s)-1\}\setminus\{\widetilde{I}\}} \|w_{i+1} - w_i\|^2 + 2\eta^2 r_0^2$$

Finally, when $s > s_0$, $\|w_{\widetilde{I}+1} - w_{\widetilde{I}}\|$ never appears in the bound of $\|y_I\|$.

**Case II.** $\sqrt{K} < 1/(\eta\lambda) \leq K$**.**

In this case, $\widetilde{I} = I(k_0', t_0, s_0) - 1$, where $k_0'$ is the minimum number that satisfies $k_0' > k_0$ and $k_0' \equiv 0 \pmod{\lceil\sqrt{K}\rceil}$. Note that $b_{k_0'} = \lceil\sqrt{K}\rceil b$.

Suppose that $s = s_0$ and $t = t_0$. Then, since $1/(\eta\lambda) \leq K$, it holds that

$$\sum_{\kappa=0}^{k-1} \frac{1}{b_{\kappa+1}} \|w_{I(\kappa+1,t,s)} - w_{I(\kappa,t,s)}\|^2 \leq \frac{1}{b} \sum_{i\in\{I(0,t,s),\dots,I(k-1,t,s)\}\setminus\{\widetilde{I}\}} \|w_{i+1} - w_i\|^2 + \frac{\eta^2 r_0^2}{\sqrt{K}b}$$

and

$$\sum_{\kappa=0}^{k-1} \|w_{I(\kappa+1,t,s)} - w_{I(\kappa,t,s)}\| \leq \sum_{i\in\{I(0,t,s),\dots,I(k-1,t,s)\}\setminus\{\widetilde{I}\}} \|w_{i+1} - w_i\| + \eta r_0$$

$$\leq \sum_{i\in\{I(0,t,s),\dots,I(k-1,t,s)\}\setminus\{\widetilde{I}\}} \|w_{i+1} - w_i\| + \eta^2 \lambda K r_0.$$

Also, $\sum_{\tau=0}^{t-1} \|w_{I(0,\tau+1,s)} - w_{I(0,\tau,s)}\|^2 = 0$.

Next, suppose that $s = s_0$ and $t > t_0$. Since $I(0, t, s) > I(k_0, t_0, s_0)$, $\|w_{I(k_0+1,t_0,s_0)} - w_{I(k_0,t_0,s_0)}\|$ does not appear in the two terms

$$\sum_{\kappa=0}^{k-1} \frac{1}{b_{\kappa+1}} \|w_{I(\kappa+1,t,s)} - w_{I(\kappa,t,s)}\|^2 \leq \frac{1}{b} \sum_{\kappa=0}^{k-1} \|w_{I(\kappa+1,t,s)} - w_{I(\kappa,t,s)}\|^2$$

and

$$\sum_{\kappa=0}^{k-1} \|w_{I(\kappa+1,t,s)} - w_{I(\kappa,t,s)}\|.$$

Also, similar to Case I, since $I(k_0', t_0, s_0) > I(0, 0, s)$,

$$\sum_{\tau=0}^{t-1} \|w_{I(0,\tau+1,s)} - w_{I(0,\tau,s)}\|^2 \leq 2K \sum_{i\in\{I(0,0,s),\dots,I(0,t,s)-1\}\setminus\{I(k_0,t_0,s_0)\}} \|w_{i+1} - w_i\|^2 + 2\eta^2 r_0^2$$

$$= 2K \sum_{i\in\{I(0,0,s),\dots,I(0,t,s)-1\}\setminus\{\widetilde{I}\}} \|w_{i+1} - w_i\|^2 + 2\eta^2 r_0^2.$$

Finally, when $s > s_0$, $\|w_{\widetilde{I}+1} - w_{\widetilde{I}}\|$ never appears in the bound of $\|y_I\|$.

**Case III.** $K < 1/(\eta\lambda) \leq KT$.

In this case, $\widetilde{I} = I(0, t_0 + 1, s_0) - 1$. Since $I + 1 = I(k, t, s) > \widetilde{I}$, if $s = s_0$, then we can see that $t \geq t_0 + 1 > t_0$. Then, $\|w_{\widetilde{I}+1} - w_{\widetilde{I}}\|$ does not appear in the two terms

$$\sum_{\kappa=0}^{k-1} \frac{1}{b_{\kappa+1}} \|w_{I(\kappa+1,t,s)} - w_{I(\kappa,t,s)}\|^2 \leq \frac{1}{b} \sum_{\kappa=0}^{k-1} \|w_{I(\kappa+1,t,s)} - w_{I(\kappa,t,s)}\|^2$$

and

$$\sum_{\kappa=0}^{k-1} \|w_{I(\kappa+1,t,s)} - w_{I(\kappa,t,s)}\|.$$

Observe that

$$
\begin{aligned}
\sum_{\tau=0}^{t-1} \|w_{I(0,\tau+1,s)} - w_{I(0,\tau,s)}\|^2 &= \sum_{\tau=\{0,\ldots,t-1\}\setminus\{t_0\}} \|w_{I(0,\tau+1,s)} - w_{I(0,\tau,s)}\|^2 + \|w_{I(0,t_0+1,s)} - w_{I(0,t_0,s)}\|^2 \\
&\leq K \sum_{i \in \{I(0,0,s),\ldots,I(0,t,s)-1\}\setminus\{I(0,t_0,s_0),\ldots,I(0,t_0+1,s_0)-1\}} \|w_{i+1} - w_i\|^2 \\
&\quad + 2K \sum_{i \in \{I(0,t_0,s),\ldots,I(0,t_0+1,s)-2\}} \|w_{i+1} - w_i\|^2 + 2\eta^2 r_0^2 \\
&\leq 2K \sum_{i \in \{I(0,0,s),\ldots,I(0,t,s)-1\}\setminus\{\widetilde{I}\}} \|w_{i+1} - w_i\|^2 + 2\eta^2 r_0^2.
\end{aligned}
$$

When $s > s_0$, $\|w_{\widetilde{I}+1} - w_{\widetilde{I}}\|$ never appears in the bound of $\|y_{I+1}\|$.

**Case IV.** $KT < 1/(\eta\lambda)$.

In this case, $\widetilde{I} = I(0, 0, s_0 + 1) - 1$. Since $I + 1 = I(k, t, s) > \widetilde{I}$, we know that $s \geq s_0 + 1 > s_0$. Hence, $\|w_{\widetilde{I}+1} - w_{\widetilde{I}}\|$ never appears in the bound of $\|y_{I+1}\|$.

In summary, we have

$$
\begin{aligned}
&\|y_{I+1}\| \\
&\leq \Bigg\{ c \sqrt{8L^2 \left( \frac{1}{b} \sum_{i \in \{I(0,t,s),\ldots,I(k-1,t,s)\}\setminus\{\widetilde{I}\}} \|w_{i+1} - w_i\|^2 + \frac{\left(\eta^2\lambda^2 K + 1/\sqrt{K}\right)\eta^2 r_0^2}{b} \right) + 32K\rho^2 U_\Delta^2 U_w (I+1)^2 + \widetilde{\varepsilon}'} \\
&\quad + \zeta \left( \sum_{i \in \{I(0,t,s),\ldots,I(k-1,t,s)\}\setminus\{\widetilde{I}\}} \|w_{i+1} - w_i\| + \eta^2\lambda K r_0 \right) + 4\rho U_\Delta U_w (I+1) \\
&\quad + \frac{c}{\sqrt{PKb}} \sqrt{8L^2 \left( 2K \sum_{i \in \{I(0,0,s),\ldots,I(0,t,s)-1\}\setminus\{\widetilde{I}\}} \|w_{i+1} - w_i\|^2 + 2\eta^2 r_0^2 \right) + 32T\rho^2 U_\Delta^2 U_w (I+1)^2 + \widetilde{\varepsilon}'} \Bigg\} \\
&\quad \times \sqrt{\log \frac{2KTSd}{q} + \log\log \frac{8KTG}{\widetilde{\varepsilon}'}}.
\end{aligned}
$$

Now, we bound $\|w_{I(\kappa+1,\tau,s)} - w_{I(\kappa,\tau,s)}\|$ for the case $I(\kappa+1,\tau,s) > \widetilde{I} + 1$.

$$\|w_{I(\kappa+1,\tau,s)} - w_{I(\kappa,\tau,s)}\|$$

$$= \left\| \eta(1-\eta\mathcal{H})^{I(\kappa+1,\tau,s)-\widetilde{I}}\hat{\xi}_{\widetilde{I}} - \eta \sum_{i=\widetilde{I}}^{I(\kappa,\tau,s)} (1-\eta\mathcal{H})^{I(\kappa,\tau,s)-i}(\Delta_i w_i + y_i) \right.$$

$$\left. - \eta(1-\eta\mathcal{H})^{I(\kappa,\tau,s)-\widetilde{I}}\hat{\xi}_{\widetilde{I}} + \eta \sum_{i=\widetilde{I}}^{I(\kappa,\tau,s)-1} (1-\eta\mathcal{H})^{I(\kappa,\tau,s)-1-i}(\Delta_i w_i + y_i) \right\|$$

$$= \left\| -\eta^2\mathcal{H}(1-\eta\mathcal{H})^{I(\kappa,t,s)-\widetilde{I}}\hat{\xi}_{\widetilde{I}} \right.$$

$$\left. + \eta \sum_{i=\widetilde{I}}^{I(\kappa,\tau,s)-1} \eta\mathcal{H}(1-\eta\mathcal{H})^{I(\kappa,\tau,s)-1-i}(\Delta_i w_i + y_i) - \eta(\Delta_{I(\kappa,\tau,s)}w_{I(\kappa,\tau,s)} + y_{I(\kappa,\tau,s)}) \right\|$$

$$\leq \eta \left\| \eta\mathcal{H}(1-\eta\mathcal{H})^{I(\kappa,t,s)-\widetilde{I}}\hat{\xi}_{\widetilde{I}} \right\|$$

$$+ \eta \sum_{i=\widetilde{I}}^{I(\kappa,\tau,s)-1} \left\| \eta\mathcal{H}(1-\eta\mathcal{H})^{I(\kappa,\tau,s)-1-i} \right\| \|\Delta_i w_i + y_i\| + \eta\|\Delta_{I(\kappa,\tau,s)}w_{I(\kappa,\tau,s)} + y_{I(\kappa,\tau,s)}\|$$

$$\leq \eta^2\lambda(1+\eta\lambda)^{I(\kappa,t,s)-\widetilde{I}}r_0$$

$$+ \eta \sum_{i=\widetilde{I}}^{I(\kappa,\tau,s)-1} \left( \eta\lambda(1+\eta\lambda)^{I(\kappa,t,s)-1-i} + \frac{e}{I(\kappa,t,s)-i} \right) \|\Delta_i w_i + y_i\| + \eta\|\Delta_{I(\kappa,\tau,s)}w_{I(\kappa,\tau,s)} + y_{I(\kappa,\tau,s)}\|.$$

For the second inequality, we used the following two facts:

$$\left\| \eta\mathcal{H}(1-\eta\mathcal{H})^J\hat{\xi}_{\widetilde{I}} \right\| \leq \eta\lambda(1+\eta\lambda)^J\|\hat{\xi}_{\widetilde{I}}\|$$

and

$$\left\| \eta\mathcal{H}(1-\eta\mathcal{H})^J \right\| \leq \eta\lambda(1+\eta\lambda)^J + \frac{e}{J+1}$$

for $J \in \mathbb{N} \cup \{0\}$. The former inequality holds because $\hat{\xi}_{\widetilde{I}} = 2\langle \xi_{\widetilde{I}}, e_{\min}\rangle e_{\min}$ and $e_{\min}$ is the minimum eigenvector of $\mathcal{H}$. The latter inequality is the direct result of the from Lemma B.1.

Then, we further bound the upper bound as follows:

$$\|w_{I(\kappa+1,\tau,s)} - w_{I(\kappa,\tau,s)}\|$$

$$\leq \eta^2\lambda(1+\eta\lambda)^{I(\kappa,t,s)-\widetilde{I}}r_0$$

$$+ \eta \sum_{i=\widetilde{I}}^{I(\kappa,\tau,s)-1} \left( \eta\lambda(1+\eta\lambda)^{I(\kappa,t,s)-1-i} + \frac{e}{I(\kappa,t,s)-i} \right) \|\Delta_i w_i + y_i\| + \eta\|\Delta_{I(\kappa,\tau,s)}w_{I(\kappa,\tau,s)} + y_{I(\kappa,\tau,s)}\|$$

$$\leq \eta^2\lambda(1+\eta\lambda)^{I(\kappa,t,s)-\widetilde{I}}r_0$$

$$+ 4e(1+\log\mathcal{J})\eta\rho U_\Delta U_w(I) + 2e(1+\log\mathcal{J})\eta(1+\eta\lambda\mathcal{J})U_y(I).$$

$$\leq 4e(1+\log\mathcal{J})\eta\rho U_\Delta U_w(I) + \left( \frac{1}{c_{\text{upper}}^{(y)}\left(L + \frac{\sqrt{K}L}{\sqrt{b}} + K\zeta + \frac{\sqrt{K T}L}{\sqrt{Pb}}\right)} + 2e(1+\log\mathcal{J})\eta(1+\eta\lambda\mathcal{J}) \right) U_y(I)$$

$$=: U_{\hat{w}}(I).$$

For the first inequality, we used $\|\Delta_i\| \leq \rho U_\Delta$, the inductive assumptions on $\|w_i\|$ and $\|y_i\|$ for $i \leq I(k,t,s)-1$ and $\sum_{i=i_0}^{i'} 1/(i+1-i_0) \leq 1 + \log(i'+1-i_0)$ for $i' \geq i_0$.

Concretely, we computed

$$\sum_{i=\widetilde{I}}^{I(\kappa,\tau,s)-1} \left(\eta\lambda(1+\eta\lambda)^{I(\kappa,t,s)-1-i} + \frac{e}{I(\kappa,t,s)-i}\right) \|\Delta_i w_i\|$$

$$\leq \sum_{i=\widetilde{I}}^{I(\kappa,\tau,s)-1} \left(\eta\lambda(1+\eta\lambda)^{I(\kappa,t,s)-1-i} + \frac{e}{I(\kappa,t,s)-i}\right) \rho U_\Delta U_w(i)$$

$$\leq \rho U_\Delta (1 + e(1+\log\mathcal{J})) U_w(I)$$

$$\leq 2e(1+\log\mathcal{J})\rho U_\Delta U_w(I).$$

Also, we computed

$$\sum_{i=\widetilde{I}}^{I(\kappa,\tau,s)-1} \left(\eta\lambda(1+\eta\lambda)^{I(\kappa,t,s)-1-i} + \frac{e}{I(\kappa,t,s)-i}\right) \|y_i\|$$

$$\leq \sum_{i=\widetilde{I}}^{I(\kappa,\tau,s)-1} \left(\eta\lambda(1+\eta\lambda)^{I(\kappa,t,s)-1-i} + \frac{e}{I(\kappa,t,s)-i}\right)$$

$$\times \left(c^{(y)}_{\mathrm{upper}}\eta^2\lambda\left(L + \frac{\sqrt{K}L}{\sqrt{b}} + K\zeta + \frac{\sqrt{KT}L}{\sqrt{Pb}}\right)(1+\eta\lambda)^{i-\widetilde{I}}r_0\right)$$

$$\leq c^{(y)}_{\mathrm{upper}}\eta^3\lambda^2\mathcal{J}\left(L + \frac{\sqrt{K}L}{\sqrt{b}} + K\zeta + \frac{\sqrt{KT}L}{\sqrt{Pb}}\right)(1+\eta\lambda)^{I(\kappa,t,s)-1-\widetilde{I}}r_0$$

$$+ c^{(y)}_{\mathrm{upper}}e(1+\log\mathcal{J})\eta^2\lambda\left(L + \frac{\sqrt{K}L}{\sqrt{b}} + K\zeta + \frac{\sqrt{KT}L}{\sqrt{Pb}}\right)(1+\eta\lambda)^{I(\kappa,t,s)-\widetilde{I}}r_0$$

$$\leq e(1+\log\mathcal{J})(1+\eta\lambda\mathcal{J})U_y(I).$$

Using the bound of $\|w_{I(\kappa+1,\tau,s)} - w_{I(\kappa,\tau,s)}\|$, we get

$$\|y_{I+1}\|$$

$$\leq \left\{ c\sqrt{8L^2\left(\frac{K}{b}U_{\hat{w}}(I) + \frac{\left(\eta^2\lambda^2 K + 1/\sqrt{K}\right)\eta^2 r_0^2}{b}\right) + 32K\rho^2 U_\Delta^2 U_w(I+1)^2 + \widetilde{\varepsilon}'}\right.$$

$$+ \zeta\left(KU_{\hat{w}}(I) + \eta^2\lambda K r_0\right) + 4\rho U_\Delta U_w(I+1)$$

$$+ \left.\frac{c}{\sqrt{PKb}}\sqrt{8L^2\left(2K^2 T U_{\hat{w}}(I) + 2\eta^2 r_0^2\right) + 32T\rho^2 U_\Delta^2 U_w(I+1)^2 + \widetilde{\varepsilon}'}\right\}$$

$$\times \sqrt{\log\frac{2KTSd}{q} + \log\log\frac{8KTG}{\widetilde{\varepsilon}'}}$$

$$\leq \left\{\left(\frac{2\sqrt{2}c\sqrt{K}L}{\sqrt{b}} + K\zeta + \frac{4c\sqrt{KT}L}{\sqrt{PKb}}\right)U_{\hat{w}}(I) + \left(\frac{2\sqrt{2}c\eta\lambda\sqrt{K}L}{\sqrt{b}} + \frac{2\sqrt{2}cL}{K^{1/4}\sqrt{b}} + \eta\lambda K\zeta + \frac{4cL}{\sqrt{PKb}}\right)\eta r_0\right.$$

$$+ \left.\left(\frac{4\sqrt{2}c\sqrt{K}}{\sqrt{b}} + 4 + \frac{4\sqrt{2}c\sqrt{T}}{\sqrt{PKb}}\right)\rho U_\Delta U_w(I+1) + 2c\sqrt{\widetilde{\varepsilon}'}\right\}$$

$$\times \sqrt{\log\frac{2KTSd}{q} + \log\log\frac{8KTG}{\widetilde{\varepsilon}'}}$$

Under $b \geq 1/(K^{1/2}\eta^2(L + \sqrt{K}L/\sqrt{b} + K\zeta + \sqrt{KT}L/\sqrt{Pb})^2\rho\varepsilon)$, from $\lambda \geq \sqrt{\rho\varepsilon}$, we have

$$\frac{2\sqrt{2}c\eta\lambda\sqrt{K}L}{\sqrt{b}} + \frac{2\sqrt{2}cL}{K^{1/4}\sqrt{b}} + \eta\lambda K\zeta + \frac{4cL}{\sqrt{PKb}}$$

$$\leq \eta\left(\frac{2\sqrt{2}c\sqrt{K}L}{\sqrt{b}} + 2\sqrt{2}c\left(L + \frac{\sqrt{K}L}{\sqrt{b}} + \frac{\sqrt{KT}L}{\sqrt{Pb}}\right) + K\zeta + 4c\left(L + \frac{\sqrt{K}L}{\sqrt{b}} + \frac{\sqrt{KT}L}{\sqrt{Pb}}\right)\right)\lambda$$

$$\leq 12c\eta\left(L + \frac{\sqrt{K}L}{\sqrt{b}} + K\zeta + \frac{\sqrt{KT}L}{\sqrt{Pb}}\right)\lambda.$$

Also, under $b \geq K$ and $b \geq T/(PK)$, we have

$$\frac{4\sqrt{2}c\sqrt{K}}{\sqrt{b}} + 4 + \frac{4\sqrt{2}c\sqrt{T}}{\sqrt{PKb}} \leq 24c.$$

We choose $\widetilde{\varepsilon}$ such that $\widetilde{\varepsilon}' \leq \eta^4 L^2\lambda^2 r_0^2/(64(\log\frac{2KTSd}{q} + \log\log\frac{8KTG}{\widetilde{\varepsilon}'})c^2)$. Then, it holds that

$$\|y_{I+1}\| \leq \left\{4c\left(\frac{\sqrt{K}L}{\sqrt{b}} + K\zeta + \frac{\sqrt{KT}L}{\sqrt{PKb}}\right)U_{\hat{w}}(I) + 12c\eta\left(L + \frac{\sqrt{K}L}{\sqrt{b}} + K\zeta + \frac{\sqrt{KT}L}{\sqrt{Pb}}\right)\eta\lambda r_0\right.$$

$$\left. + 24c\rho U_\Delta U_w(I+1)\right\} \times \sqrt{\log\frac{2KTSd}{q} + \log\log\frac{8KTG}{\widetilde{\varepsilon}'}} + 0.25U_y(I+1).$$

From the definition of $U_{\hat{w}}(I)$:

$$U_{\hat{w}}(I) := 4e(1 + \log\mathcal{J})\eta\rho U_\Delta U_w(I) + \left(\frac{1}{c^{(y)}_{\text{upper}}\left(L + \frac{\sqrt{K}L}{\sqrt{b}} + K\zeta + \frac{\sqrt{KT}L}{\sqrt{Pb}}\right)} + 2e(1 + \log\mathcal{J})\eta(1 + \eta\lambda\mathcal{J})\right)U_y(I),$$

we get

$$\|y_{I+1}\| \leq \left\{4c\left(\frac{1}{c^{(y)}_{\text{upper}}} + 2e(1 + \log\mathcal{J})\eta\left(\frac{\sqrt{K}L}{\sqrt{b}} + K\zeta + \frac{\sqrt{KT}L}{\sqrt{PKb}}\right)(1 + \eta\lambda\mathcal{J})\right)U_y(I)\right.$$

$$+ 12c\eta\left(L + \frac{\sqrt{K}L}{\sqrt{b}} + K\zeta + \frac{\sqrt{KT}L}{\sqrt{Pb}}\right)\eta\lambda r_0$$

$$\left. + \left(24c + 16ce(1 + \log\mathcal{J})\eta\left(\frac{\sqrt{K}L}{\sqrt{b}} + K\zeta + \frac{\sqrt{KT}L}{\sqrt{PKb}}\right)\right)\rho U_\Delta U_w(I+1)\right\}$$

$$\times \sqrt{\log\frac{2KTSd}{q} + \log\log\frac{8KTG}{\widetilde{\varepsilon}'}} + 0.25U_y(I+1).$$

From the definitions of $\mathcal{J}$ and $U_\Delta$ with $r = c_r\varepsilon$ with $c_r = \widetilde{O}(1)$, we have

$$\rho U_\Delta U_w(I) \leq \rho U_\Delta \frac{c^{(w)}_{\text{upper}}}{c^{(y)}_{\text{upper}}\eta\lambda\left(L + \frac{\sqrt{K}L}{\sqrt{b}} + K\zeta + \frac{\sqrt{KT}L}{\sqrt{Pb}}\right)}U_y(I+1)$$

$$\leq \frac{4c_r\sqrt{c_\mathcal{F} + 2}c_\mathcal{J}\eta\rho\varepsilon}{\eta\lambda^2}\frac{c^{(w)}_{\text{upper}}}{c^{(y)}_{\text{upper}}\eta\left(L + \frac{\sqrt{K}L}{\sqrt{b}} + K\zeta + \frac{\sqrt{KT}L}{\sqrt{Pb}}\right)}U_y(I+1)$$

$$\leq \frac{4c_r\sqrt{c_\mathcal{F} + 2}c_\mathcal{J}}{\eta\left(L + \frac{\sqrt{K}L}{\sqrt{b}} + K\zeta + \frac{\sqrt{KT}L}{\sqrt{Pb}}\right)}U_y(I+1).$$

Here, for the last inequality, we used $\lambda \geq \sqrt{\rho\varepsilon}$ and $c_{\text{upper}}^{(y)} = c_{\text{upper}}^{(w)}$.

Therefore, we arrive at

$$\|y_{I+1}\| \leq \left\{ 4c \left( \frac{1}{c_{\text{upper}}^{(y)}} + 2e(1 + \log\mathcal{J})\eta \left( \frac{\sqrt{K}L}{\sqrt{b}} + K\zeta + \frac{\sqrt{KT}L}{\sqrt{PKb}} \right) (1 + \eta\lambda\mathcal{J}) \right) U_y(I) \right.$$

$$+ \frac{12c}{c_{\text{upper}}^{(y)}} U_y(I+1)$$

$$+ \left. \left( \frac{96cc_r\sqrt{c_{\mathcal{F}} + 2c_{\mathcal{J}}}}{\eta \left( L + \frac{\sqrt{K}L}{\sqrt{b}} + K\zeta + \frac{\sqrt{KT}L}{\sqrt{Pb}} \right)} + 64ce(1 + \log\mathcal{J})c_r\sqrt{c_{\mathcal{F}} + 2c_{\mathcal{J}}} \right) U_y(I+1) \right\}$$

$$\times \sqrt{\log\frac{2KTSd}{q} + \log\log\frac{8KTG}{\widetilde{\varepsilon}'}} + 0.25U_y(I+1).$$

We set $c_{\text{upper}}^{(y)} = c_{\text{upper}}^{(w)} = \max\{3, 48c\sqrt{\log\frac{2KTSd}{q} + \log\log\frac{8KTG}{\widetilde{\varepsilon}'}}\}$. Then, since $\eta\lambda\mathcal{J} \leq c_{\mathcal{J}}$, if we choose $\eta$ such that $\eta(L + \sqrt{K}L/\sqrt{b} + K\zeta + \sqrt{KT}L/\sqrt{Pb})) \leq 1/(48ce(1 + c_{\mathcal{J}})(1 + \log\mathcal{J})\sqrt{\log\frac{2KTSd}{q} + \log\log\frac{8KTG}{\widetilde{\varepsilon}'}})$, the first term can be bounded by $0.25U_y(I+1)$.

Next, from the definition of $c_{\text{upper}}^{(y)}$, we can see that the second term is bounded by $0.25U_y(I+1)$.

Finally, we can choose $\eta$ such that $\eta(L + \sqrt{K}L/\sqrt{b} + K\zeta + \sqrt{KT}L/\sqrt{Pb})) \geq \widetilde{\Theta}(\sqrt{c_\eta} + 1/(c_{\mathcal{J}}))$. Then if we appropriately choose $c_r \leq \widetilde{O}((\sqrt{c_\eta} + 1/c_{\mathcal{J}})/(\sqrt{c_{\mathcal{F}} + 2c_{\mathcal{J}}}))$, the third term can be bounded by $0.25U_y(I+1)$. Therefore, we conclude that

$$\|y_{I+1}\| \leq U_y(I+1).$$

This finishes the proof of the mathematical induction.

Let $\widetilde{\mathcal{J}} := \mathcal{J} - (\widetilde{I} - I(k_0, t_0, s_0))$. From (9), (12) and (13), we have

$$\|w_{I(k_0,t_0,s_0)+\mathcal{J}}\| = \left\| \eta(1 - \eta\mathcal{H})^{\widetilde{\mathcal{J}}}\hat{\xi}_{\widetilde{I}} - \eta\sum_{i=\widetilde{I}}^{\widetilde{I}+\widetilde{\mathcal{J}}} (1 - \eta\mathcal{H})^{\widetilde{I}+\widetilde{\mathcal{J}}-i}(\Delta_i w_i + y_i) \right\|$$

$$\geq \|\eta(1 - \eta\mathcal{H})^{\widetilde{\mathcal{J}}}\hat{\xi}_{\widetilde{I}}\|$$

$$- \left\| \eta\sum_{i=\widetilde{I}}^{\widetilde{I}+\widetilde{\mathcal{J}}} (1 - \eta\mathcal{H})^{\widetilde{I}+\widetilde{\mathcal{J}}-i}\Delta_i w_i \right\|$$

$$- \left\| \eta\sum_{i=\widetilde{I}}^{\widetilde{I}+\widetilde{\mathcal{J}}} (1 - \eta\mathcal{H})^{\widetilde{I}+\widetilde{\mathcal{J}}-i}y_i \right\|$$

$$\geq \eta(1 + \eta\lambda)^{\widetilde{\mathcal{J}}}r_0 - \frac{1}{3c_{\text{upper}}^{(w)}}U_w(I(k_0, t_0, s_0) + \mathcal{J})$$

$$= \frac{2\eta(1 + \eta\lambda)^{\widetilde{\mathcal{J}}}r_0}{3}.$$

Now, we define $c_{\mathcal{J}}$ as the minimum positive number that satisfies

$$c_{\mathcal{J}} \geq 1 + 2\log(48\sqrt{c_{\mathcal{F}} + 2}\mathcal{J}\sqrt{d}/q).$$

From (6), we can see that

$$\frac{2\eta(1 + \eta\lambda)^{\widetilde{\mathcal{J}}}r_0}{3} \geq 4U_\Delta.$$

This is because we have

$$\log\left((1+\eta\lambda)^{\widetilde{\mathcal{J}}}\right) = \widetilde{\mathcal{J}}\log(1+\eta\lambda)$$

$$\geq \widetilde{\mathcal{J}}\left(1 - \frac{1}{1+\eta\lambda}\right)$$

$$\geq \frac{\eta\lambda\widetilde{\mathcal{J}}}{2}$$

$$\geq \frac{\eta\lambda(\mathcal{J} - 1/(\eta\lambda))}{2}$$

$$= \frac{c_{\mathcal{J}} - 1}{2}$$

$$\geq \log(48\sqrt{c_{\mathcal{F}} + 2}\mathcal{J}\sqrt{d}/q)$$

and thus

$$\frac{2\eta(1+\eta\lambda)^{\widetilde{\mathcal{J}}}r_0}{3} \geq \frac{\eta(1+\eta\lambda)^{\widetilde{\mathcal{J}}}qr}{3\sqrt{d}}$$

$$\geq 4 \times 4\sqrt{c_{\mathcal{F}} + 2}\eta\mathcal{J}r = 4U_\Delta.$$

Here, the first inequality holds from (6). This contradicts with $\|w_{I(k_0,t_0,s_0)+\mathcal{J}}\| \leq 2U_\Delta$.

$\square$

**Proof of Proposition 4.3**

Now, we prove Proposition 4.3. Combining Proposition B.3 with Proposition B.4, we have

$$\min\{f(x_{I(k_0,t_0,s_0)+\mathcal{J}_{I(k_0,t_0,s_0)}}) - f(x_{I(k_0,t_0s_0)}), f(x'_{I(k_0,t_0,s_0)+\mathcal{J}_{I(k_0,t_0,s_0)}}) - f(x'_{I(k_0,t_0s_0)})\}$$

$$\leq -\mathcal{F}_{I(k_0,t_0,s_0)}$$

$$+ \frac{2c_\eta}{\eta}\left(\frac{\mathcal{J}_{I(k_0,t_0,s_0)} \wedge K}{K}\sum_{i=I(0,t_0,s_0)}^{I(k_0,t_0,s_0)-1}\|x_{i+1} - x_i\|^2 + \frac{\mathcal{J}_{I(k_0,t_0,s_0)} \wedge KT}{KT}\sum_{i=I(0,0,s_0)}^{I(0,t_0,s_0)-1}\|x_{i+1} - x_i\|^2\right).$$

with probability at least $1 - 9q$.

Finally, since $\{x_i\}_{i=0}^{KTS}$ has the same marginal distribution as $\{x'_i\}_{i=0}^{KTS}$, we conclude that

$$f(x_{I(k_0,t_0,s_0)+\mathcal{J}_{I(k_0,t_0,s_0)}}) - f(x_{I(k_0,t_0s_0)})$$

$$\leq -\mathcal{F}_{I(k_0,t_0,s_0)}$$

$$+ \frac{2c_\eta}{\eta}\left(\frac{\mathcal{J}_{I(k_0,t_0,s_0)} \wedge K}{K}\sum_{i=I(0,t_0,s_0)}^{I(k_0,t_0,s_0)-1}\|x_{i+1} - x_i\|^2 + \frac{\mathcal{J}_{I(k_0,t_0,s_0)} \wedge KT}{KT}\sum_{i=I(0,0,s_0)}^{I(0,t_0,s_0)-1}\|x_{i+1} - x_i\|^2\right).$$

(16)

with probability at least $1/2 - 9q/2$. This finishes the proof of Proposition 4.3. $\square$

### B.5 Finding Second Order Stationary Points

Let $\mathcal{R}_1 := \{x \in \mathbb{R}^d | \|\nabla f(x)\| > \varepsilon\}$, $\mathcal{R}_2 := \{x \in \mathbb{R}^d | \|\nabla f(x)\| \leq \varepsilon \wedge \lambda_{\min}(\nabla^2 f(x)) < -\sqrt{\rho\varepsilon}\}$ and $\mathcal{R}_3 := \mathbb{R}^d \setminus (\mathcal{R}_1 \cup \mathcal{R}_2) = \{x \in \mathbb{R}^d | \|\nabla f(x)\| \leq \varepsilon \wedge \lambda_{\min}(\nabla^2 f(x)) \geq -\sqrt{\rho\varepsilon}\}$.

We define

$$\iota_{m+1} = \begin{cases} \iota_m + 1 & (\widetilde{x}_{\iota_m} \in \mathcal{R}_1 \cup \mathcal{R}_3) \\ \iota_m + \mathcal{J}_{\iota_m} & (\widetilde{x}_{\iota_m} \in \mathcal{R}_2) \end{cases}$$

with $\iota_1 := 0$. Note that $\mathcal{J}_{\iota_m} \leq c_{\mathcal{J}}/(\eta\sqrt{\rho\varepsilon})$. Let $M := \min\{m \in \mathbb{N} | \mathbb{E}[\iota_m] \geq KTS/8\}$. Observe that $\iota_M \leq M \times c_{\mathcal{J}}/(\eta\sqrt{\rho\varepsilon}) \leq (KTS/8) \times c_{\mathcal{J}}/(\eta\sqrt{\rho\varepsilon})$ because $\iota_{KTS/8} \geq KTS/8$ always

holds. We define $\check{S}$ as the minimum number that satisfies $\check{S} \geq (S/8) \times c_{\mathcal{J}}/(\eta\sqrt{\rho\varepsilon})) \vee S$ with $S = \Theta(1 + (f(\widetilde{x}_0) - f(x_*))/(\eta KT\varepsilon^2))$, where in the definition of $\eta$ we set $S \leftarrow \check{S}$. Then $\iota_M \leq KT\check{S}$ always holds. We will use Propositions 4.1 and 4.3 with $S \leftarrow \check{S}$. $s(\iota_m)$ denotes the maximum natural number $s'$ satisfying $\iota_m \geq I(0,0,s')$ and $t(\iota_m)$ denotes the maximum natural number $t'$ satisfying $\iota_m \geq I(0,t',s(\iota_m))$. We will show that $\widetilde{x}_i \in \mathcal{R}_3$ for some $i \in [KTS]\cup\{0\}$ with probability at least $1/2$. Let $E_i$ is the event that $\widetilde{x}_{i'} \notin \mathcal{R}_3$ for all $i' \leq i$ for $i \in [KTS] \cup \{0\}$. Note that $E_{i+1} \subset E_i$ for every $i$. We can say that the objective of this section is to show $\mathbb{P}(E_{KTS}) \leq 1/2$.

**Proposition B.5.** *Suppose that Assumptions 1, 2, 3, 4 and 5 hold. Under $K = O(L/\zeta \wedge b \wedge Pb/T)$, if we appropriately choose $\eta = \widetilde{\Theta}(1/L \wedge 1/(K\zeta) \wedge \sqrt{b/K}/L \wedge \sqrt{Pb}/(\sqrt{KT}L))$ and $r = \Theta(\varepsilon)$, then it holds that*

$$\frac{7\eta}{512}\mathbb{E}[\iota_M]\varepsilon^2 \leq f(\widetilde{x}_0) - f(x_*) + \frac{\eta}{64}\sum_{m=1}^{M-1}\mathbb{P}(\widetilde{x}_{\iota_m} \in \mathcal{R}_3)\varepsilon^2.$$

**Proof of Proposition B.5**

First, we consider the difference $\mathbb{E}[f(x_{\iota_{m+1}}) - f(x_{\iota_m})]$.

**Bounding** $\mathbb{E}[f(x_{\iota_{m+1}}) - f(x_{\iota_m})|\widetilde{x}_{\iota_m} \in \mathcal{R}_1]$

Let $H_1$ be the event where (4) with $I(k_0,t_0,s_0) \leftarrow \iota_m$ and $I(k,t,s) \leftarrow \iota_{m+1}$ holds. Note that $\mathbb{P}(H_1|\widetilde{x}_{\iota_m} \in \mathcal{R}_1) \geq 1 - 3q$. From Proposition 4.3 and (5), we have for every $q \in (0, 1/6)$,

$$\mathbb{E}[f(x_{\iota_{m+1}}) - f(x_{\iota_m})|\widetilde{x}_{\iota_m} \in \mathcal{R}_1]$$
$$= \mathbb{E}[f(x_{\iota_{m+1}}) - f(x_{\iota_m})|\widetilde{x}_{\iota_m} \in \mathcal{R}_1, H_1]\mathbb{P}(H_1|\widetilde{x}_{\iota_m} \in \mathcal{R}_1)$$
$$\quad + \mathbb{E}[f(x_{\iota_{m+1}}) - f(x_{\iota_m})|\widetilde{x}_{\iota_m} \in \mathcal{R}_1, H_1^{\complement}]\mathbb{P}(H^{\complement}|\widetilde{x}_{\iota_m} \in \mathcal{R}_1)$$
$$\leq -(1-3q)\frac{\eta}{2}\mathbb{E}[\|\nabla f(x_{\iota_m})\|^2|\widetilde{x}_{I(k,t,s)} \in \mathcal{R}_1, H_1] + \eta r^2$$
$$\quad + \frac{c_\eta}{\eta}\mathbb{E}\left[\frac{(\iota_{m+1} - \iota_m) \wedge K}{K}\sum_{i=I(0,t(\iota_m),s(\iota_m))}^{\iota_m-1}\|x_{i+1} - x_i\|^2\right.$$
$$\quad\quad\left. + \frac{(\iota_{m+1} - \iota_m) \wedge KT}{KT}\sum_{i=I(0,0,s(\iota_m))}^{I(0,t(\iota_m),s(\iota_m))-1}\|x_{i+1} - x_i\|^2\Big|\widetilde{x}_{\iota_m} \in \mathcal{R}_1, H_1\right]\mathbb{P}(H_1|\widetilde{x}_{\iota_m} \in \mathcal{R}_1)$$
$$\quad + 3q \times 36\eta K^2T^2S(G^2 + r^2)$$
$$\leq -\frac{\eta}{8}\varepsilon^2 + 2\eta r^2$$
$$\quad + \frac{c_\eta}{\eta}\mathbb{E}\left[\frac{(\iota_{m+1} - \iota_m) \wedge K}{K}\sum_{i=I(0,t(\iota_m),s(\iota_m))}^{\iota_m-1}\|x_{i+1} - x_i\|^2\right.$$
$$\quad\quad\left. + \frac{(\iota_{m+1} - \iota_m) \wedge KT}{KT}\sum_{i=I(0,0,s(\iota_m))}^{I(0,t(\iota_m),s(\iota_m))-1}\|x_{i+1} - x_i\|^2\Big|\widetilde{x}_{\iota_m} \in \mathcal{R}_1, H_1\right]\mathbb{P}(H_1|\widetilde{x}_{\iota_m} \in \mathcal{R}_1)$$
$$\quad + 3q \times (36\eta K^2T^2S(G^2 + r^2))$$
$$\leq -\frac{\eta}{8}\varepsilon^2 + 2\eta r^2$$
$$\quad + \frac{c_\eta}{\eta}\mathbb{E}\left[\frac{(\iota_{m+1} - \iota_m) \wedge K}{K}\sum_{i=I(0,t(\iota_m),s(\iota_m))}^{\iota_m-1}\|x_{i+1} - x_i\|^2\right.$$
$$\quad\quad\left. + \frac{(\iota_{m+1} - \iota_m) \wedge KT}{KT}\sum_{i=I(0,0,s(\iota_m))}^{I(0,t(\iota_m),s(\iota_m))-1}\|x_{i+1} - x_i\|^2\Big|\widetilde{x}_{\iota_m} \in \mathcal{R}_1\right]$$
$$\quad + 3q \times 36\eta K^2T^2S(G^2 + r^2).$$

For the second inequality, we used $1/(1 - 3q) \leq 1/2$ and $-\|\nabla f(x_{I(k,t,s)})\|^2 \leq -(1/2)\|\nabla f(\widetilde{x}_{I(k,t,s)})\|^2 + \|\nabla f(x_{I(k,t,s)}) - f(\widetilde{x}_{I(k,t,s)})\|^2 \leq -(1/2)\|\nabla f(\widetilde{x}_{I(k,t,s)})\|^2 + \eta^2 L^2 r^2 \leq -(1/2)\|\nabla f(\widetilde{x}_{I(k,t,s)})\|^2 + r^2$ since $\eta \leq 1/L$.

Thus, setting $q := (\eta \varepsilon^2/16)/(96 K^2 T^2 S(G^2 + \eta r^2))$ and $c_r \leq 1/\sqrt{96}$, we get

$$
\mathbb{E}[f(x_{\iota_{m+1}}) - f(x_{\iota_m})|\widetilde{x}_{\iota_m} \in \mathcal{R}_1]
$$
$$
\leq - \frac{\eta}{16}\varepsilon^2 + 3\eta r^2
$$
$$
+ \frac{c_\eta}{\eta}\mathbb{E}\left[\frac{(\iota_{m+1} - \iota_m) \wedge K}{K} \sum_{i=I(0,t(\iota_m),s(\iota_m))}^{\iota_m - 1} \|x_{i+1} - x_i\|^2 \right.
$$
$$
\left. + \frac{(\iota_{m+1} - \iota_m) \wedge KT}{KT} \sum_{i=I(0,0,s(\iota_m))}^{I(0,t(\iota_m),s(\iota_m))-1} \|x_{i+1} - x_i\|^2 | \widetilde{x}_{\iota_m} \in \mathcal{R}_1\right]
$$
$$
\leq - \frac{\eta}{32}\mathbb{E}[\iota_{m+1} - \iota_m|\widetilde{x}_{i_m} \in \mathcal{R}_1]\varepsilon^2
$$
$$
+ \frac{c_\eta}{\eta}\mathbb{E}\left[\frac{(\iota_{m+1} - \iota_m) \wedge K}{K} \sum_{i=I(0,t(\iota_m),s(\iota_m))}^{\iota_m - 1} \|x_{i+1} - x_i\|^2 \right.
$$
$$
\left. + \frac{(\iota_{m+1} - \iota_m) \wedge KT}{KT} \sum_{i=I(0,0,s(\iota_m))}^{I(0,t(\iota_m),s(\iota_m))-1} \|x_{i+1} - x_i\|^2 | \widetilde{x}_{\iota_m} \in \mathcal{R}_1\right]. \tag{17}
$$

Here, we used $\mathbb{E}[\iota_{m+1} - \iota_m|\widetilde{x}_{i_m} \in \mathcal{R}_1] = 1$.

**Bounding** $\mathbb{E}[f(x_{\iota_{m+1}}) - f(x_{\iota_m})|\widetilde{x}_{\iota_m} \in \mathcal{R}_2]$

$H_2$ denotes the event where (16) with $I(k_0, t_0, s_0) \leftarrow \iota_m$ holds. Note that $\mathbb{P}(H_2|\widetilde{x}_{\iota_m} \in \mathcal{R}_2) \geq 1/2 - 7q/2$ by Proposition 4.3. Let $q \in (0, 1/14)$ and with $c_{\mathcal{F}} \geq 16$. We will use Proposition 4.3, (8) and (5).

$$
\mathbb{E}[f(x_{\iota_{m+1}}) - f(x_{\iota_m})|\widetilde{x}_{\iota_m} \in \mathcal{R}_2]
$$
$$
= \mathbb{E}[f(x_{\iota_{m+1}}) - f(x_{\iota_m})|\widetilde{x}_{\iota_m} \in \mathcal{R}_2, H_2]\mathbb{P}(H_2|\widetilde{x}_{\iota_m} \in \mathcal{R}_2)
$$
$$
+ \mathbb{E}[f(x_{\iota_{m+1}}) - f(x_{\iota_m})|\widetilde{x}_{\iota_m} \in \mathcal{R}_2, H_1, H_2^{\complement}]P(H_1, H_2^{\complement}|\widetilde{x}_{\iota_m} \in \mathcal{R}_2)
$$
$$
+ \mathbb{E}[f(x_{\iota_{m+1}}) - f(x_{\iota_m})|\widetilde{x}_{\iota_m} \in \mathcal{R}_2, H_1^{\complement}, H_2^{\complement}]P(H_1^{\complement}, H_2^{\complement}|\widetilde{x}_{\iota_m} \in \mathcal{R}_2).
$$

The first term can be bouded as

$$
\mathbb{E}[f(x_{\iota_{m+1}}) - f(x_{\iota_m})|\widetilde{x}_{\iota_m} \in \mathcal{R}_2, H_2]\mathbb{P}(H_2|\widetilde{x}_{\iota_m} \in \mathcal{R}_2)
$$
$$
\leq \left\{ -\mathbb{E}[\mathcal{F}_{\iota_m}|\widetilde{x}_{\iota_m} \in \mathcal{R}_2, H_2] \right.
$$
$$
+ \frac{2c_\eta}{\eta}\mathbb{E}\left[\frac{(\iota_{m+1} - \iota_m) \wedge K}{K} \sum_{i=I(0,t(\iota_m),s(\iota_m))}^{\iota_m - 1} \|x_{i+1} - x_i\|^2 \right.
$$
$$
\left. \left. + \frac{(\iota_{m+1} - \iota_m) \wedge KT}{KT} \sum_{i=I(0,0,s(\iota_m))}^{I(0,t(\iota_m),s(\iota_m))-1} \|x_{i+1} - x_i\|^2 | \widetilde{x}_{\iota_m} \in \mathcal{R}_2, H_2\right]\right\}\mathbb{P}(H_2|x_{\iota_m} \in \mathcal{R}_2).
$$

Here, the inequality holds from Proposition 4.3. The second term can be bounded as

$$
\mathbb{E}[f(x_{\iota_{m+1}}) - f(x_{\iota_m})|\widetilde{x}_{\iota_m} \in \mathcal{R}_2, H_1, H_2^{\complement}]P(H_1, H_2^{\complement}|\widetilde{x}_{\iota_m} \in \mathcal{R}_2)
$$

$$
\leq \left\{ \frac{2}{c_{\mathcal{F}}}\mathbb{E}[\mathcal{F}_{\iota_m}|\widetilde{x}_{\iota_m} \in \mathcal{R}_2, H_1, H_2^{\complement}] \right.
$$

$$
+ \frac{c_\eta}{\eta}\mathbb{E}\left[ \frac{(\iota_{m+1} - \iota_m) \wedge K}{K} \sum_{i=I(0,t(\iota_m),s(\iota_m))}^{\iota_m - 1} \|x_{i+1} - x_i\|^2 \right.
$$

$$
\left. \left. + \frac{(\iota_{m+1} - \iota_m) \wedge KT}{KT} \sum_{i=I(0,0,s(\iota_m))}^{I(0,t(\iota_m),s(\iota_m))-1} \|x_{i+1} - x_i\|^2|\widetilde{x}_{\iota_m} \in \mathcal{R}_2, H_1, H_2^{\complement} \right] \right\} \mathbb{P}(H_1, H_2^{\complement}|\widetilde{x}_{\iota_m} \in \mathcal{R}_2).
$$

Here, we used (8).

Finally, the last term can be bounded as

$$
\mathbb{E}[f(x_{\iota_{m+1}}) - f(x_{\iota_m})|\widetilde{x}_{\iota_m} \in \mathcal{R}_2, H_1^{\complement}, H_2^{\complement}]P(H_1^{\complement}, H_2^{\complement}|\widetilde{x}_{\iota_m} \in \mathcal{R}_2)
$$

$$
\leq 36\eta K^2 T^2 S(G^2 + r^2)\mathbb{P}(H_1^{\complement}, H_2^{\complement}|\widetilde{x}_{\iota_m} \in \mathcal{R}_2).
$$

From these bounds, we have

$$
\mathbb{E}[f(x_{\iota_{m+1}}) - f(x_{\iota_m})|\widetilde{x}_{\iota_m} \in \mathcal{R}_2]
$$

$$
\leq -\left( \frac{1}{2} - \frac{7q}{2} - \frac{2}{c_{\mathcal{F}}} \right)\mathbb{E}[\mathcal{F}_{\iota_m}|\widetilde{x}_{\iota_m} \in \mathcal{R}_2]
$$

$$
+ \frac{3c_\eta}{\eta}\mathbb{E}\left[ \frac{(\iota_{m+1} - \iota_m) \wedge K}{K} \sum_{i=I(0,t(\iota_m),s(\iota_m))}^{\iota_m - 1} \|x_{i+1} - x_i\|^2 \right.
$$

$$
\left. + \frac{(\iota_{m+1} - \iota_m) \wedge KT}{KT} \sum_{i=I(0,0,s(\iota_m))}^{I(0,t(\iota_m),s(\iota_m))-1} \|x_{i+1} - x_i\|^2|\widetilde{x}_{\iota_m} \in \mathcal{R}_2 \right]
$$

$$
+ 3q \times 36\eta K^2 T^2 S(G^2 + r^2)
$$

$$
\leq -\frac{c_{\mathcal{F}}c_r^2\eta}{8}\mathbb{E}[\mathcal{J}_{\iota_m}|\widetilde{x}_{\iota_m} \in \mathcal{R}_2]\varepsilon^2
$$

$$
+ \frac{3c_\eta}{\eta}\mathbb{E}\left[ \frac{(\iota_{m+1} - \iota_m) \wedge K}{K} \sum_{i=I(0,t(\iota_m),s(\iota_m))}^{\iota_m - 1} \|x_{i+1} - x_i\|^2 \right.
$$

$$
\left. + \frac{(\iota_{m+1} - \iota_m) \wedge KT}{KT} \sum_{i=I(0,0,s(\iota_m))}^{I(0,t(\iota_m),s(\iota_m))-1} \|x_{i+1} - x_i\|^2|\widetilde{x}_{\iota_m} \in \mathcal{R}_2 \right]
$$

$$
+ 3q \times 36\eta K^2 T^2 S(G^2 + r^2)
$$

Here, for the first inequality, we used the facts that $\mathcal{F}_{\iota_m}$ only depends on the start point $\widetilde{x}_{\iota_m} \in \mathcal{R}_2$ and does not depend on $H_2$, which only captures the randomness after iteration index $\iota_m$, and $\mathbb{P}(H_2|\widetilde{x}_{\iota_m} \in \mathcal{R}_2, E_{\iota_m}) \geq 1/2 - 7q/2$. For the last inequality, we used $\mathcal{F}_{I(k,t,s)} = c_{\mathcal{F}}\eta\mathcal{J}_{I(k,t,s)}r^2$ with $c_{\mathcal{F}} \geq 16$ and $r = c_r\varepsilon^2$.

Thus, setting $q := (c_{\mathcal{F}} c_r^2 \eta \varepsilon^2/16)/(96 K^2 T^2 S(G^2 + \eta r^2))$, we get

$$
\mathbb{E}[f(x_{\iota_{m+1}}) - f(x_{\iota_m})|\widetilde{x}_{\iota_m} \in \mathcal{R}_2]
$$
$$
\leq -\frac{c_{\mathcal{F}} c_r^2 \eta}{8} \mathbb{E}[\mathcal{J}_{\iota_m}|\widetilde{x}_{\iota_m} \in \mathcal{R}_2]\varepsilon^2
$$
$$
+ \frac{3c_\eta}{\eta} \mathbb{E}\left[ \frac{(\iota_{m+1} - \iota_m) \wedge K}{K} \sum_{i=I(0,t(\iota_m),s(\iota_m))}^{\iota_m - 1} \|x_{i+1} - x_i\|^2 \right.
$$
$$
\left. + \frac{(\iota_{m+1} - \iota_m) \wedge KT}{KT} \sum_{i=I(0,0,s(\iota_m))}^{I(0,t(\iota_m),s(\iota_m))-1} \|x_{i+1} - x_i\|^2 \Big| \widetilde{x}_{\iota_m} \in \mathcal{R}_2 \right]
$$
$$
+ \frac{c_{\mathcal{F}} c_r^2 \eta \varepsilon^2}{16}
$$
$$
= -\frac{c_{\mathcal{F}} c_r^2 \eta}{16} \mathbb{E}[\iota_{m+1} - \iota_m|\widetilde{x}_{\iota_m} \in \mathcal{R}_2]\varepsilon^2
$$
$$
+ \frac{3c_\eta}{\eta} \mathbb{E}\left[ \frac{(\iota_{m+1} - \iota_m) \wedge K}{K} \sum_{i=I(0,t(\iota_m),s(\iota_m))}^{\iota_m - 1} \|x_{i+1} - x_i\|^2 \right.
$$
$$
\left. + \frac{(\iota_{m+1} - \iota_m) \wedge KT}{KT} \sum_{i=I(0,0,s(\iota_m))}^{I(0,t(\iota_m),s(\iota_m))-1} \|x_{i+1} - x_i\|^2 \Big| \widetilde{x}_{\iota_m} \in \mathcal{R}_2 \right] \tag{18}
$$

**Bounding** $\mathbb{E}[f(x_{\iota_{m+1}}) - f(x_{\iota_m})|\widetilde{x}_{\iota_m} \in \mathcal{R}_3]$

Similar to the arguments for bounding $\mathbb{E}[f(x_{\iota_{m+1}}) - f(x_{\iota_m})|\widetilde{x}_{\iota_m} \in \mathcal{R}_1]$, we have

$$
\mathbb{E}[f(x_{\iota_{m+1}}) - f(x_{\iota_m})|\widetilde{x}_{\iota_m} \in \mathcal{R}_3]
$$
$$
\leq 3\eta r^2 + \frac{c_\eta}{\eta} \mathbb{E}\left[ \frac{(\iota_{m+1} - \iota_m) \wedge K}{K} \sum_{i=I(0,t(\iota_m),s(\iota_m))}^{\iota_m - 1} \|x_{i+1} - x_i\|^2 \right.
$$
$$
\left. + \frac{(\iota_{m+1} - \iota_m) \wedge KT}{KT} \sum_{i=I(0,0,s(\iota_m))}^{I(0,t(\iota_m),s(\iota_m))-1} \|x_{i+1} - x_i\|^2 \Big| \widetilde{x}_{\iota_m} \in \mathcal{R}_3 \right]
$$
$$
= -\left( \frac{\eta}{32} \wedge \frac{c_{\mathcal{F}} c_r^2 \eta}{16} \right) \mathbb{E}[\iota_{m+1} - \iota_m|\widetilde{x}_{\iota_m} \in \mathcal{R}_3] + \frac{\eta}{32} \wedge \frac{c_{\mathcal{F}} c_r^2 \eta}{16} + 3\eta r^2
$$
$$
+ \frac{c_\eta}{\eta} \mathbb{E}\left[ \frac{(\iota_{m+1} - \iota_m) \wedge K}{K} \sum_{i=I(0,t(\iota_m),s(\iota_m))}^{\iota_m - 1} \|x_{i+1} - x_i\|^2 \right.
$$
$$
\left. + \frac{(\iota_{m+1} - \iota_m) \wedge KT}{KT} \sum_{i=I(0,0,s(\iota_m))}^{I(0,t(\iota_m),s(\iota_m))-1} \|x_{i+1} - x_i\|^2 \Big| \widetilde{x}_{\iota_m} \in \mathcal{R}_3 \right]. \tag{19}
$$

Here, we used the fact that $\mathbb{E}[\iota_{m+1} - \iota_m|\widetilde{x}_{i_m} \in \mathcal{R}_3]\mathbb{P}(\widetilde{x}_{\iota_m} \in \mathcal{R}_3) = \mathbb{P}(\widetilde{x}_{\iota_m} \in \mathcal{R}_3)$.

Hence, combining (17), (18) and (19) yields

$$
\mathbb{E}[f(x_{\iota_{m+1}}) - f(x_{\iota_m})]
$$

$$
\leq - \left( \frac{\eta}{32} \wedge \frac{c_{\mathcal{F}} c_r^2 \eta}{16} \right) \mathbb{E}[\iota_{m+1} - \iota_m] \varepsilon^2 + \left( \frac{\eta}{32} \wedge \frac{c_{\mathcal{F}} c_r^2 \eta}{16} + 3\eta c_r^2 \right) \mathbb{P}(\widetilde{x}_{\iota_m} \in \mathcal{R}_3) \varepsilon^2
$$

$$
+ \frac{3c_\eta}{\eta} \mathbb{E} \left[ \frac{(\iota_{m+1} - \iota_m) \wedge K}{K} \sum_{i=I(0,t(\iota_m),s(\iota_m))}^{\iota_m - 1} \|x_{i+1} - x_i\|^2 \right.
$$

$$
\left. + \frac{(\iota_{m+1} - \iota_m) \wedge KT}{KT} \sum_{i=I(0,0,s(\iota_m))}^{I(0,t(\iota_m),s(\iota_m))-1} \|x_{i+1} - x_i\|^2 \right]
$$

$$
\leq - \frac{c_{\mathcal{F}} c_r^2 \eta}{16} \mathbb{E}[\iota_{m+1} - \iota_m] \varepsilon^2 + \left( \frac{c_{\mathcal{F}} c_r^2 \eta}{16} + 3\eta c_r^2 \right) \mathbb{P}(\widetilde{x}_{\iota_m} \in \mathcal{R}_3) \varepsilon^2
$$

$$
+ \frac{3c_\eta}{\eta} \mathbb{E} \left[ \frac{(\iota_{m+1} - \iota_m) \wedge K}{K} \sum_{i=I(0,t(\iota_m),s(\iota_m))}^{\iota_m - 1} \|x_{i+1} - x_i\|^2 \right.
$$

$$
\left. + \frac{(\iota_{m+1} - \iota_m) \wedge KT}{KT} \sum_{i=I(0,0,s(\iota_m))}^{I(0,t(\iota_m),s(\iota_m))-1} \|x_{i+1} - x_i\|^2 \right]
$$

under $c_r \leq 1/\sqrt{2c_{\mathcal{F}}}$. Summing this inequality from $m = 1$ to $M - 1$ results in

$$
\mathbb{E}[f(x_{\iota_M}) - f(x_0)]
$$

$$
\leq - \frac{c_{\mathcal{F}} c_r^2 \eta}{16} \mathbb{E}[\iota_M] \varepsilon^2 + \left( \frac{c_{\mathcal{F}} c_r^2 \eta}{16} + 3\eta c_r^2 \right) \sum_{m=1}^{M-1} \mathbb{P}(\widetilde{x}_{\iota_m} \in \mathcal{R}_3) \varepsilon^2
$$

$$
+ \frac{3c_\eta}{\eta} \mathbb{E} \left[ \sum_{m=1}^{M-1} \frac{(\iota_{m+1} - \iota_m) \wedge K}{K} \sum_{i=I(0,t(\iota_m),s(\iota_m))}^{\iota_m - 1} \|x_{i+1} - x_i\|^2 \right.
$$

$$
\left. + \frac{(\iota_{m+1} - \iota_m) \wedge KT}{KT} \sum_{i=I(0,0,s(\iota_m))}^{I(0,t(\iota_m),s(\iota_m))-1} \|x_{i+1} - x_i\|^2 \right]. \tag{20}
$$

Here, we used the definition $\iota_1 = 0$.

By the way, from (4) and (5), we can also derive a different bound for $\mathbb{E}[f(x_{\iota_{m+1}}) - f(x_{\iota_m})]$. For every $q \in (0, 1/6)$, we have

$$
\mathbb{E}[f(x_{\iota_{m+1}}) - f(x_{\iota_m})]
$$

$$
= \mathbb{E}[f(x_{\iota_{m+1}}) - f(x_{\iota_m})|H_1]\mathbb{P}(H_1)
$$

$$
+ \mathbb{E}[f(x_{\iota_{m+1}}) - f(x_{\iota_m})|H_1^{\complement}]\mathbb{P}(H^{\complement})
$$

$$
\leq - (1 - 3q)\frac{1}{8\eta} \mathbb{E} \left[ \sum_{i=\iota_m}^{\iota_{m+1}-1} \|x_{i+1} - x_i\|^2 |H_1 \right] \mathbb{P}(H_1) + 2\eta r^2 \mathbb{E}[\iota_{m+1} - \iota_m|H_1]\mathbb{P}(H_1)
$$

$$
+ \frac{c_\eta}{\eta} \mathbb{E} \left[ \frac{(\iota_{m+1} - \iota_m) \wedge K}{K} \sum_{i=I(0,t(\iota_m),s(\iota_m))}^{\iota_m - 1} \|x_{i+1} - x_i\|^2 \right.
$$

$$
\left. + \frac{(\iota_{m+1} - \iota_m) \wedge KT}{KT} \sum_{i=I(0,0,s(\iota_m))}^{I(0,t(\iota_m),s(\iota_m))-1} \|x_{i+1} - x_i\|^2 |H_1 \right] \mathbb{P}(H_1)
$$

$$
+ 3q \times 36\eta K^2 T^2 S(G + r^2).
$$

Observe that

$$-\mathbb{E}\left[\sum_{i=\iota_m}^{\iota_{m+1}-1}\|x_{i+1}-x_i\|^2|H_1\right]\mathbb{P}(H_1)$$

$$=-\mathbb{E}\left[\sum_{i=\iota_m}^{\iota_{m+1}-1}\|x_{i+1}-x_i\|^2\right]+\mathbb{E}\left[\sum_{i=\iota_m}^{\iota_{m+1}-1}\|x_{i+1}-x_i\|^2|H_1^{\complement}\right]P(H_1^{\complement})$$

$$\leq-\mathbb{E}\left[\sum_{i=\iota_m}^{\iota_{m+1}-1}\|x_{i+1}-x_i\|^2\right]+3q\times192\eta^2(KTG^2+r^2)$$

Here, for the inequality, we used

$$\|x_{i+1}-x_i\|^2\leq3\eta^2\|v_i-\nabla f(x_i)\|^2+3\eta^2\|\nabla f(x_I)\|^2+3\eta^2r^2$$
$$\leq96\eta^2KTG^2+3\eta^2G^2+3\eta^2r^2$$
$$\leq192\eta^2(KTG^2+r^2)$$

Hence, with $q:=\eta r^2/\{(96K^2T^2S(G+\eta r^2)+72\eta(KTG^2+r^2)(c_{\mathcal{J}}/(\eta\sqrt{\rho\varepsilon}))\}$ we get

$$\mathbb{E}[f(x_{\iota_{m+1}})-f(x_{\iota_m})]$$

$$\leq-\frac{1}{16\eta}\mathbb{E}\left[\sum_{i=\iota_m}^{\iota_{m+1}-1}\|x_{i+1}-x_i\|^2\right]+2\eta r^2\mathbb{E}[\iota_{m+1}-\iota_m]$$

$$+\frac{c_\eta}{\eta}\mathbb{E}\left[\frac{(\iota_{m+1}-\iota_m)\wedge K}{K}\sum_{i=I(0,t(\iota_m),s(\iota_m)}^{\iota_m-1}\|x_{i+1}-x_i\|^2\right.$$

$$\left.+\frac{(\iota_{m+1}-\iota_m)\wedge KT}{KT}\sum_{i=I(0,0,s(\iota_m))}^{I(0,t(\iota_m),s(\iota_m))-1}\|x_{i+1}-x_i\|^2\right]$$

$$+\eta r^2$$

$$\leq-\frac{1}{16\eta}\mathbb{E}\left[\sum_{i=\iota_m}^{\iota_{m+1}-1}\|x_{i+1}-x_i\|^2\right]+3\eta r^2\mathbb{E}[\iota_{m+1}-\iota_m]$$

$$+\frac{2c_\eta}{\eta}\mathbb{E}\left[\frac{(\iota_{m+1}-\iota_m)\wedge K}{K}\sum_{i=I(0,t(\iota_m),s(\iota_m)}^{\iota_m-1}\|x_{i+1}-x_i\|^2\right.$$

$$\left.+\frac{(\iota_{m+1}-\iota_m)\wedge KT}{KT}\sum_{i=I(0,0,s(\iota_m))}^{I(0,t(\iota_m),s(\iota_m))-1}\|x_{i+1}-x_i\|^2\right].$$

Summing this inequality from $m=1$ to $M-1$ gives

$$\mathbb{E}[f(x_{\iota_M})-f(x_0)]$$

$$\leq-\frac{1}{16\eta}\sum_{m=1}^{M-1}\mathbb{E}\left[\sum_{i=\iota_m}^{\iota_{m+1}-1}\|x_{i+1}-x_i\|^2\right]+3\eta r^2\mathbb{E}[\iota_M]$$

$$+\frac{c_\eta}{\eta}\mathbb{E}\left[\sum_{m=1}^{M-1}\frac{(\iota_{m+1}-\iota_m)\wedge K}{K}\sum_{i=I(0,t(\iota_m),s(\iota_m)}^{\iota_m-1}\|x_{i+1}-x_i\|^2\right.$$

$$\left.+\frac{(\iota_{m+1}-\iota_m)\wedge KT}{KT}\sum_{i=I(0,0,s(\iota_m))}^{I(0,t(\iota_m),s(\iota_m))-1}\|x_{i+1}-x_i\|^2\right]$$

Combining this inequality with (20), with we obtain

$$
\begin{aligned}
\mathbb{E}&[f(x_{\iota_M}) - f(x_0)]\\
&\le -\frac{1}{2}\left(\frac{c_{\mathcal{F}}c_r^2\eta}{16} - 3\eta c_r^2\right)\mathbb{E}[\iota_M]\varepsilon^2 + \frac{1}{2}\left(\frac{c_{\mathcal{F}}c_r^2\eta}{16} + 3\eta c_r^2\right)\sum_{m=1}^{M-1}\mathbb{P}(\widetilde{x}_{\iota_m}\in\mathcal{R}_3)\varepsilon^2\\
&\quad -\frac{1}{16\eta}\mathbb{E}\left[\sum_{i=0}^{\iota_M-1}\|x_{i+1}-x_i\|^2\right]\\
&\quad +\frac{2c_\eta}{\eta}\mathbb{E}\left[\sum_{m=1}^{M-1}\left(\frac{(\iota_{m+1}-\iota_m)\wedge K}{K}\sum_{i=I(0,t(\iota_m),s(\iota_m)}^{\iota_m-1}\|x_{i+1}-x_i\|^2\right.\right.\\
&\quad\quad\left.\left.+\frac{(\iota_{m+1}-\iota_m)\wedge KT}{KT}\sum_{i=I(0,0,s(\iota_m))}^{I(0,t(\iota_m),s(\iota_m))-1}\|x_{i+1}-x_i\|^2\right)\right].
\end{aligned}
$$

We want to show that

$$
\begin{aligned}
\sum_{i=0}^{\iota_M-1}&\|x_{i+1}-x_i\|^2\\
&\ge\frac{1}{4}\sum_{m=1}^{M-1}\left(\frac{(\iota_{m+1}-\iota_m)\wedge K}{K}\sum_{i=I(0,t(\iota_m),s(\iota_m))}^{\iota_m-1}\|x_{i+1}-x_i\|^2\right.\\
&\quad\quad\left.+\frac{(\iota_{m+1}-\iota_m)\wedge KT}{KT}\sum_{i=I(0,0,s(\iota_m))}^{I(0,t(\iota_m),s(\iota_m))-1}\|x_{i+1}-x_i\|^2\right).
\end{aligned}
$$

To prove this inequality, we fix $i'\in[\iota_M-1]\cup\{0\}$ and show that the coefficient of $\|x_{i'+1}-x_{i'}\|^2$ of the left hand side is greater than or equal to the one of the right hand side. At first, the coefficient of $\|x_{i'+1}-x_{i'}\|^2$ of the left hand side is trivially 1. Next we consider the right hand side. Let $s'$ be the natural number that satisfies $I(0,0,s')\le i'<I(0,0,s'+1)$. Also, $t'$ be the natural number that satisfies $I(0,t',s')\le i'<I(0,t'+1,s')$. We define $\boldsymbol{m}_1 := \{m\in\mathbb{N}|I(0,t',s')\le\iota_m<I(0,t'+1,s')\}$ and $\boldsymbol{m}_2 := \{m\in\mathbb{N}|I(0,0,s')\le\iota_m<I(0,0,s'+1)\}$. We can see that the coefficient of $\|x_{i'+1}-x_{i'}\|^2$ in the right hand side is

$$
\begin{aligned}
\frac{1}{4}&\left(\sum_{m=1}^{M-1}\frac{(\iota_{m+1}-\iota_m)\wedge K}{K}\mathbb{1}_{I(0,t(\iota_m),s(\iota_m))\le i'\le\iota_m-1}\right.\\
&\quad\left.+\sum_{m=1}^{M-1}\frac{(\iota_{m+1}-\iota_m)\wedge KT}{KT}\mathbb{1}_{I(0,0,s(\iota_m))\le i'\le I(0,t(\iota_m),s(\iota_m))-1}\right)\\
&\le\frac{1}{4}\left(\sum_{m\in\boldsymbol{m}_1}\frac{(\iota_{m+1}-\iota_m)\wedge K}{K}+\sum_{m\in\boldsymbol{m}_2}\frac{(\iota_{m+1}-\iota_m)\wedge KT}{KT}\right)\\
&=\frac{1}{4}\left(1+\sum_{m\in\boldsymbol{m}_1\setminus\{\max\{\boldsymbol{m}_1\}\}}\frac{(\iota_{m+1}-\iota_m)\wedge K}{K}+1+\sum_{m\in\boldsymbol{m}_2\setminus\{\max\{\boldsymbol{m}_2\}\}}\frac{(\iota_{m+1}-\iota_m)\wedge KT}{KT}\right)\\
&\le 1.
\end{aligned}
$$

Here, for the first inequality we used the facts that (i) $I(0,t(\iota_m),s(\iota_m))\le i'\le\iota_m-1)$ implies $m\in\boldsymbol{m}_1$ and (ii) $I(0,0,s(\iota_m))\le i'\le I(0,t(\iota_m),s(\iota_m))-1$ implies $m\in\boldsymbol{m}_2$. To show (i), note that $\iota_m<I(0,t',s')$ implies $\iota_m-1<i'$ and $\iota_m\ge I(0,t'+1,s')$ implies $I(0,t(\iota_m),s(\iota_m))\ge I(0,t'+1,s')>i'$. Similarly, to show (ii), observe that $\iota_m<I(0,0,s')$ implies $i'>\iota_m>I(0,t(\iota_m),s(\iota_m))-1$ and $\iota_m\ge I(0,0,s'+1)$ implies $i'<I(0,0,s'+1)\le I(0,0,s(\iota_m))$. For the last inequality we used $\sum_{m\in\boldsymbol{m}_1\setminus\{\max\{\boldsymbol{m}_1\}\}}(\iota_{m+1}-\iota_m)\le K$ and $\sum_{m\in\boldsymbol{m}_2\setminus\{\max\{\boldsymbol{m}_2\}\}}(\iota_{m+1}-\iota_m)\le KT$.

We choose $c_\eta \le 1/128$. Then, we obtain

$$f(x_*) - f(\widetilde{x}_0) \le \mathbb{E}[f(x_{\iota_M}) - f(x_0)] + \eta r^2$$

$$\le -\left(\frac{c_\mathcal{F} c_r^2 \eta}{32} - 3c_r^2 \eta\right) \mathbb{E}[\iota_M]\varepsilon^2 + (\frac{c_\mathcal{F} c_r^2 \eta}{32} + 3c_r^2 \eta) \sum_{m=1}^{M-1} \mathbb{P}(\widetilde{x}_{\iota_m} \in \mathcal{R}_3)\varepsilon^2.$$

Here, for the first inequality, we used $\mathbb{E}[f(x_{\iota_M})] \ge f(x_*)$ and $\mathbb{E}[f(x_0)] \le f(\widetilde{x}_0) + \langle \nabla f(\widetilde{x}_0), \mathbb{E}[x_0 - \widetilde{x}_0]\rangle + (L/2)\|x_0 - \widetilde{x}_0\|^2 = f(\widetilde{x}_0) + \eta^2 L r^2/2 \le f(\widetilde{x}_0) + \eta r^2$ by the smoothness of $f$. For the second inequality, we used the above bounds with the definition of $c_\eta$ for $\mathbb{E}[f(x_{\iota_M}) - f(x_0)]$. This finishes the proof. $\qquad\square$

## Proof of Theorem 4.4

Now, we choose $S \ge 48(f(x_0) - f(x_*))/(c_r^2 \eta K T \varepsilon^2) = \widetilde{\Theta}((f(x_0) - f(x_*))/(\eta K T \varepsilon^2))$. Note that $\mathbb{E}[\iota_M] \ge KTS/8 \ge 6(f(x_0) - f(x_*))/(c_r^2 \eta \varepsilon^2)$.

Suppose that $\mathbb{P}(\widetilde{x}_{\iota_m} \in \mathcal{R}_3) \le 3/4$ for every $m \in [M-1]$. Then, since $c_\mathcal{F} c_r^2 \eta/32 - 3c_r^2 \eta - (3/4) \times (c_\mathcal{F} c_r^2 \eta/32 + 3c_r^2 \eta) \ge 1/4(c_\mathcal{F}/32 - 21)c_r^2 \eta \ge c_r^2 \eta/4$ under $c_\mathcal{F} \ge 32 \times 22$, we have

$$f(x_*) - f(x_0) \le -\frac{c_r^2 \eta}{4} \mathbb{E}[\iota_M]\varepsilon^2$$

and thus

$$\mathbb{E}[\iota_M] \le \frac{4(f(x_0) - f(x_*))}{c_r^2 \eta \varepsilon^2}$$

from Proposition B.5. This contradicts the previous lower bound of $\mathbb{E}[\iota_M]$. Therefore, we conclude that there exists $m \in [M-1]$ such that $\mathbb{P}(\widetilde{x}_{i_m} \in \mathcal{R}_3) > 3/4$. Remember that $E_i$ is the event that $\widetilde{x}_{i'} \notin \mathcal{R}_3$ for all $i' \le i$. This implies $\mathbb{P}(E_{\iota_{M-1}}^\complement) > 3/4$, and thus $\mathbb{P}(E_{\iota_{M-1}}) \le 1/4$.

Finally, we bound $\mathbb{P}(E_{KTS})$. From the definition of $M$, we have $\mathbb{E}[\iota_{M-1}] < KTS/8$. Thus, from Markov's inequality, it holds that $\mathbb{P}(\iota_{M-1} \ge KTS) \le 1/8$.

This yields

$$\mathbb{P}(E_{KTS}) = \mathbb{P}(E_{KTS}|\iota_{M-1} \ge KTS)\mathbb{P}(\iota_{M-1} \ge KTS) + \mathbb{P}(E_{KTS}|\iota_{M-1} < KTS)\mathbb{P}(\iota_{M-1} < KTS)$$

$$\le 1 \times \frac{1}{8} + \mathbb{P}(E_{\iota_{M-1}})$$

$$\le 1/2.$$

This finishes the proof. $\qquad\square$