# OpenReview forum: "Escaping Saddle Points with Bias-Variance Reduced Local Perturbed SGD for Communication Efficient Nonconvex Distributed Learning"
_NeurIPS.cc/2022/Conference — NeurIPS 2022 Accept_

### Official Review · Reviewer_RYcp · 2022-07-09

**Rating:** 6
**Confidence:** 4
**Soundness:** 3 good
**Presentation:** 3 good
**Contribution:** 3 good

**Summary:**

This work proposes a new local algorithm called Bias-Variance Reduced Local Perturbed SGD (BVR-L-PSGD), via a simple combination of the existing bias and variance reduced gradient estimator and parameter perturbation. BVR-L-PSGD converges to second-order stationary point of non-convex federated optimization problem with nearly the same communication complexity as the best known one of BVR-L-SGD to find first-order stationary point, which is smaller than non-local minibatch SGD when the local datasets heterogeneity is smaller than the smoothness of the local loss. The communication efficiency over BVR-L-SGD is demonstrated via experiments.

**Questions:**

(1) Could you explain what is a local method in introduction? (i.e., the difference between local and non-local methods) Which methods in Table 1 are non-local? Why are local methods in related work not listed in Table 1?

(2) The first sentence in "**Open question**" in the introduction has grammar mistake that hinders understanding. Do you mean "it is not well-studied how to find second-order optimal points with **low communication cost** and **thus** we have the following research question**s**"?

(3) At the beginning of "**Main Contributions**", you might mention something like "which positively answered the above research questions".

(4) Gradient norm bound $G$ is very strong. There are some works that did not use this assumption, like [13,14] cited in your paper. Is it possible use a weaker assumption instead? (e.g. $\frac{1}{P}\sum_{p=1}^{P}\|f_p(x)\|^2\le G^2+B^2\|\nabla f(x)\|^2$ in [13]) Also, do all the works in Table 1 use Assumptions 2-5? This relates to fairness of comparison.

(5) From Table 1, it seems that to let your commmunication complexity is smaller than non-local minibatch SGD, we not only require small $\zeta$, but also small $n$ relative to $\epsilon$, right?

(6) (Minor) In line 136, I think it sufficient to write #supp$(D_p)=n/P$ for every $p\in[P]$.

(7) In Algorithm 1, $x_0=\widetilde{x} _ 0+\eta\xi_{-1}$ seems equivalent to directly initializing $x_0$. How about removing this line and replacing input $\widetilde{x}_0$ with $x_0$?

(8) In Algorithm 1, lines 8-13 can be put in the block "if $t\ge 1$" as we do not need to do these when $t=0$. Acccordingly, line 11 can be simplified. Similarly, lines 16-18 can be put in "if $k\ge 1$" and line 18 can be simplified.

(9) STORM variance reduction does not need large batchsize (you may see Appendix B of [1] for the complexity result that does not need assumption on gradient norm bound), but both achieves the same near-optimal sample complexity as SARAH/SPIDER. They may further reduce complexity and might be a future direction.

[1] Cutkosky, Ashok, and Francesco Orabona. "Momentum-based variance reduction in non-convex sgd." Advances in neural information processing systems 32 (2019).

(10) (Grammar) In line 251, you may use "poly-logarithmic factors **that** depend on".

(11) In line 264, should the total number of communication rounds be $2TS+1$?

(12) The hyperparameter choices in the propostions, theorems and corollaries are expressed using $\widetilde{\Theta}$. It is highly recommended to express their full form either in the main text, or in the appendix and tell the locations right after $\widetilde{\Theta}$.

(13) In Figs. 1-3, the gradient norms of both algorithms increase and converge to 1 (looks very large) instead of 0. How to explain? Also, you may add plot of $-\lambda_{\min} \big(\nabla^2 f(x_t)\big)$ to demonstrate convergence of your proposed algorithm to second-order stationary point. In addition, you have made the claim that your communication complexity is lower than non-local minibatch SGD for $\zeta\ll L$, yes? If so, you would better demonstrate that by comparing with non-local minibatch SGD.

**Ethics Review Area:**

["I don’t know"]

**Limitations:**

I only found the limitation mentioned in the footnote in page 8. Is that correct? Just make sure.
That limitation is also in existing studies and thus I think it fine that this work has not addressed this limitation.

**Strengths And Weaknesses:**

Originality: This work simply combines BVR-L-PSGD and parameter perturbation in algorithm design, but it has better complexity in achieving second-order stationary point for nonconvex federated optimization. Technical novelty is also explicitly expressed. Lit review looks good to me. Therefore, I think the originality is sufficient for Neurips.

Quality: The claims are well supported by theoretical results on communication complexity and partially supported by experiments which have some flaws as I mentioned in my final question.

Clarity: This paper is clearly written and well organized.

Significance: The results are important to me since BVR-L-PSGD converges to second-order stationary point of non-convex federated optimization problem with nearly the same communication complexity as the best known one of BVR-L-SGD to find first-order stationary point, which is smaller than non-local minibatch SGD when the local datasets heterogeneity is smaller than the smoothness of the local loss. The strong gradient norm assumption and flaws in experiment kind of undermines significance, as elaborated in my questions.

---

> ### Author Response · Authors · 2022-08-01
> **Reply to Reviewer RYcp**
>
> Thank you very much for your careful reading and a lot of valuable feedback.
>
> **A to Q (1)**: A local method means the algorithm that runs optimization in the own local dataset in each communication round.
> In Table 1, minibatch SGD, Noisy minibatch SGD and minibatch SARAH are non-local methods and the others are local methods. The reason why the local methods described in related work are not listed in Table 1 is that these methods don't have second-order guarantees and not so popular compared with Local SGD and SCAFFOLD in Table 1. Since our aim is to study second-order guaranteed methods, the most important thing here is the comparison of the communication complexities of the second-order guaranteed methods. Although we can list the rates of the methods in related work, the table becomes quite large and messy. Due to this (and space limitation) we did not list the methods described in related work for simple presentation.
>
> **A to Q (4)**:
> (**Dependence on $G$**) As it says under the sentences of Assumption 5, the communication complexity of our method only depends on gradient norm bound $G$ in logarithmic order. In this sense, the communication complexity of our method is quite robust to $G$. Also, please note that (non-perturbed) BVR-L-SGD does not need this condition for first-order optimality guarantees.
> (**Assumptions of previous methods**) Actually, some work don't use some of Assumptions 2-5. I agree with you and we have clarified the assumptions used in the methods in Table 1 in the revised paper.
>
> **A to Q (5)**: Yes. More precisely, $\varepsilon^2 \leq 1/\sqrt{n}$ is required. The settings of $\varepsilon$ depends on the case, but it is often required that $\varepsilon = O(1/\sqrt{n})$ from the statistical learning theory and thus this condition is not so restricted.
>
> **A to Q (6)**: Right. We have fixed it in the revised paper.
>
> **A to Q (7)**: Thank you for your suggestion. We will follow that.
>
> **A to Q (8)**: You are definitely right. As a matter of fact, we had to reduce the numbers of lines in Algorithm 1 due to the 9 pages limitation and we reluctantly decided to use the notation in line 13 and 18.
>
> **A to Q (9)**: As you said, STORM is a nice variance reduction type algorithm that does not need the gradient norm bound assumption. However, STORM only guarantees first-order optimality and its extension to second-order optimality has not studied yet. The usage of large batch size is a potential limitation of our method and it is an interesting future work to consider a federated variant of STORM with second-order optimality guarantees.
>
> **A to Q (11)**: As you pointed out, we should write $2TS+1$ or $\Theta(TS)$ instead of $TS$. We have fixed this in the revised paper.
>
> **A to Q (12)**: As you said, it is generally nice to express the constants and logarithmic factors hidden in $\widetilde \Theta$ for completeness. However, the proofs are quite complex and the expressions may become exceedingly messy. This is why we use $\widetilde \Theta$ symbol for readability.
>
> **A to Q (13)**:
> (**Gradient norm**) We think that the reason why the gradient norms didn't converge to 1 is simply that the algorithms have not yet converged. In our experiments, we used 1,000 communication rounds, that is not so small in practice but may not enough for convergences. It seems that the used model and dataset require extremely large rounds to observe small gradient norms.
> (**Eigenvalue of Hessian**) As you said, it is desirable to track the minimum eigenvalue of Hessian for each algorithm. However, this process requires huge computational time due to the Hessian computation of the objective function and thus we have given up on conducting that experiment.
> (**Comparison with minibatch SGD**) We agree with your suggestion and we have conducted numerical experiments for minibatch SGD. We have added the results in Figure 1 in the revised paper (we also replaced the results of the proposed method since we additionally tuned the noise radius $r$ for both noisy minibatch SGD and BVR-L-PSGD). We still observed that our proposed method outperformed the baseline methods (particularly, to noisy minibatch SGD).
>
> **A to Q (2), (3), (10)**: Thank you for your pointing that out. We have fixed these grammar issues in the revised paper.
>
> **Q**: I only found the limitation mentioned in the footnote in page 8. Is that correct?
> **A**: Yes. But after reading your review, the usage of large batch size may also be one of the limitations of our method. We believe that removing this limitation is an interesting future direction of this work.

---

> > ### Comment · Reviewer_RYcp · 2022-08-06
> > **Reviewer RYcp is satisfied with the authors' response.**
> >
> > Reviewer RYcp is satisfied with the authors' response and keeps rating 6.
> > For Q13, it is interesting that for small number of communication rounds, the gradient norm is smaller than that for 1k communication rounds.

---

### Official Review · Reviewer_hqiv · 2022-07-10

**Rating:** 6
**Confidence:** 3
**Soundness:** 3 good
**Presentation:** 3 good
**Contribution:** 2 fair

**Summary:**

This paper proposes a new algorithm called Bias-variacne reduced local perturbed SGD for nonconvex distributed learning to efficiently find second-order optimal points, and furthermore, it is proved in this paper that this new algorithm escapes saddle points, which implies the convergence to approximate second-order stationarity.

This result is built on the former 1st-order convergence guarantee of BVR-L-SGD, the main techniques of the approximate second order convergence/escaping saddle points come from the well-known work of Ge. Et al and other related works since 2015. The novelty is clear.


**Questions:**

One question I am interested in is presented above.
Additionally, can the authors discuss a bit on the main challenge comparing to the work of [10]?
Is it possible to get rate of escaping saddle points without perturbation for BVR-L-SGD?

**Limitations:**

The limitation of the result in this paper might be the dependence on the parameters, how to set these parameters in real applications? From theoretical perspective, assuming the Hessian to be Lipschitz is reasonable but this constrains the application of perturbed GD in contrast to the gradient descent which only requires Lipschitz gradient to get saddle avoidance result.

**Strengths And Weaknesses:**

Strength: The main strength is the theoretical contribution, it is good to see the idea of [10] on perturbed SGD can be extended to other settings.

Weaknesses: Considering the complexity of the algorithm itself, it is better if the authors can provide more guidance on the setting of parameters.

It is not clear whether current algorithm really outperforms BVR-L-SGD (without perturbation) in the following sense: Combined with the result of “On the almost sure convergence of stochastic gradient descent in non-convex problems, 2020”, it is not hard to believe that BVR-L-SGD likely avoids saddle points.

 Is it possible to compare the quality of the second order information of the outputs by using these two algorithms, BVR-L-SGD and BVR-L-PSGD?

The authors could have done a better survey on “escaping saddle points” literature, e.g.

Escaping saddle points with adaptive gradient methods, Staib et al.

---

> ### Author Response · Authors · 2022-08-01
> **Reply to Reviewer hqiv**
>
> Thank you for your important questions and positive evaluation.
>
> **Q**: It is not clear whether current algorithm really outperforms BVR-L-SGD (without perturbation) in the following sense: Combined with the result of “On the almost sure convergence of stochastic gradient descent in non-convex problems, 2020”, it is not hard to believe that BVR-L-SGD likely avoids saddle points.
> **A**: We believe that applying the theory in “On the almost sure convergence of stochastic gradient descent in non-convex problems, 2020” to BVR-L-SGD is difficult. This is because the uniformly exciting assumption, which is essentially used to derive convergence rates in the paper, does not seem to hold in variance reduction methods; The uniformly exciting assumption roughly means that a gradient estimator contains some level of uniform noise, but BVR-L-SGD and other variance reduction methods aim to reduce the variance of the gradient estimator's noise and thus the noise size will become very small. Thus, at least in a theoretical point of view, parameter perturbation is necessary for BVR-L-SGD to guarantee second-order optimality. Also, from an empirical point of view, our numerical results show that BVR-L-PSGD much efficiently escaped a stuck region than BVR-L-SGD. This also suggests the effectivity and necessity of parameter perturbation.
>
> **Q**: The limitation of the result in this paper might be the dependence on the parameters, how to set these parameters in real applications? From theoretical perspective, assuming the Hessian to be Lipschitz is reasonable but this constrains the application of perturbed GD in contrast to the gradient descent which only requires Lipschitz gradient to get saddle avoidance result.
> **A**: Please note that the tuning parameters in our algorithm are only learning rate $\eta$ and noise radius $r$, and we don't need to know the actual value (or estimated value) of Hessian Lipschitzness $\rho$. This situation is completely same as the case of noisy gradient descent.

---

### Official Review · Reviewer_sztZ · 2022-07-11

**Rating:** 6
**Confidence:** 4
**Soundness:** 3 good
**Presentation:** 3 good
**Contribution:** 3 good

**Summary:**

The previously proposed method BVR-L-SGD (using bias-variance reduced gradient estimator) is claimed to be one of the best first-order optimality methods for nonconvex distributed learning. This paper integrates the parameter perturbation into the existed BVR-L-SGD, named BVR-L-PSGD, to give it second-order optimality.

**Questions:**

I think the author should clarify and emphasize the significance of such a second-order up-gradation.

**Limitations:**

No potential negative societal impact.

**Strengths And Weaknesses:**

Strengths：
It’s a second-order optimality upgraded version of the existed BVR-L-SGD. This is a nice extension. It will be better if the author can add more contents to show this work is the state-of-the-art second-order method.

Combining the parameter perturbation, which is an easy-installed approach that has been validated by previous works. So, it is intuitive and totally makes sense.

Weaknesses：
There is only one second-order method mentioned in the theoretical baseline.

For the experiment side, it only compares the BVR-L-SGD and BVR-L-PSGD proposed.

---

> ### Author Response · Authors · 2022-08-01
> **Reply to Reviewer sztZ**
>
> Thank you for your helpful comments and positive evaluation.
>
> **Q**: There is only one second-order method mentioned in the theoretical baseline.
> **A**: The most natural baseline is definitely (perturbed) minibatch SGD, which we compare with our method in Table 1. There are many non-local second-order guaranteed methods, e.g., Stabilized SVRG and SSRGD, but these methods never achieve a better rate than $L/\varepsilon^2$, which is in the same situation as minibatch SGD. Hence, in Table 1 we don't list these methods for simple presentation. However, for more completeness, we have added SSRGD to Table 1, which is the state-of-the-art non-local method with second-order optimality guarantees, in the revision paper.
>
> **Q**: For the experiment side, it only compares the BVR-L-SGD and BVR-L-PSGD proposed.
> **A**: To enhance the content of the experimental results, we have added the results of minibatch SGD and perturbed minibatch SGD as a baseline in the revised paper.

---

> > ### Comment · Reviewer_sztZ · 2022-08-06
> > **Response to the author**
> >
> > Thank you for the response and the new content added.
> > I have no further concerns.

---

### Meta-Review · Area_Chair_iopD · 2022-08-21

**Recommendation:** Accept
**Confidence:** Certain

**Metareview:**

The reviewers have reached a consensus of accepting the paper.

**Award:**

No

---

### Decision · Program_Chairs · 2022-09-14

Accept